# Dynamical Decoupling of Generalization and Overfitting in Large Two-Layer Networks

**Andrea Montanari**
Department of Statistics and
Department of Mathematics,
Stanford University

**Pierfrancesco Urbani**
Université Paris-Saclay, CNRS, CEA,
Institut de Physique Théorique,
91191, Gif-Sur-Yvette, France

## Abstract

Understanding the inductive bias and generalization properties of large over-parametrized machine learning models requires to characterize the dynamics of the training algorithm. We study the learning dynamics of large two-layer neural networks via dynamical mean field theory, a well established technique of non-equilibrium statistical physics. We show that, for large network width $m$, and large number of samples per input dimension $n/d$, the training dynamics exhibits a separation of timescales which implies: $(i)$ The emergence of a slow time scale associated with the growth in Gaussian/Rademacher complexity of the network; $(ii)$ Inductive bias towards small complexity if the initialization has small enough complexity; $(iii)$ A dynamical decoupling between feature learning and overfitting regimes; $(iv)$ A non-monotone behavior of the test error, associated 'feature unlearning' regime at large times.

## 1 Introduction

Machine learning (ML) models are trained using stochastic gradient descent (SGD), or one of its variants to minimize the error on training data (empirical risk function). Classically, their good behavior on unseen test data is explained by the fact that model complexity is kept small by regularization techniques: these models do not 'overfit.' Traditional ML theory decouples the analysis of the model from the optimization algorithm, which is assumed to converge to an approximate global minimizer [47].

In contrast, in modern ML, the empirical risk is highly non-convex, the number of parameters is comparable with the number of training samples, and the model complexity is only weakly controlled. As a consequence, there can be many assignments of the model parameters (many global empirical risk minimizers) that perfectly interpolate the data —even when these are noisy. While all of these *interpolators* are indistinguishable on the training data, they behave very differently (and some of them very poorly) on test data. It has been hypothesized that models trained by SGD generalize well to test data because the algorithm selects a near global minimizer with low complexity, although a mechanistic understanding of this process is lacking. For this reason, the generalization properties cannot be decoupled from the training dynamics.

Several striking consequences of this lack of decoupling are documented in the literature (and have long been familiar to practitioners): $(i)$ Test error after training is observed to depend strongly on the initial weights distribution [28]; $(ii)$ Test error depends strongly on the optimization algorithm (SGD, RMSProp, ADAM, to name a few), even when these algorithms achieve the same train error [55]; $(iii)$ Careful choice of the hyperparameters in the optimization algorithm is crucial [34, 59], and the optimal choice is often different from the one that minimizes train error; $(iv)$ Models learned by training for a shorter time have smaller complexity and can generalize better [44, 11].

39th Conference on Neural Information Processing Systems (NeurIPS 2025).

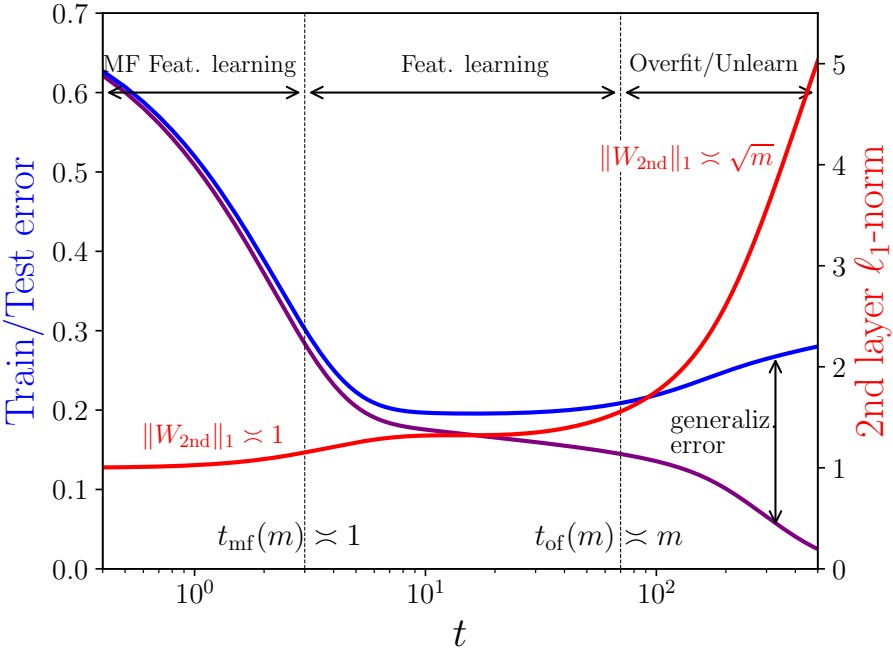

Figure 1: **Three dynamical regimes of learning in a two-layer neural networks, with $m$ hidden neurons.** Training data comprises $n$ points in $d$ dimensions distributed according to a single index model. We assume $n, m, d$ all large with $n/md = \alpha$ (here $\alpha = 0.3$). Blue: test error. Purple: train error. Red: $\ell_1$ norm of second-layer weights (a proxy for model complexity).

These observations have motivated a broad effort to encapsulate the effect of the dynamics as 'implicit regularization' [48, 3, 15, 56]: the algorithm selects an empirical risk minimizer that also minimizes a specific notion of model complexity. While this *implicit regularization hypothesis* has been fruitful, it can only be validated if we can precisely understand the training dynamics.

In this work we leverage tools from theoretical physics to directly analyze the training dynamics and derive quantitative predictions on the implicit bias of neural network training, in a simple setting. This allows us to capture feature learning and lazy/overfitting regimes within the same unified picture. We discover a time-scale separation in the training dynamics, between an early stage in which the model learns the relevant features representation of the data, and a late stage of training that is characterized by overfitting, feature 'unlearning,' and hence test error that increases with training. While the regularizing effect of early stopping has been an important object of study (for simpler models) in the past [44, 11, 61, 57], our work is the first to point out a time-scale separation between feature learning (on a faster timescale) and overfitting (on a slower time scale), thus reconciling the feature learning and neural tangent theories of learning.

We study two-layer fully connected neural networks $f(\cdot; \boldsymbol{\theta}) : \mathbb{R}^d \to \mathbb{R}$, i.e.

$$f(\boldsymbol{x}; \boldsymbol{\theta}) = \frac{1}{m} \sum_{i=1}^{m} a_i \, \sigma(\langle \boldsymbol{w}_i, \boldsymbol{x} \rangle), \tag{1.1}$$

where $\boldsymbol{\theta} = (\boldsymbol{a}, \boldsymbol{W})$, where $\boldsymbol{W} = (\boldsymbol{w}_1, \ldots, \boldsymbol{w}_m) \in \mathbb{R}^{d \times m}$ and $\boldsymbol{a} = (a_1, \ldots, a_m) \in \mathbb{R}^m$ are, respectively, first- and second-layer weights. For convenience, we fix the normalization $\|\boldsymbol{w}_i\| = 1$, and assume that $\sigma$ does not depend on $m$. We apply model (1.1) to a supervised learning task. We are given i.i.d. data $(y_i, \boldsymbol{x}_i)$, $i \leq n$, with $y_i \in \mathbb{R}$ a response variable and $\boldsymbol{x}_i \in \mathbb{R}^d$ a feature vector, and try to learn a model $f(\cdot; \boldsymbol{\theta})$ to predict the response $y_{\text{new}}$ corresponding to a new input $\boldsymbol{x}_{\text{new}}$. We use gradient flow (GF) to minimize the empirical risk under square loss, namely

$$\dot{\boldsymbol{\theta}}(t) = -\frac{n}{d} \boldsymbol{P}_{\boldsymbol{\theta}} \nabla \widehat{\mathscr{R}}_n(\boldsymbol{\theta}(t)), \qquad \widehat{\mathscr{R}}_n(\boldsymbol{\theta}) := \frac{1}{2n} \sum_{i=1}^{n} \left( y_i - f(\boldsymbol{x}_i; \boldsymbol{\theta}) \right)^2. \tag{1.2}$$

Here $\boldsymbol{P_\theta}$ is a projection matrix that guarantees that $\boldsymbol{w}_i(t) \in \mathbb{S}^{d-1}$ at all times. The factor $n/d$ is introduced for convenience and simply amounts to a rescaling of time. We will typically initialize the training by setting $(\boldsymbol{w}_i)_{i\le m} \sim_{iid} \mathsf{Unif}(\mathbb{S}^{d-1})$, and $a_i = a_0$ for all $i \le m$, and study the dependence of the training dynamics on three key parameters:

$$\text{Network width: } m, \quad \text{Overparametrization ratio: } \alpha := \frac{n}{md}, \quad \text{Initialization scale: } a_0 .$$

Alongside the train error, we will be interested in the test error at time $t$, i.e. $\mathscr{R}(\boldsymbol{\theta}(t)) := \mathbb{E}\{(y_\text{new} - f(\boldsymbol{x}_\text{new}; \boldsymbol{\theta}(t)))^2\}/2$, and the generalization error $\mathscr{R}(\boldsymbol{\theta}(t)) - \widehat{\mathscr{R}}_n(\boldsymbol{\theta}(t))$.

Model (1.1) is much simpler than state-of-the-art architectures [52], but is rich enough to investigate several general questions, which we summarize below:

When the network is sufficiently overparametrized ($\alpha$ small) and $a_0$ is large, neural tangent kernel (NTK) theory predicts that GF converges to an interpolator [30, 22, 16] .

Q1. For which region of $\alpha, a_0$ does convergence take place, beyond NTK theory?

Q2. Does the selected model provide good generalization or not [27, 37]?

In contrast, when $a_0$ is small, gradient-based algorithms can learn non-linear low-dimensional representation of the data [5, 21, 1, 6]. In these results, the difference between train and test error (generalization error) is negligible: the model does not overfit.

Q3. Can we reconcile this feature-learning/no-overfitting behavior with the lazy-training/overfitting regime described previously?

In the early phase of training, the generalization error vanishes. However, training longer times can be beneficial, despite leading to overfitting.

Q4. When does the test error start increasing with training time? When should we stop training?

Finally, scaling with the network size is crucial:

Q5. How does the generalization error depend on network size and number of iterations?

Q6. Does overfitting start earlier for larger networks or later?

In Section 2, we will present our analysis using theoretical physics techniques. Section 3 presents rigorous results confirming the picture emerging from this analysis. Finally, in Section 4 we discuss how our results address the above questions.

## 2 Main results: Dynamical mean field theory

We study the dynamics of model (1.1) under the simplest data distribution in which genuine non-linear learning is required to efficiently learn a good prediction rule, the so called *k-index model*. Namely, we assume $\boldsymbol{x}_i \sim \mathsf{N}(0, \boldsymbol{I}_d)$ and $y_i$ that depends on a low-dimensional projection $\boldsymbol{U}^\mathsf{T} \boldsymbol{x}_i$:

$$y_i = \varphi(\boldsymbol{U}^\mathsf{T} \boldsymbol{x}_i) + \varepsilon_i , \quad \varepsilon_i \sim \mathsf{N}(0, \tau^2) , \tag{2.1}$$

where the noise $\varepsilon_i$ is independent of $\boldsymbol{x}_i$, $\boldsymbol{U} \in \mathbb{R}^{d\times k}$ is an orthogonal matrix ($\boldsymbol{U}^\mathsf{T}\boldsymbol{U} = \boldsymbol{I}_k$) and $\varphi : \mathbb{R}^k \to \mathbb{R}$ is a nonlinear function, $\mathbb{E}\{\varphi(\boldsymbol{g})^2\} < \infty$ for $\boldsymbol{g}$ standard Gaussian.

An important aspect of this data distribution is that (for large $d$) it presents the largest possible gap between linear/kernel learning, which requires sample size to be superpolynomial in $d$ [27, 58], and nonlinear/neural network learning which only requires $n = O(d)$ (generically, for constant $k$). When the dimension $d$ becomes large, discovering the latent features $\boldsymbol{U}^\mathsf{T}\boldsymbol{x}$ is crucial for learning and requires nonlinear processing of the labels $y_i$ [5, 21, 1, 6].

Our main focus will be on the simplest case, namely $k = 1$, with $\varphi$ a generic function (in particular $\mathbb{E}\{\varphi(G)G\} \ne 0$ for $G \sim \mathsf{N}(0, 1)$, which corresponds *information exponent* equal to one according to the classification of [4].). Some of our results apply to $k$-index models for general fixed $k$ (in particular, the rigorous results of Section 3). We defer to future work a more complete analysis of the DMFT for $k \ge 2$.

We discover a separation of time scales at large $m$ (or large $n/d$), for sufficiently small initialization $a_0$: feature learning takes place on a fast time scale, followed by overfitting/reversal to kernel learning. This scenario is summarized in Figure 1, which plots numerical evaluations of our theoretical predictions at $k = 1$, $\tau > 0$ data distribution, in the limit $n, d, m \to \infty$ at overparametrization ratio $\alpha = 0.3$.

More precisely, we observe three regimes (below $\boldsymbol{W}_{\text{2nd}} := \boldsymbol{a}/m$ is the vector of second-layer weights in model (1.1)):

$(i)$ *Mean field feature learning.* $t = O(1)$. The network learns the low-dimensional features $\boldsymbol{U}^\top \boldsymbol{x}$; the train error and test error decrease while their difference (generalization error) is negligible; the second layer weights remain small $\|\boldsymbol{W}_{\text{2nd}}\|_1 = O(1)$.

$(ii)$ *Extended feature learning.* $1 \ll t \ll m$. The train error decreases slowly; the generalization error increases is small, i.e. $\mathscr{R}(\boldsymbol{\theta}(t)) - \widehat{\mathscr{R}}_n(\boldsymbol{\theta}(t)) = o(1)$; the test error can evolve non-monotonically, but remains approximately constant. Second-layer weights become large $1 \ll \|\boldsymbol{W}_{\text{2nd}}\|_1 \ll \sqrt{m}$.

$(iii)$ *Overfitting and feature unlearning.* $t \gtrsim m$. Train error and test error diverge significantly, i.e. $\mathscr{R}(\boldsymbol{\theta}(t)) - \widehat{\mathscr{R}}_n(\boldsymbol{\theta}(t))$ becomes of order one. At the end of this regime, the train error converges to 0, i.e. the neural network interpolates the noisy data. The test error instead grows, and its limit value is the one of a (data independent) kernel method: in other words, the model unlearns the low-dimensional structure. Finally, the second weights grow to $\|\boldsymbol{W}_{\text{2nd}}\|_1 \asymp \sqrt{m}$, which indeed is the scale required for interpolation.

In this section we outline our results based on 'dynamical mean field theory' (DMFT). The next section will present rigorous results that are proven independently.

## 2.1 Technique

Our DMFT analysis is based on the following two steps:

*Step 1:* We leverage techniques from theoretical physics to derive an approximate asymptotic characterization of the gradient flow dynamics (1.2) in the limit $n, d \to \infty$, with $n/d \to \overline{\alpha}$. This characterization consists of a set of integral-differential equations for the following asymptotic quantities (here p-lim denotes limit in probability, and we use the superscripts $n$ to emphasize the dependence of the right-hand side on $n, d$)

$$
\begin{aligned}
C_{ij}(t_1, t_2) &:= \operatorname*{p-lim}_{n,d\to\infty} \langle \boldsymbol{w}_i^n(t_1), \boldsymbol{w}_j^n(t_2) \rangle, \\
\boldsymbol{v}_i(t) &:= \operatorname*{p-lim}_{n,d\to\infty} \boldsymbol{U}^\top \boldsymbol{w}_i^n(t), \quad a_i(t) := \operatorname*{p-lim}_{n,d\to\infty} a_i^n(t).
\end{aligned}
\tag{2.2}
$$

A rigorous derivation of the DMFT in a setting that includes two-layer networks is given in [13].

However, the asymptotically exact DMFT characterization of [13] is rather complex to integrate numerically or to study analytically. In order to circumvent this problem, we use a DMFT that is is asymptotically exact for a well-defined Gaussian version of the original model. Namely, we observe that the empirical risk of Eq. (1.2) takes the form

$$
\widehat{\mathscr{R}}_n(\boldsymbol{\theta}) = \frac{1}{2n} \big\| \boldsymbol{F}(\boldsymbol{\theta}) \big\|^2,
\tag{2.3}
$$

where $\boldsymbol{F} : (\mathbb{S}^{d-1})^m \times \mathbb{R}^m \to \mathbb{R}^n$ is s stochastic process with i.i.d. components $F_i(\boldsymbol{\theta}) = y_i - f(\boldsymbol{x}_i; \boldsymbol{\theta})$. We replace these by Gaussian processes with matching mean and covariance, and study the DMFT for gradient flow with respect to the associated risk $\widetilde{\widehat{\mathscr{R}}}_n^g(\boldsymbol{\theta})$.

The Gaussian approximation comes with an error which we show analytically is vanishing on time scales of order one ( indeed on these time scales we correctly recover the mean field theory of [38, 14]) and we demonstrate empirically to be small on larger time scales ( see for instance example Fig. 4.) The curves in Fig. 1 were obtained by solving numerically the DMFT equations, see Appendix C for details.

*Step 2:* We study this DMFT, with special attention to the large network limit $m \to \infty$, and large sample size $\overline{\alpha} \to \infty$, with $\alpha = \overline{\alpha}/m$ fixed, for a generic single index model ($k = 1$). We obtain a separation of time scales in the dynamics, corresponding to distinct learning regimes.

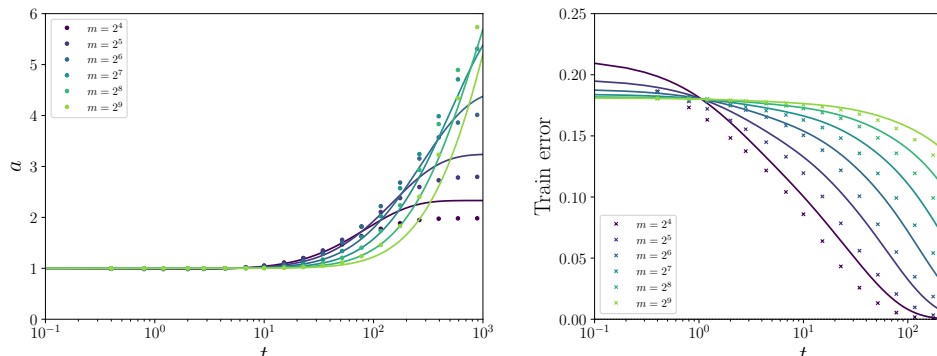

Figure 2: **Evolution of second-layer weights (left) and train error (right) when fitting pure noise data.** Here we use mean field initialization, $h(z) = (9/10)z + (1/6)z^3$, $\alpha = 0.4$ and $\tau = 0.6$. Symbols: SGD results on actual 2-layer networks with $d = 200$, $n = \alpha m d$ (averaged over 10 simulations). Continuous viridis lines: Numerical solution of the DMFT equations. Note that the second layer weights are given in terms of a scalar quantity as the result of the statistically symmetric initialization.

The analysis of the DMFT equations in the double limit $m, t \to \infty$ is an example of singular perturbation theory [9, 29]. Making this type of analysis rigorous is notoriously challenging and we proceed by a combination of numerical solutions and analytical derivations.

In the following, we will first consider the simplest possible setting, pure noise data, and subsequently consider the single-index model. The structure of the activation function and target nonlinearity will be encoded in the functions

$$h(q) := \mathbb{E}\{\sigma(G_1)\sigma(G_q)\}, \quad \widehat{\varphi}(q) := \mathbb{E}\{\varphi(G_1)\sigma(G_q)\},$$

where $G_1, G_q$ are standard jointly Gaussian with $\mathbb{E}\{G_1 G_q\} = q$. The relation between $\sigma, \varphi$ and $h, \widehat{\varphi}$ is conveniently expressed in terms of the expansions in Hermite polynomials $\sigma(x) = \sum_{k \geq 0} s_k \mathrm{He}_k(x)$, $\varphi(x) = \sum_{k \geq 0} f_k \mathrm{He}_k(x)$, which corresponds to the analytic expansion $h(q) = \sum_{k \geq 0} s_k^2 q^k$, $\widehat{\varphi}(q) = \sum_{k \geq 0} s_k f_k q^k$.

As mentioned above, we assume throughout $n, d \to \infty$, with $n/d \to \overline{\alpha} \in (0, \infty)$, with the limit $m, \overline{\alpha} \to \infty$ taken afterwards. To further simplify our analysis, we assume a symmetric initialization whereby $a_i(0) = a_0$ is independent of $i \leq m$ and $(\boldsymbol{w}_i(0) : i \leq m) \sim_{iid} \mathsf{Unif}(\mathbb{S}^{d-1})$. Throughout, we use 'with high probability' for 'with probability converging to one as $n, d \to \infty$.'

In Section 3 we present rigorous results that do not require either of these simplifying assumptions.

## 2.2 Training on pure noise

We begin by the case in which the data is pure noise: $y_i = \varepsilon_i \sim \mathsf{N}(0, \tau^2)$. A by-now-classic experiment [60] showed that deep learning models have sufficient capacity to achieve vanishing training error even when actual labels are replaced by random ones: they 'interpolate pure noise.'

The ability of a model $\mathcal{F}_\Theta = (f(\,\cdot\,; \boldsymbol{\theta}) : \boldsymbol{\theta} \in \Theta)$ to interpolate pure noise is intimately connected to its Gaussian complexity $\mathcal{G}(\mathcal{F}_\Theta; n) := \mathbb{E} \sup_{\boldsymbol{\theta} \in \Theta} \langle \boldsymbol{g}, f(\boldsymbol{X}; \boldsymbol{\theta}) \rangle / n$ [53] (where $\boldsymbol{g} \sim \mathsf{N}(\boldsymbol{0}, \boldsymbol{I}_n)$ is independent of $f(\boldsymbol{X}, ; \boldsymbol{\theta}) = (f(\boldsymbol{x}_i; \boldsymbol{\theta}) : i \leq n)$. Indeed, interpolation is impossible unless $\mathcal{G}(\mathcal{F}_\Theta; n) \geq \tau$. Viceversa, $\mathcal{G}(\mathcal{F}_\Theta; n) \ll \tau$ ensures good generalization.

By a theorem of [7] for the network (1.1), $\mathcal{G}(\mathcal{F}_\Theta; n) \leq L_\sigma \|\boldsymbol{a}/m\|_1 \sqrt{d/n}$ (with $L_\sigma$ depending uniquely on $\sigma$). This means that, in order to interpolate noise, the average magnitude of second layer weights must be $\|\boldsymbol{a}/m\|_1 \geq L_\sigma^{-1} \tau \sqrt{n/d} = (L_\sigma^{-1} \alpha^{1/2}) \tau \sqrt{m}$.

However, complexity bounds do not have implications on the convergence of GF to an interpolator.

Figure 2 compares the DMFT predictions to simulations using SGD to train an actual two layer networks. In this figure we initialize $a(0) = 1$, and let $a(t)$ evolve with GF alongside the first layer weigths. We observe that the theory describes well the empirical results, despite the Gaussian

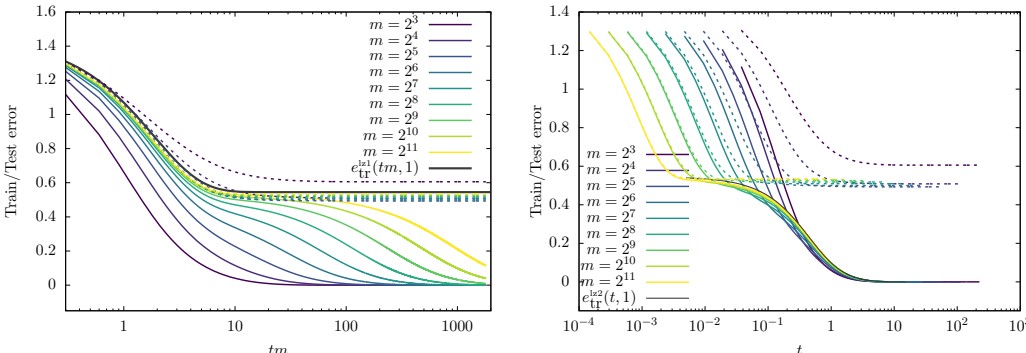

Figure 3: **Train/test error (right) when fitting data from a single index model**. We set $h(z) = \widehat{\varphi}(z) = (9/10)z + z^2/2$, $\tau = 0.3$ and $\alpha = 0.3$. Lines correspond to predictions from the DMFT (continuous: train error; dashed: test error). Black continuous line is the $m \to \infty$ value. Right: Same data plotted versus $t$.

approximation in our DMFT and the difference between SGD and GF. We also observe that second-layer weights remain roughly constant until a large time $t_\#(m)$, which appears to increase with $m$. Roughly at the same time, train error starts to decrease and converges to zero.

In Section G.1 of the appendix, we will make precise the above picture of the evolution of $a(t)$. Here, we consider a simplified setting in which $a(t) = \gamma \sqrt{m}$ with $\gamma$ independent of $m$, not evolving with training. Note that $\mathcal{G}(\mathcal{F}_\Theta; n) \asymp \gamma/\sqrt{\alpha}$ and hence such a network can interpolate pure noise if $\gamma$ is larger than threshold depending on $\alpha$. Our DMFT predicts a sharp phase transition. For $\alpha \in (0, 1)$, GF converges to vanishing train error with high probability if $\gamma > \gamma_{\text{GF}}(\alpha, m)\tau$, and converges to a strictly positive training error if $\gamma < \gamma_{\text{GF}}(\alpha, m)\tau$. The threshold $\gamma_{\text{GF}}(\alpha, m)$ converges to a limit $\gamma_{\text{GF}}^*(\alpha) \in (0, 1)$ as $m \to \infty$.

A rephrasing of the same phenomenon states that $\lim_{n,d\to\infty} \widehat{\mathscr{R}}_n^g(\boldsymbol{\theta}(t)) = e_{\text{tr}}(t; m, \gamma)$, and

$$\lim_{t\to\infty} \lim_{m\to\infty} e_{\text{tr}}(t; m, \gamma_0) = \begin{cases} e_*(\gamma) > 0 & \text{for } \gamma < \gamma_{\text{GF}}^*(\alpha)\tau, \\ 0 & \text{for } \gamma \geq \gamma_{\text{GF}}^*(\alpha)\tau. \end{cases} \quad (2.4)$$

Informally $\gamma_{\text{GF}}^*(\alpha)$ is the minimum complexity $\gamma$ for a very large network to interpolate noise via gradient flow. The functions $\gamma_{\text{GF}}^*(\alpha)$, $e_*(\gamma)$ will play an important role below.

We will next consider training on data from a single-index model. The initial scale of second-layer weights $\|\boldsymbol{a}(0)/m\|_1$ plays a crucial role and we will separately analyze lazy and mean field initializations.

## 2.3 Training on data with latent structure: lazy initialization

We initialize $a(0) = \gamma_0 \sqrt{m}$, and let $a(t)$ evolve according to GF alongside first-layer weights. DMFT predicts the emergence of three dynamical regimes for large $m$ and large $\overline{\alpha}$ (with $n/d \to \overline{\alpha}$). For an illustration, we refer to Fig. 3.

*First dynamical regime:* $t = O(1/m)$. Second layer weights do not change significantly $\gamma(t) = \gamma_0 + o_m(1)$, while first layer-weights move by $\|\boldsymbol{w}_i(t) - \boldsymbol{w}_i(0)\| = \Theta(1/\sqrt{m})$. Because the weights $a_i(t)$ are of order $\sqrt{m}$, even an $O(1/\sqrt{m})$ change in the $\boldsymbol{w}_i$ leads to a significant decrease in test error and train error.

Train and test error are close to each other. Namely, the following limits are well defined

$$\lim_{n,d\to\infty} \widehat{\mathscr{R}}_n^g(\boldsymbol{\theta}(t)) = e_{\text{tr}}(t; \varphi, \gamma_0, m, \alpha), \qquad \lim_{n,d\to\infty} \mathscr{R}^g(\boldsymbol{\theta}(t)) = e_{\text{ts}}(t; \varphi, \gamma_0, m, \alpha). \quad (2.5)$$

with $\lim_{m\to\infty} e_{\text{tr}}(\hat{t}/m; \varphi, \gamma_0, m, \alpha) = \lim_{m\to\infty} e_{\text{ts}}(\hat{t}/m; \varphi, \gamma_0, m, \alpha) =: e^{\text{lz1}}(\hat{t}; \varphi, \gamma_0, \alpha)$.

For large scaled time $\hat{t}$, the error $e^{\text{lz1}}(\hat{t}; \varphi, \gamma_0, \alpha)$ converges to the error of the best linear approximation to $f_*$. This dynamical regime follows the qualitative predictions of NTK theory, and is essentially linear in the weights $\boldsymbol{w}_i$, but the time is too short for the model to overfit the data.

*Second dynamical regime:* $t = \Theta(1)$. Second layer weights do not change significantly: $\gamma(t) = \gamma_0 + o_m(1)$, while first layer weights change significantly $\|\boldsymbol{w}_i(t) - \boldsymbol{w}_i(0)\| = \Theta(1)$. However they change orthogonally to the latent subspace $\boldsymbol{U}$ and hence the test error does not change: no actual learning takes place in this regime, but the model starts to overfit the data.

More formally, train and test error have well defined limits as the network width diverges:

$$e_{\text{tr}}^{\text{lz2}}(t; \varphi, \gamma_0, \alpha) := \lim_{m \to \infty} e_{\text{tr}}(t; \varphi, \gamma_0, m, \alpha), \quad e_{\text{ts}}^{\text{lz2}}(t; \varphi, \gamma_0, \alpha) := \lim_{m \to \infty} e_{\text{ts}}(t; \varphi, \gamma_0, m, \alpha). \quad (2.6)$$

However, the scaling function $e_{\text{ts}}^{\text{lz2}}(t; \varphi, \gamma_0, \alpha)$ for the test error is constant in time and equal to the value achieved at the end of the first dynamical regime. Namely

$$e_{\text{ts}}^{\text{lz2}}(t; \varphi, \gamma_0, \alpha) = \lim_{\hat{t} \to \infty} e^{\text{lz1}}(\hat{t}; \varphi, \gamma_0, \alpha) = \frac{1}{2} \left( \tau^2 + \|\varphi\|^2 - \frac{\|\nabla \widehat{\varphi}(\boldsymbol{0})\|^2}{h'(0)} + \gamma_0^2(h(1) - h'(0)) \right). \quad (2.7)$$

Since the $\boldsymbol{w}_i$'s move orthogonally to the latent space, their dynamics is equivalent (for large $m$) to the one in the pure noise setting, modulo a redefinition of $h$. The right plot in Fig. 3 illustrates this.

*Third dynamical regime:* $t = \Theta(m)$. The qualitative properties of this regime depend whether or not $\gamma_0$ is larger than an interpolation threshold $\gamma_{\text{GF}}^*(\alpha, \varphi, \tau)$, which generalizes the threshold $\gamma_{\text{GF}}^*(\alpha) = \gamma_{\text{GF}}^*(\alpha, 0, 1)$ introduced in the pure noise case. Because the dynamics of weights $\boldsymbol{w}_i$ in the subspace orthogonal to $\boldsymbol{U}$ is equivalent to dynamics in pure noise, we expect the interpolation threshold $\gamma_{\text{GF}}^*(\alpha, \varphi, \tau)$ to be given in terms of pure noise threshold $\gamma_{\text{GF}}^*(\alpha)$ as follows:

$$\gamma_{\text{GF}}^*(\alpha, \varphi, \tau) = \left( \tau^2 + \|\varphi\|^2 - \frac{\|\nabla \hat{\varphi}(\boldsymbol{0})\|^2}{h'(0)} \right)^{1/2} \gamma_{\text{GF}}^*(\alpha). \quad (2.8)$$

For $\gamma_0 > \gamma_{\text{GF}}^*(\alpha, \varphi, \tau)$, interpolation is achieved during the second dynamical regime, no further evolution takes place.

For $\gamma_0 < \gamma_{\text{GF}}^*(\alpha, \varphi, \tau)$, a non-trivial evolution takes place for $t = \Theta(m)$. Introducing the rescaled time $z \in (0, \infty)$, we obtain, as $m \to \infty$,

$$\gamma(mz) = \gamma^{\text{lz3}}(z) + o_m(1), \quad e_{\text{tr}}(mz) = e_{\text{tr}}^{\text{lz3}}(z) + o_m(1), \quad e_{\text{ts}}(mz) = e_{\text{ts}}^{\text{lz3}}(z) + o_m(1). \quad (2.9)$$

Further, for large values of the rescaled time $z \to \infty$, $\gamma^{\text{lz3}}(z)$ grows to $\overline{\gamma}_{\text{GF}}^*(\alpha, \varphi, \tau) \approx \gamma_{\text{GF}}^*(\alpha, \varphi, \tau)$, while $e_{\text{tr}}^{\text{lz3}}(z)$ decreases to 0. In other words, interpolation is achieved on this third regime.

Further the test error $e_{\text{ts}}^{\text{lz3}}(z)$ increases from $e_{\text{ts}}^{\text{lz2}}(t; \varphi, \gamma_0, \alpha)$ to $e_{\text{ts}}^{\text{lz2}}(t; \varphi, \gamma_{\text{GF}}^*, \alpha)$, with $\gamma_{\text{GF}}^* = \gamma_{\text{GF}}^*(\alpha, \varphi, \tau)$ whereby $e_{\text{ts}}^{\text{lz2}}(\cdots)$ is given by Eq. (2.7).

### 2.4 Training on data with latent structure: mean field initialization

We initialize $a(0) = a_0$, independent of $m$ and let second layer weights evolve. Note that at initialization the network's Rademacher complexity is small, namely of order $a_0 \sqrt{d/n} = a_0/\sqrt{\alpha m}$. Our DMFT analyisis predicts two dynamical regimes for large $m$. We will refer to them as 'first' and 'third regime' for consistency with other settings ( see Sec.G.2 of the appendix). For an illustration, we refer to Figs. 4 and 5.

*First dynamical regime:* $t = O(1)$. Both first and second layer weights change by order one: $a(t) = a_0 + \Theta(1)$ and $\|\boldsymbol{w}_i(t) - \boldsymbol{w}_i(0)\| = \Theta(1)$. and as a consequence test and train error decrease significantly. In this regime, the two errors remain close to each other and their evolution is well captured by the mean field theory of [38, 14], as specialized to the case of spherically invariant distributions [10, 2].

Namely, $\lim_{m \to \infty} a(t) = a^{\text{mf1}}(t)$, $\lim_{m \to \infty} \boldsymbol{v}(t) = \boldsymbol{v}^{\text{mf1}}(t)$, and DMFT reduces to a system of $k + 1$ ordinary differential equations for the $k + 1$ scalar variables $(a^{\text{mf1}}(t), \boldsymbol{v}^{\text{mf1}}(t))$

$$\begin{aligned} \partial_t \boldsymbol{v}^{\text{mf1}}(t) &= \alpha a^{\text{mf1}}(t) \boldsymbol{Q}_{\boldsymbol{v}^{\text{mf1}}(t)} \left( \nabla \hat{\varphi}(\boldsymbol{v}^{\text{mf1}}(t)) - a^{\text{mf1}}(t) h'(\|\boldsymbol{v}^{\text{mf1}}(t)\|^2) \boldsymbol{v}^{\text{mf1}}(t) \right), \\ \partial_t a^{\text{mf1}}(t) &= \alpha \hat{\varphi}(\boldsymbol{v}^{\text{mf1}}(t)) - \alpha a^{\text{mf1}}(t) h(\|\boldsymbol{v}^{\text{mf1}}(t)\|^2), \end{aligned} \quad (2.10)$$

where $\boldsymbol{Q}_{\boldsymbol{v}} := \boldsymbol{I}_k - \boldsymbol{v}\boldsymbol{v}^{\mathsf{T}}$. As mentioned above, train and test error coincide in the large width limit

$$\lim_{m \to \infty} e_{\text{tr}}(t) = \lim_{m \to \infty} e_{\text{ts}}(t) = e^{\text{mf1}}(t).$$

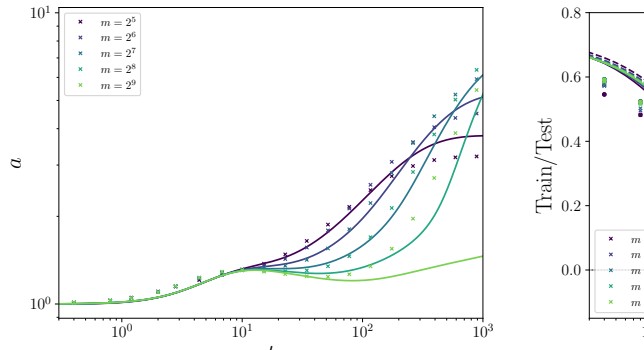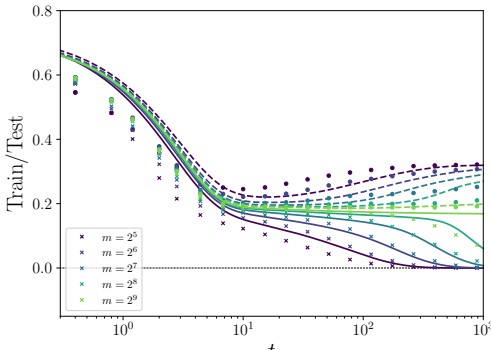

Figure 4: **Training dynamics under a single-index model.** We set $h(q) = \widehat{\varphi}(q) = (9/10)q + q^3/6$, $\tau = 0.3$ and $\alpha = 0.3$, under mean field initialization. Left: second-layer weights. Right: train and test error. Symbols are empirical results for SGD with actual two-layer neural networks with $d = 200$, $n = \alpha m d$ (averaged over 10 simulations). Lines correspond to predictions from the DMFT (on the right, continuous: train error; dashed: test error).

An explicit formula for $e^{\text{mf1}}(t)$ is given in Appendix G.2.1. In the case $k = 1$ and $\widehat{\varphi}(z) = h(z)$, we have that $a^{\text{mf1}} = 1$, $v^{\text{mf1}} = 1$ is a fixed point of Eq. (2.10), and indeed the only fixed point with $v^{\text{mf1}} > 0$. If $h'(0) > 0$, then, we have $(a^{\text{mf1}}(t), v^{\text{mf1}}(t)) \to (1, 1)$ as $t \to \infty$, and therefore test and train error converge to the Bayes error $e^{\text{mf1}}(t) \to \tau^2/2$. This is significantly smaller than the test error achieved with lazy initialization. The separation between lazy and mean-field initialization is expected because feature learning takes place in the mean field regime.

*Third dynamical regime:* $t = \Omega(m)$. Computing the local stability of DMFT solutions around the mean field asymptotics (see Appendix G.2.2) suggests that the latter breaks down for $t = \Theta(m)$. For $t \gtrsim m$, we observe that the second layer weights grow to achieve $a(t) \asymp \sqrt{m}$, the projection onto the latent space decreases to $v(t) \asymp 1/\sqrt{m}$, and train and test error diverge, eventually achieving $e_{\text{tr}}(t) \approx 0$ and test error significantly larger than the Bayes error achieved earlier. We refer to this phenomenon as 'feature unlearning.'

Denoting by $t_0(m; c)$ the time at which $a(t) = c\sqrt{m}$ (for $c$ a small constant), we expect the existence of a window size $w(m)$ such that

$$\lim_{m \to \infty} \frac{a\big(t_0(m; c) + z\,w(m)\big)}{\sqrt{m}} = \gamma^{\text{mf3}}(z), \qquad \lim_{m \to \infty} e_{\text{tr/ts}}\big(t_0(m; c) + z\,w(m)\big) = e^{\text{mf3}}_{\text{tr/ts}}(z),$$
(2.11)

where $\gamma^{\text{mf3}}(z)$, $e^{\text{mf3}}_{\text{tr}}(z)$, $e^{\text{mf3}}_{\text{ts}}(z)$ are scaling functions describing the dynamics on this timescale. We expect $t_0(m; c) = t_*(c)m + o(m)$, and $w(m) \lesssim t_0(m; c)$, but our numerical solutions are not sufficient to determine the precise scaling. On the other hand, it appears that at large times, the complexity converges close the interpolation threshold:

$$\lim_{z \to \infty} \gamma^{\text{mf3}}(z) = \overline{\gamma}^*_{\text{GF}}(\alpha, \varphi, \tau) \approx \gamma^*_{\text{GF}}(\alpha, \varphi, \tau).$$
(2.12)

Finally, the evolution of train and test error for $a(t) \asymp \sqrt{m}$ appears to match the behavior at fixed second-layer weights. Namely, we define two functions

$$\varepsilon^{\text{mf}}_{\text{tr/ts}}(\gamma) := \lim_{m \to \infty} e_{\text{tr/ts}}(t_0(m; \gamma), m).$$
(2.13)

We observe that the limit curves $(\gamma, \varepsilon^{\text{mf}}_{\text{tr}}(\gamma))$, $(\gamma, \varepsilon^{\text{mf}}_{\text{ts}}(\gamma))$, match closely asymptotic train and test error obtained by fixing $a(t) = \gamma\sqrt{m}$, and not letting second-layer weight evolve. This confirms the hypothesis that $\gamma(t)$ is a slow variable, while others converge as if $\gamma$ was fixed.

## 3 Lower bounding the overfitting timescale

In this section we rigorously establish two results that confirm elements of the scenario outlined in the previous sections. We emphasize that the result presented here are non-asymptotic, i.e. hold at finite

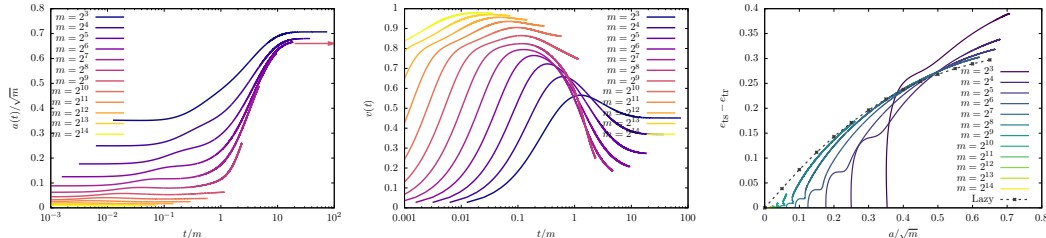

Figure 5: **Left: second layer weights on the scale $\sqrt{m}$ as a function of $t/m$.** Curves appear to collapse on a master curve. The red arrow denotes $\gamma_{GF}^*$ and the curves appear to converge to that limit. **Center: the projection of the first layer weights on the latent space** in the single index model as a function of time on timescales of order $m$. **Right: difference between test and train error** as a function of the second layer weights on the scale $\sqrt{m}$. The finite $m$ curve are approaching a scaling curve which coincides with the one obtained by evaluating the same quantity but with a lazy initialization and fixed second layer weights.

$n, m, d$ modulo unspecified absolute constants. Further, we do not assume a symmetric initialization of the weights. Throughout this section setting, it is more convenient to rescale time defining $\hat{t} = t\alpha$. Hence. instead of the flow (1.2), we study

$$\dot{\boldsymbol{\theta}}(\hat{t}) = -m\boldsymbol{P}_{\boldsymbol{\theta}}\nabla\widehat{\mathscr{R}}_n(\boldsymbol{\theta}(\hat{t})).\tag{3.1}$$

For $\alpha = \Theta(1)$ the parametrizations $t$ and $\hat{t}$ are equivalent.

The first result of this section implies that (under mean field initialization) overfitting cannot take place on times of order one.

**Theorem 1.** *Under the GF dynamics* (1.2), *and the data distribution in the introduction (with $k$ arbitrary), further assume $\|\sigma\|_{\mathrm{Lip}}, \|\sigma\|_{\infty} \leq L, |\varphi(0)|, \|\varphi\|_{\mathrm{Lip}} \leq L, \|\boldsymbol{a}(0)\|_{\infty} \leq a_0$, for some $a_0 \geq 1$ and that the $\boldsymbol{w}_i(0), i \leq m$ are independent of the data $\{(y_i, \boldsymbol{x}_i) : i \leq n\}$. Finally assume $n \geq d \vee m$. Then, there exist universal constants $C_0, C_1$, and the following holds for all $\hat{t} \geq 0$,*

$$\|\boldsymbol{a}(\hat{t})\|_{\infty} \leq a_0 + a_1\hat{t}, \quad a_1 := C_0L(\tau + \sqrt{k} + a_0L),\tag{3.2}$$

$$\left|\mathscr{R}(\boldsymbol{a}(\hat{t}), \boldsymbol{W}(\hat{t})) - \widehat{\mathscr{R}}_n(\boldsymbol{a}(\hat{t}), \boldsymbol{W}(\hat{t}))\right| \leq C_1(L^2(a_0 + a_1\hat{t})^2 + \tau^2) \cdot \sqrt{\frac{d}{n}}.\tag{3.3}$$

Under mean field initialization, $a_0$ is a fixed constant and hence $a_1$ is also bounded, whence the generalization error in Eq. (3.3) is small as long as $\hat{t} = o((n/d)^{1/4})$ (equivalently, for $\alpha$ fixed, $\hat{t} = o(m^{1/4})$).

By itself, this result implies a separation of timescales between learning and overfitting, thus confirming the picture developed within DMFT, but falls short of characterizing the overfitting timescale.

The second result implies that, up to time-scale of order one, the dynamics is closely tracked by the mean field equations (2.10). Since the $a_i(0)$ at initialization are not necessarily all equal, these are generalized as

$$\partial_{\hat{t}}\boldsymbol{v}_i^{\mathrm{mfl}}(\hat{t}) = a_i^{\mathrm{mfl}}(\hat{t})\boldsymbol{Q}_{\boldsymbol{v}_i^{\mathrm{mfl}}(\hat{t})}\left(\nabla\hat{\varphi}(\boldsymbol{v}_i^{\mathrm{mfl}}(\hat{t})) - \frac{1}{m}\sum_{j=1}^m a_j^{\mathrm{mfl}}(\hat{t})h'(\langle\boldsymbol{v}_i^{\mathrm{mfl}}(\hat{t}), \boldsymbol{v}_j^{\mathrm{mfl}}(\hat{t})\rangle)\boldsymbol{v}_j^{\mathrm{mfl}}(\hat{t})\right),$$

$$\partial_{\hat{t}}a_i^{\mathrm{mfl}}(\hat{t}) = \hat{\varphi}(\boldsymbol{v}_i^{\mathrm{mfl}}(\hat{t})) - \frac{1}{m}\sum_{j=1}^m a_j^{\mathrm{mfl}}(\hat{t})h(\langle\boldsymbol{v}_i^{\mathrm{mfl}}(\hat{t}), \boldsymbol{v}_j^{\mathrm{mfl}}(\hat{t})\rangle).\tag{3.4}$$

The mean field prediction for test error is the same as for training error and given by

$$e_{\mathsf{ts}}(\hat{t}) = \frac{1}{2}\|\varphi\|_{L^2}^2 - \frac{1}{m}\sum_{j=1}^m a_j^{\mathrm{mfl}}(\hat{t})\hat{\varphi}(\boldsymbol{v}_j^{\mathrm{mfl}}(\hat{t})) + \frac{1}{2m^2}\sum_{j=1}^m a_i^{\mathrm{mfl}}(\hat{t})a_j^{\mathrm{mfl}}(\hat{t})\, h(\langle\boldsymbol{v}_i^{\mathrm{mfl}}(\hat{t}), \boldsymbol{v}_j^{\mathrm{mfl}}(\hat{t})\rangle)$$

**Theorem 2.** *Under the the GF dynamics* (1.2), *and the data distribution in the introduction (with $k$ arbitrary), further assume that $\|\varphi\|_{\infty}, \|\varphi'\|_{\infty}, \|\varphi'\|_{\mathrm{Lip}} \leq L, \|\sigma\|_{\infty}, \|\sigma'\|_{\infty}, \|\sigma'\|_{\mathrm{Lip}} \leq L$. Further*

*assume $|a_i(0)| \leq L$ for all $i \leq m$, $(\boldsymbol{w}_i(0))_{i \leq m} \sim_{iid} \mathsf{Unif}(\mathbb{S}^{d-1})$. Then for any $\delta > 0$ there exist constants $c_0$ $c_1$, $C$ depending on $L, \tau, \delta, k$ such that, letting $T_{\mathsf{lb}} = c_0(\log m)^{1/3} \wedge (\log n/d)^{1/3}$, the following happens with probability at least $1 - 2\exp(-c_1 d)$,*

$$\sup_{\hat{t} \leq T_{\mathsf{lb}}} \frac{1}{m} \sum_{i=1}^{m} \left( |a_i(\hat{t}) - a_i^{\mathsf{mfl}}(\hat{t})| + \|\boldsymbol{v}_i(\hat{t}) - \boldsymbol{v}_i^{\mathsf{mfl}}(\hat{t})\| \right) \leq C \left( \frac{1}{m} \vee \frac{1}{d} \vee \frac{d}{n} \right)^{1/2-\hat{\delta}}, \tag{3.5}$$

$$\sup_{t \leq T_{\mathsf{lb}}} \left| \mathscr{R}(\boldsymbol{a}(\hat{t}), \boldsymbol{W}(\hat{t})) - e_{\mathsf{ts}}(\hat{t}) \right| \leq C \left( \frac{1}{m} \vee \frac{1}{d} \vee \frac{d}{n} \right)^{1/2-\delta}. \tag{3.6}$$

**Remark 3.1.** While the analysis in the previous section requires $m \to \infty$ *after* $n, d \to \infty$, neither Theorem 3.1 nor Theorem 3.2 make the assumption. In particular, Eq. (3.3) implies that the generalization error is small for $\hat{t} = o((n/d)^{1/4})$ *irrespective of* $m$.

Similarly, Eqs. (3.5), (3.6) imply that the mean field theory of [38, 14, 45] captures well the evolution of the system for times $t = o((\log m)^{1/3} \wedge (\log n/d)^{1/3})$.

## 4 Discussion

We conclude by highlighting a few qualitative conclusions of our work, and how they address questions raised in Section 1. In the following remarks, we consider $\alpha = n/md$ as constant.

**Interpolation mechanism.** In the current setting, the neural model complexity is proportional to $\|\boldsymbol{a}(t)\|_1/\sqrt{m} = \gamma(t) + o_n(1)$. We observe two alternative scenarios. If the complexity at initialization is large enough $\gamma_0 > \gamma_{\mathsf{GF}}^*(\alpha)\tau$, then the gradient flow rapidly converges to a near interpolator without significant change in $\gamma(t)$. If instead, $\gamma_0 < \gamma_{\mathsf{GF}}^*(\alpha)\tau$, then $\gamma(t)$ grows to reach the interpolation threshold at which point the training error converges to $0$.

**Adiabatic evolution of model complexity.** In the latter case, the complexity $\gamma(t)$ evolves on a slower time scale than other degrees of freedom. The dynamics on shorter timescales is well approximated by the one at fixed $\gamma$ (given by the current value $\gamma(t)$). The generalization error becomes of order one only when $\gamma(t)$ is of order one.

**Decoupling of learning and overfitting.** When $\gamma_0 = o_m(1)$, the fact that $\gamma(t)$ acts as a slow variable implies a large-$m$ decoupling between learning (which takes place on faster timescales, as long as $\gamma(t) = o_m(1)$), and overfitting (which takes place on slower timescales, when $\gamma(t) = \Omega_m(1)$). This has several implications for the questions outlined in the introduction.

Q3: Lazy initialization $a(0) \asymp \sqrt{m}$ leads to poor generalization because the feature-learning phase is skipped either partially or altogether.

Q2: Training until interpolation is generally suboptimal.

Q4: The optimal tradeoff is obtained at the end of the first phase.

Q5, Q6: Further, at fixed overerparametrization $n/md = \alpha$, overfitting starts later for larger models.

**Overfitting and feature unlearning.** The above description points at a non-monotonicity of the model quality, which improves on short time scales, and deteriorates at larger time scales. Reciprocally, early stopping acts as a regularization. While this phenomenon is well understood for linear models [24, 57], our analysis provides an analogous (quantitative) scenario for training neural network models. In particular, it clarifies the underlying mechanism: in the same dynamical regime in which network complexity grows ($\gamma(t)$ becomes of order one), and training error becomes negligible, the low-dimensional latent features are 'unlearned' ($\boldsymbol{v}(t)$ becomes of order $1/\sqrt{m}$). We expect that these findings also allow to understand the beneficial effect of regularization on the second layer.

### Acknowledgments

This work was supported by the NSF through award DMS-2031883, the Simons Foundation through Award 814639 for the Collaboration on the Theoretical Foundations of Deep Learning, and the ONR grant N00014-18-1-2729. This work was supported by the French government under the France 2030 program (PhOM - Graduate School of Physics) with reference ANR-11-IDEX-0003.

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

# A Setting

We recall for reference some basic definitions and notations. We consider the 2-layer network defined by

$$f(\boldsymbol{x}; \boldsymbol{a}, \boldsymbol{W}) = \frac{1}{m} \sum_{i=1}^{m} a_i \, \sigma(\langle \boldsymbol{w}_i, \boldsymbol{x} \rangle). \tag{A.1}$$

Throughout, we assume an offset to be subtracted so that $\mathbb{E}\sigma(G) = 0$, for $G \sim \mathsf{N}(0,1)$. The network input $\boldsymbol{x}$ is a $d$-dimensional real vector and the output is a scalar variable. The parameters of the network are the weights of the first layer collected in the matrix $\boldsymbol{W}$ defined as

$$\boldsymbol{W} = \begin{pmatrix} \boldsymbol{w}_1 \\ \boldsymbol{w}_2 \\ \cdot \\ \cdot \\ \cdot \\ \boldsymbol{w}_m \end{pmatrix} \in \mathbb{R}^{m \times d}, \quad \boldsymbol{w}_i \in \mathbb{R}^d. \tag{A.2}$$

We will assume that $\|\boldsymbol{w}_i\|^2 = 1$. The weights of the second layer are instead $(a_1, \ldots, a_m)$ and are real, possibly unbounded, variables.

We consider a dataset of $n$ points independent and identically distributed $(y_i, \boldsymbol{x}_i)_{i \leq n}$ where $\boldsymbol{x}_i \sim \mathsf{N}(0, \boldsymbol{I}_d)$, and the labels $y_i$ are generated according to the following $k$-index models:

$$y_i = \varphi(\boldsymbol{U}^\mathsf{T} \boldsymbol{x}_i) + \varepsilon_i. \tag{A.3}$$

Therefore, labels depend on the projection of the covariates on a fixed subspace $\boldsymbol{U} \in \mathbb{R}^{d \times k}$, with $\boldsymbol{U}^\mathsf{T} \boldsymbol{U} = \boldsymbol{I}_k$ (there is no loss of generality in assuming $\boldsymbol{U}$ orthogonal). Efficient learning requires to estimate this subspace. Since we consider learning with square loss, we assume

$$\|\varphi\|_2^2 := \mathbb{E}\{\varphi(\boldsymbol{U}^\mathsf{T} \boldsymbol{x}_i)^2\} = \mathbb{E}\{\varphi(\boldsymbol{g})^2\},$$

where $\boldsymbol{g} \sim \mathsf{N}(0, \boldsymbol{I}_k)$. We refer to the case $\varphi = 0$ as the 'pure noise case' or 'pure noise data'.

We now discuss the covariance structure of the network given by Eq. (A.1). For two sets of weights $(\boldsymbol{a}_1, \boldsymbol{W}_1)$ and $(\boldsymbol{a}_2, \boldsymbol{W}_2)$ we have

$$\mathbb{E}\{f(\boldsymbol{x}; \boldsymbol{a}_1, \boldsymbol{W}_1) \, f(\boldsymbol{x}; \boldsymbol{a}_2, \boldsymbol{W}_2)\} = \frac{1}{m^2} \sum_{i,j=1}^{m} a_{1,i} a_{2,j} h\left(\langle \boldsymbol{w}_{1,i}, \boldsymbol{w}_{2,j} \rangle\right). \tag{A.4}$$

The average in the rhs of Eq. (A.4) is over the data distribution while the function $h(q)$ is defined as

$$h(q) = \mathbb{E}\{\sigma(G_1)\sigma(G_2)\} \tag{A.5}$$

for $(G_1, G_2)$ centered jointly Gaussian with $\mathbb{E}\{G_i^2\} = 1$, $\mathbb{E}\{G_1 G_2\} = q$.

Furthermore we have that:

$$\mathbb{E}\{f(\boldsymbol{x}; \boldsymbol{a}, \boldsymbol{W}) \, \varphi(\boldsymbol{U}^\mathsf{T} \boldsymbol{x})\} = \frac{1}{m} \sum_{i=1}^{m} a_i \widehat{\varphi}(\boldsymbol{U}^\mathsf{T} \boldsymbol{w}_i). \tag{A.6}$$

where $\widehat{\varphi}$ is given by

$$\widehat{\varphi}(\boldsymbol{v}) := \mathbb{E}\left\{\sigma(\langle \boldsymbol{v}, \boldsymbol{G} \rangle + \sqrt{1 - \|\boldsymbol{v}\|^2} G_0)\varphi(\boldsymbol{G})\right\}, \tag{A.7}$$

for $\boldsymbol{G} \sim \mathsf{N}(0, \boldsymbol{I}_k)$ independent of $G_0 \sim \mathsf{N}(0,1)$.

We consider Gaussian process $f^g(\boldsymbol{a}, \boldsymbol{W})$, $\varphi^g$ with the same covariance function defined above and define the empirical risk under Gaussian approximation as

$$\widehat{\mathscr{R}}_n^g(\boldsymbol{a}, \boldsymbol{W}) = \frac{1}{2n} \sum_{i=1}^{n} \left(f_i^g(\boldsymbol{a}, \boldsymbol{W}) - \varphi_i^g - \varepsilon_i\right)^2 \tag{A.8}$$

$$= \frac{1}{2n} \|\boldsymbol{f}^g(\boldsymbol{a}, \boldsymbol{W}) - \boldsymbol{\varphi}^g - \boldsymbol{\varepsilon}\|^2,$$

where $\boldsymbol{f}^g(\cdots) = (f_i^g(\cdots) : i \leq n)$, $\boldsymbol{\varphi}^g = (\varphi_i^g : i \leq n)$, $\boldsymbol{\varepsilon} = (\varepsilon_i : i \leq n)$ are vectors containing $n$ i.i.d. copies of the above processes. We will also write $\boldsymbol{y}^g = \boldsymbol{\varphi}^g + \boldsymbol{\varepsilon}$.

Given a model with estimated parameters $\hat{\boldsymbol{a}}, \widehat{\boldsymbol{W}}$, the test error is given by

$$\mathscr{R}(\hat{\boldsymbol{a}}, \widehat{\boldsymbol{W}}) = \frac{1}{2}\mathbb{E}\big\{\big(f^g(\hat{\boldsymbol{a}}, \widehat{\boldsymbol{W}}) - \varphi^g - \varepsilon\big)^2\big\} \tag{A.9}$$

$$= \frac{1}{2}\mathbb{E}\big\{\big(f(\boldsymbol{x}, \hat{\boldsymbol{a}}, \widehat{\boldsymbol{W}}) - \varphi(\boldsymbol{U}^\mathsf{T}\boldsymbol{x}) - \varepsilon\big)^2\big\}\,,$$

where the expectation in the first line is over a triple $(f^g, \varphi^g, \varepsilon)$ independent of the data, and in the second line with respect to $\boldsymbol{x}$. The two expectations coincide because they depend uniquely on the second moments of these processes.

We are interested in studying the gradient flow dynamics in the random landscape $\widehat{\mathscr{R}}_n(\boldsymbol{a}, \boldsymbol{W})$

$$\dot{\boldsymbol{a}}(t) = -\frac{n}{d}\nabla_{\boldsymbol{a}}\widehat{\mathscr{R}}_n(\boldsymbol{a}(t), \boldsymbol{W}(t))\,,$$
$$\dot{\boldsymbol{w}}_i(t) = -\frac{n}{d}\nabla_{\boldsymbol{w}_i}\widehat{\mathscr{R}}_n(\boldsymbol{a}(t), \boldsymbol{W}(t)) - \nu_i(t)\boldsymbol{w}_i(t) \quad \forall i = 1, \dots, m\,. \tag{A.10}$$

The Lagrange multipliers $\nu_i$ are added to enforce the spherical constraint $\|\boldsymbol{w}_i(t)\|^2 = 1$. While we consider the case of normalized first-layer weights, our approach can be generalized to unconstrained weights or to include weight decay (ridge regularization). As explained in the main text, we will replace this by gradient flow in the Gaussian model $\widehat{\mathscr{R}}_n^g(\boldsymbol{a}, \boldsymbol{W})$. We refer to Section K for a discussion of DMFT in the original non-Gaussian model.

In our analysis we will always consider the proportional asymptotics

$$n, d \to \infty, \quad \frac{n}{d} \to \overline{\alpha} \in (0, \infty)\,. \tag{A.11}$$

We typically index sequences and limits by $n$, but it is understood that $d = d(n) \to \infty$ as well. After $n, d \to \infty$ proportionally, we will consider the large network asymptotics $m \to \infty$ at fixed $\alpha = \overline{\alpha}/m$.

In the following we will drop the superscript $g$ and write, for instance $\widehat{\mathscr{R}}_n(\boldsymbol{a}, \boldsymbol{W})$ instead of $\widehat{\mathscr{R}}_n^g(\boldsymbol{a}, \boldsymbol{W})$ whenever clear from the context. All of our analytical predictions (except for Section 3) are obtained within the Gaussian model.

## B  Technique

Notice that each fitting error $F_i(\boldsymbol{\theta}) = y_i - f(\boldsymbol{x}_i; \boldsymbol{\theta})$, $i \in \{1, \dots, n\}$ is a random function of the model parameters $\boldsymbol{\theta}$. The randomness is due to the randomness in $\boldsymbol{x}_i$ and in the noise $\varepsilon_i$. The empirical risk in Eq. (1.2) can be rewritten as

$$\widehat{\mathscr{R}}_n(\boldsymbol{\theta}) = \frac{1}{2n}\|\boldsymbol{F}(\boldsymbol{\theta})\|^2\,, \quad \boldsymbol{F}(\boldsymbol{\theta}) = \big(F_1(\boldsymbol{\theta}), \dots, F_n(\boldsymbol{\theta})\big)\,. \tag{B.1}$$

Our key approximation consists in replacing the i.i.d. random functions $(F_i)_{i \leq n}$ by i.i.d. Gaussian processes $(F_i^g)_{i \leq n}$ with matching mean and covariance. While DMFT equations have been recently proven without recurring to this approximation (see [13] and appendices), their structure is simpler in the Gaussian case, which allows us to carry out the large-$m$ analysis.

Computing the covariance of $\boldsymbol{F}(\cdot)$ is a straightforward exercise. We assume for simplicity that an intercept is subtracted so that $\mathbb{E}[\sigma(G)] = 0$, $\mathbb{E}[\varphi(\boldsymbol{G})] = 0$ and otherwise these functions are generic ($G, G_1, \boldsymbol{G}$ and so on will denote standard Gaussian vectors). We then have

$$\mathbb{E}\big\{f(\boldsymbol{x}; \boldsymbol{\theta}_1)f(\boldsymbol{x}; \boldsymbol{\theta}_2)\big\} = \frac{1}{m^2}\langle\boldsymbol{a}_1, h(\boldsymbol{W}_1^\mathsf{T}\boldsymbol{W}_2)\boldsymbol{a}_2\rangle\,, \tag{B.2}$$

$$\mathbb{E}\big\{f(\boldsymbol{x}; \boldsymbol{\theta})y\big\} = \frac{1}{m^2}\langle\boldsymbol{a}, \widehat{\varphi}(\boldsymbol{W}^\mathsf{T}\boldsymbol{U})\rangle\,. \tag{B.3}$$

Recall that $\boldsymbol{\theta} = (\boldsymbol{a}, \boldsymbol{W})$ where $\boldsymbol{a} \in \mathbb{R}^m$, $\boldsymbol{W} = (\boldsymbol{w}_1, \dots, \boldsymbol{w}_m) \in \mathbb{R}^{d \times m}$ are the first layer weights Finally, $h : \mathbb{R} \to \mathbb{R}$, $\widehat{\varphi} : \mathbb{R}^k \to \mathbb{R}$ encode the activations $\sigma$ and the target function $\varphi$, with $h$ applied entrywise to the matrix $\boldsymbol{W}_1^\mathsf{T}\boldsymbol{W}_2$.

The covariance of $F_i(\boldsymbol{\theta}) = y_i - f_i(\boldsymbol{x}; \boldsymbol{\theta})$ is easily computed from the above, and this defines completely the corresponding Gaussian process $(F_i^g)_{i \leq n}$. We denote the associated risk function $\widehat{\mathscr{R}}_n^g(\boldsymbol{\theta}) := \|\boldsymbol{F^g}(\boldsymbol{\theta})\|^2/2n$.

Let us emphasize that the cost function $\widehat{\mathscr{R}}_n^g(\boldsymbol{\theta})$ remains highly non-trivial despite the fact that the functions $F_i$ are replaced by Gaussian processes. Near-minima of high-dimensional Gaussian processes have a very rich structure, which is a central theme in spin glass theory [39, 50]. Additional layers of complexity arise here for two reasons. First, $\widehat{\mathscr{R}}_n^g(\boldsymbol{\theta})$ is a sum of *squares of Gaussians* and, second, the underlying Gaussian process has a significantly more intricate covariance than in standard spin glasses (where typically depends only on the inner product $\langle \boldsymbol{\theta}_1, \boldsymbol{\theta}_2 \rangle$). Recent work explored the simpler case in which $F_i^g(\,\cdot\,)$ is a Gaussian process with covariance $\mathbb{E}\{F_i^g(\boldsymbol{\theta}_1)F_i^g(\boldsymbol{\theta}_2)\} = \xi(\langle \boldsymbol{\theta}_1, \boldsymbol{\theta}_2 \rangle)$ depending uniquely on the inner product [25, 26, 51, 49, 42, 43, 33]. Gradient descent dynamics on these models has been recently studied via DMFT in [31, 32]: our work builds on these advances. DMFT was leveraged before to address other questions in high-dimensional statistics and ML [35, 12]. We refer to [8, 13] for mathematical results on the DMFT approach.

While $\widehat{\mathscr{R}}_n^g(\boldsymbol{\theta})$ has a non-trivial structure, methods from statistical physics can be brought to bear to derive an asymptotic characterization. Namely, define the functions

$$C_{ij}^n(t_1, t_2) = \langle \boldsymbol{w}_i(t_1), \boldsymbol{w}_j(t_2) \rangle, \quad \boldsymbol{v}_i^n(t) := \boldsymbol{U}^\mathsf{T}\boldsymbol{w}_i(t), \quad a_i^n(t). \tag{B.4}$$

These functions are random (because of the random initialization and the randomness in $\boldsymbol{F^g}$) and depend on $n, d$. However, as $n, d \to \infty$ with $n/d \to \overline{\alpha}$, they converge to non-random limits $(C_{ij}(t_1, t_2))_{i < j \leq m}, (\boldsymbol{v}_i(t))_{i \leq m}, (a_i(t))_{i < j \leq m}$ that are the unique solution of a set of coupled integro-differential equations, see the appendices. We refer to these as to the DMFT equations.

Our main focus is on the behavior of the solutions of these equations for large $m$ and, at first sight, the complexity of the DMFT increases with $m$. An important simplification arises when choosing a symmetric initial condition $a_i(0) = a_0$ for all $i \leq m$, and $(\boldsymbol{w}_i(0))_{i \leq m} \sim_{iid} \mathsf{Unif}(\mathbb{S}^{d-1})$. Namely, the solution of the DMFT equations is symmetric under permutations of the neurons: $C_{ii}(t_1, t_2) = C_d(t_1, t_2)$ for $i \leq m$ and $C_{ij}(t_1, t_2) = C_o(t_1, t_2)$ for $i \neq j \leq m$, while $\boldsymbol{v}_i(t) = \boldsymbol{v}(t)$, $a_i(t) = a(t)$ for $i \leq m$. We then have a reduction to a set of integro-differential equations on $k + 3$ functions, that depend parametrically on $m$.

We use two approaches to study these equations (see appendix):

    (a) Numerical integration for increasing values of $m$ under different initial conditions.

    (b) Asymptotics as $m \to \infty$ (at fixed $\alpha = \overline{\alpha}/m$) via singular perturbation theory [9, 29].

For $(b)$, a specific dynamical regime is identified by a scaling of the time variable, which in our case will take the form $t = t_\#(m) \cdot \hat{t}$ for a certain fixed function $t_\#(m)$ and $\hat{t} = O(1)$ a scaled time. The asymptotics of DMFT quantities in that regime takes the form

$$\lim_{m \to \infty} \boldsymbol{v}\Big(t_\#(m) \cdot \hat{t}; m, \alpha = \frac{\overline{\alpha}}{m}\Big) = \boldsymbol{v}_*(\hat{t}; \alpha). \tag{B.5}$$

## C   Dynamical Mean Field Theory (DMFT)

In this section we state the results of Dynamical Mean Field Theory (DMFT). We will outline a heuristic derivation in Section L. We first introduce the general DMFT equations in Section C.1 and the corresponding predictions for certain observable of interest in Section C.2. These are a set of $\Theta(m^2)$ integro-differential equations in as many unknown functions.

We then specialize these equations to the case of a symmetric initialization, in which $\boldsymbol{w}_i(0) \sim \mathsf{Unif}(\mathbb{S}^{d-1})$ and $a_i(0) = a_0$ for all $i \leq m$, see Section C.3 In this case, the dynamics is characterized by a set of $k + 3$ equations which are stated in Sections C.4 and C.5.

### C.1   General DMFT equations

Let $a_i^n(t), \boldsymbol{w}_i^n(t), \nu_i^n(t)$ the the solution of Eq (A.10) when the dynamics is initialized at non-random $a_i^n(0) = a_{0,i}$, $i \leq n$ and possibly random, $\boldsymbol{w}_i^n(0)$ such that $\langle \boldsymbol{w}_i^n(0), \boldsymbol{w}_j^n(0) \rangle \to C_{ij}^0$ for $i, j \leq n$, $\boldsymbol{U}^\mathsf{T}\boldsymbol{w}_i^n(0) \to \boldsymbol{v}_i^0$ for $i \leq n$. While random, the $\boldsymbol{w}_i^n(0)$ are assumed here to be independent of the random processes $\boldsymbol{f}^g, \boldsymbol{\varphi}^g, \boldsymbol{\varepsilon}$.

For $t, s \geq 0$ consider the quantities

$$C_{ij}^n(t, s) := \langle \boldsymbol{w}_i^n(t), \boldsymbol{w}_j^n(s) \rangle, \quad \boldsymbol{v}_i^n(t) := \boldsymbol{U}^\mathsf{T} \boldsymbol{w}_i^n(t). \tag{C.1}$$

Then DMFT predicts that these quantities have a well defined non-random limit as $n, d \to \infty$,

$$C_{ij}(t, s) = \lim_{n,d \to \infty} C_{ij}^n(t, s), \quad \boldsymbol{v}_i(t) = \lim_{n,d \to \infty} \boldsymbol{v}_i^n(t), \quad a_i(t) = \lim_{n,d \to \infty} a_i^n(t), \tag{C.2}$$

where the limits are understood to hold in almost sure sense. These limits are the unique solution of a set of integro-differential equations in the unknowns $\{C_{ij}(t, s), R_{ij}(t, s), \boldsymbol{v}_i(t), a_i(t) : i, j \leq m\}$, which we next state as three sets: (1) Dynamical equations; (2) Equations for auxiliary functions; (3) Boundary conditions. Before that, we mention some constraints that need to be satisfied by the solution of these equations.

**(0) Constraints.** The functions $C_{ij}(t, s)$, $R_{ij}(t, s)$ satisfy:

$$C_{ii}(t, t) = 1 \quad \forall 0 \leq t, \tag{C.3}$$
$$C_{ij}(t, s) = C_{ji}(s, t) \quad \forall 0 \leq t, s, \tag{C.4}$$
$$R_{ij}(t, s) = 0 \quad \forall 0 \leq t < s. \tag{C.5}$$

The first condition in particular implies the following useful relation:

$$\frac{\mathrm{d}C_{ij}(t, t)}{\mathrm{d}t} = \lim_{t' \to t} \left[ \frac{\partial C_{ij}(t, t')}{\partial t} + \frac{\partial C_{ji}(t, t')}{\partial t} \right]. \tag{C.6}$$

We refer to the property (C.5) (and similar ones for $R$ functions appearing below) as 'causality constraint.'

**(1) Dynamical equations.** These equations determine the dynamics of $\{C_{ij}(t, s), R_{ij}(t, s), \boldsymbol{v}_i(t), a_i(t) : i, j \leq m\}$, and involve the auxiliary functions (memory kernels) $M_{ij}^C(t, s)$, $M_{ij}^R(t, s)$ and (Lagrange multipliers) $\nu_i(t)$ (the last equations assume implicitly $t_a > t_b$):

$$\frac{\mathrm{d}a_i(t)}{\mathrm{d}t} = -\frac{\overline{\alpha}}{m} \int_0^t R_A(t, s) \left[ \frac{1}{m} \sum_{l=1}^m a_l(s) h\left(C_{li}(s, t)\right) - \hat{\varphi}(\boldsymbol{v}_i(t)) \right] \mathrm{d}s \tag{C.7}$$

$$-\frac{\overline{\alpha}}{m} \int_0^t C_A(t, s) \frac{1}{m} \sum_{l=1}^m a_l(s) h'\left(C_{li}(s, t)\right) R_{il}(t, s) \, \mathrm{d}s,$$

$$\frac{\mathrm{d}\boldsymbol{v}_i(t)}{\mathrm{d}t} = -\nu_i(t)\boldsymbol{v}_i(t) + \frac{\overline{\alpha}}{m} a_i(t) \nabla\hat{\varphi}(\boldsymbol{v}_i(t)) \int_0^t R_A(t, s) \, \mathrm{d}s - \frac{1}{m} \sum_{j=1}^m \int_0^t M_{ij}^R(t, s) \, \boldsymbol{v}_j(s) \, \mathrm{d}s, \tag{C.8}$$

$$\frac{\partial C_{ij}(t_a, t_b)}{\partial t_a} = -\nu_i(t_a)C_{ij}(t_a, t_b) + \frac{\overline{\alpha}}{m} a_i(t_a) \langle \nabla\hat{\varphi}(\boldsymbol{v}_i(t_a)), \boldsymbol{v}_j(t_b) \rangle \int_0^{t_a} R_A(t_a, s) \, \mathrm{d}s \tag{C.9}$$

$$-\frac{1}{m} \sum_{l=1}^m \int_0^{t_a} M_{il}^R(t_a, s) \, C_{lj}(s, t_b) \, \mathrm{d}s - \frac{1}{m} \sum_{l=1}^m \int_0^{t_b} M_{il}^C(t_a, s) \, R_{jl}(t_b, s) \mathrm{d}s,$$

$$\frac{\partial R_{ij}(t_a, t_b)}{\partial t_a} = -\nu_i(t_a)R_{ij}(t_a, t_b) + \delta_{ij}\delta(t_a - t_b) - \frac{1}{m} \sum_{l=1}^m \int_{t_b}^{t_a} M_{il}^R(t_a, s) \, R_{lj}(s, t_b) \, \mathrm{d}s. \tag{C.10}$$

We point out that the $\delta(t_a - t_b)$ in the last equation (together with Eq. (C.5)) has to be interpreted as follows: $R_{ij}(t, t') = 0$ for $t < t'$ while, for $\varepsilon > 0$, $R_{ij}(t + \varepsilon, t) = \delta_{ij} + o_\varepsilon(1)$.

Equations (C.9) and (C.10) can also be written in terms of an effective stochastic process in $\mathbb{R}^m$: $\boldsymbol{w}^e(t) = (w_i^e(t) : i \leq m)$. This is defined as the solution of the following set of ODEs (for

$i \in \{1, \ldots, m\}$):

$$\frac{\mathrm{d}w_i^e(t)}{\mathrm{d}t} = -\nu_i(t)w_i^e(t) + \alpha a_i(t)\langle \nabla\widehat{\varphi}(\boldsymbol{v}(t)), \boldsymbol{v}(t')\rangle \int_0^t R_A(t,s)\,\mathrm{d}s \tag{C.11}$$

$$- \frac{1}{m}\sum_{l=1}^m \int_0^t M_{il}^R(t,s)w_l^e(s)\,\mathrm{d}s + \eta_i(t) + b_i(t)\,, \tag{C.12}$$

$$w_i^e(0) \sim \mathsf{N}(0,1)\,, \tag{C.13}$$

where $(\eta_i(t):\ i \le m)$ is a centered Gaussian process with covariance

$$\mathbb{E}[\eta_i(t)\eta_j(t')] = -\frac{1}{m}M_{ij}^C(t,t')\,. \tag{C.14}$$

Define $\underline{b}(t) = (b_i(t): i \le m)$. The solution of Eqs. (C.9) and (C.10) can be written as

$$C_{ij}(t,t') = \lim_{\underline{b}\to 0}\mathbb{E}\left[w_i(t)w_j(t')\right]\,, \tag{C.15}$$

$$R_{ij}(t,t') = \lim_{\underline{b}\to 0}\frac{\delta\mathbb{E}[w_i(t)]}{\delta b_j(t')}\,. \tag{C.16}$$

In fact the stochastic process of Eq. (C.11) is expected to describe the limit distribution of the second-layer weights $\boldsymbol{W}(t)$. Namely, for $i \le d$, define $\tilde{\boldsymbol{w}}_i(t) = \boldsymbol{W}(t)\boldsymbol{e}_i \in \mathbb{R}^m$ be a vector containing the $i$-th coordinate of each neuron. Then, for any fixed $i$ and any $T$,

$$(\tilde{\boldsymbol{w}}_i(t) : 0 \le t \le T) \overset{d}{\Rightarrow} (\boldsymbol{w}^e(t) : 0 \le t \le T)\,. \tag{C.17}$$

Here $\overset{d}{\Rightarrow}$ denotes convergence in distribution as $n, d \to \infty$, in $C([0,T],\mathbb{R}^m)$.

**(2) Equations for auxiliary functions.** The memory kernels $M^R$ and $M^C$ are defined by

$$\begin{aligned}
M_{ij}^R(t,s) &= \frac{\overline{\alpha}}{m}\left[R_A(t,s)h'(C_{ij}(t,s)) + C_A(t,s)h''(C_{ij}(t,s))R_{ij}(t,s)\right]a_i(t)a_j(s)\,,\\
M_{ij}^C(t,s) &= \frac{\overline{\alpha}}{m}C_A(t,s)h'(C_{ij}(t,s))a_i(t)a_j(s)\,.
\end{aligned} \tag{C.18}$$

where the functions $R_A$ and $C_A$ satisfy the symmetry properties $C_A(t,s) = C_A(s,t)$ and $R_A(t,s) = 0$ for $t < s$, and are the unique solution

$$\begin{aligned}
&\int_{t'}^t \left[\delta(t-s) + \Sigma_R(t,s)\right]R_A(s,t')\,\mathrm{d}s = \delta(t-t')\,,\\
&\int_0^t \left[\delta(t-s) + \Sigma_R(t,s)\right]C_A(s,t')\,\mathrm{d}s + \int_0^{t'}\Sigma_C(t,s)R_A(t',s)\,\mathrm{d}s = 0\,,
\end{aligned} \tag{C.19}$$

where

$$\begin{aligned}
\Sigma_C(t,s) &:= \tau^2 + \|\varphi\|^2 + \frac{1}{m^2}\sum_{i,j=1}^m a_i(t)a_j(s)h\big(C_{ij}(t,s)\big)\\
&\quad - \frac{1}{m}\sum_{l=1}^m a_l(t)\hat{\varphi}(\boldsymbol{v}_l(t)) - \frac{1}{m}\sum_{l=1}^m a_l(s)\hat{\varphi}(\boldsymbol{v}_l(s))\,,
\end{aligned} \tag{C.20}$$

$$\Sigma_R(t,s) := \frac{1}{m^2}\sum_{i,j=1}^m a_i(t)a_j(s)h'\big(C_{ij}(t,s)\big)R_{ij}(t,s)\,.$$

The Lagrange multipliers $\nu_i(t)$ have to be fixed to enforce the constraint $C_{ii}(t,t) = 1$ which follows from $\boldsymbol{w}_\alpha \in \mathbb{S}^{d-1}$. The corresponding equations are

$$\nu_i(t_a) = \frac{\overline{\alpha}}{km}a_i(t_a)\langle \boldsymbol{v}_i(t_a), \nabla\hat{\varphi}(\boldsymbol{v}_i(t_a))\rangle \int_0^{t_a} R_A(t_a,s)\,\mathrm{d}s \tag{C.21}$$

$$- \frac{1}{m}\sum_{j=1}^m \int_0^{t_a} M_{ij}^R(t_a,s)\,C_{ij}(s,t_a)\,\mathrm{d}s - \frac{1}{m}\sum_{j=1}^m \int_0^{t_a} M_{ij}^C(t_a,s)\,R_{ji}(t_a,s)\,\mathrm{d}s\,.$$

**(3) Boundary conditions.** The dynamical equations (C.7) to (C.10) can be integrated from a set of initial conditions that reflect initial conditions of the GF dynamics:

$$
\begin{aligned}
\boldsymbol{v}_i(0) = \boldsymbol{v}_i^0, \quad & a_i(0) = a_i^0 \qquad \forall i \in \{1, \ldots, m\}, \\
C_{ij}(0,0) = C_{ij}^0 \qquad & \forall i, j \in \{1, \ldots, m\}, \\
R_{ij}(0,0) = 0 \qquad & \forall i, j \in \{1, \ldots, m\}.
\end{aligned}
\tag{C.22}
$$

## C.2 Expressions for train and test error

The asymptotics of many quantities of interest can be expressed in terms of the solutions of the DMFT equations stated in the last section. In particular, the train error $\widehat{\mathscr{R}}_n(\boldsymbol{W}(t), \boldsymbol{a}(t))$ and test error $\mathscr{R}(\boldsymbol{W}(t), \boldsymbol{a}(t))$ at time $t$ have well defined limits under the proportional asymptotics:

$$
\lim_{n \to \infty} \widehat{\mathscr{R}}_n^g(\boldsymbol{W}(t), \boldsymbol{a}(t)) = e_{\text{tr}}(t), \qquad \lim_{n \to \infty} \mathscr{R}^g(\boldsymbol{W}(t), \boldsymbol{a}(t)) = e_{\text{ts}}(t).
\tag{C.23}
$$

The functions $e_{\text{tr}}(t)$ $e_{\text{ts}}(t)$ are given by

$$
e_{\text{tr}}(t) = -\frac{1}{2} C_A(t,t),
\tag{C.24}
$$

$$
e_{\text{ts}}(t) = \frac{1}{2} \left\{ \tau^2 + \frac{1}{k} \|\varphi\|^2 + \frac{1}{m^2} \sum_{i,j=1}^m h\big(C_{ij}(t,t)\big) - \frac{2}{m} \sum_{i=1}^m \hat{\varphi}(\boldsymbol{v}_i(t)) \right\}
\tag{C.25}
$$

More generally, $C_A(t,s)$ gives the asymptotics of the correlation of residuals:

$$
\lim_{n \to \infty} \frac{1}{n} \big\langle \boldsymbol{\Delta}(t), \boldsymbol{\Delta}(s) \big\rangle = -C_A(t,s),
\tag{C.26}
$$

$$
\boldsymbol{\Delta}(t) := \boldsymbol{y}^g - \boldsymbol{f}^g(\boldsymbol{a}(t), \boldsymbol{W}(t)).
\tag{C.27}
$$

where we recall that $\boldsymbol{y}^g = \boldsymbol{\varphi}^g + \boldsymbol{\varepsilon}$.

## C.3 Symmetric initialization and solutions

As anticipated, we consider the uninformative initialization $\boldsymbol{w}_i^n(0) \sim \text{Unif}(\mathbb{S}^{d-1})$ and $a_i^n(0) = a_0$ for all $i \le m$. This results in the following initialization for the DMFT equations of

$$
\begin{aligned}
\boldsymbol{v}_i(0) = \boldsymbol{v}_i^0 = \boldsymbol{0} \qquad & \forall i \in \{1, \ldots, m\}, \\
C_{i \ne j}(0,0) = C_{i \ne j}^0 = 0 \qquad & \forall i \ne j, i, j \in \{1, \ldots, m\}, \\
C_{ii}(0,0) = C_{ii}^0 = 1 \qquad & \forall i \in \{1, \ldots, m\}.
\end{aligned}
\tag{C.28}
$$

This initialization is invariant under permutations of the $m$ neurons. Since the DMFT equations of Section C.1 are equivariant under such permutations, their solution is also invariant under permutations. This means that it takes the form:

$$
C_{ij}(t,t') = \begin{cases} C_d(t,t') & \text{if } i = j, \\ C_o(t,t') & \text{if } i \ne j, \end{cases} \qquad R_{ij}(t,t') = \begin{cases} R_d(t,t') & \text{if } i = j, \\ R_o(t,t') & \text{if } i \ne j, \end{cases}
\tag{C.29}
$$

$$
\boldsymbol{v}_i(t) = \boldsymbol{v}(t), \qquad \nu_i(t) = \nu(t), \qquad a_i(t) = a(t) \quad \forall i.
\tag{C.30}
$$

As a consequence, the memory kernels in Eq. (C.18) take the form

$$
M_{ij}^C(t,t') = \begin{cases} M_d^C(t,t') & \text{if } i = j, \\ M_o^C(t,t') & \text{if } i \ne j, \end{cases} \qquad M_{ij}^R(t,t') = \begin{cases} M_d^R(t,t') & \text{if } i = j, \\ M_o^R(t,t') & \text{if } i \ne j. \end{cases}
\tag{C.31}
$$

We will refer to the reduced DMFT under symmetry as to the SymmDMFT.

## C.4 DMFT equations for symmetric initialization (SymmDMFT)

**(1) Dynamical equations.** Substituting the ansats of the previous section in the equations of Section C.1, we obtain the following equations for the functions $a(t)$, $\boldsymbol{v}(t)$, $C_d(t,s)$, $C_o(t,s)$, $R_d(t,s)$, $R_o(t,s)$:

$$\frac{\mathrm{d}a}{\mathrm{d}t}(t) = \frac{\overline{\alpha}}{m}\hat{\varphi}(\boldsymbol{v}(t)) \int_0^t R_A(t,s)\,\mathrm{d}s \tag{C.32}$$

$$- \frac{\overline{\alpha}}{m} \int_0^t R_A(t,s)a(s)\left[\frac{1}{m}h(C_d(t,s)) + \frac{m-1}{m}h(C_o(t,s))\right]\mathrm{d}s$$

$$- \frac{\overline{\alpha}}{m} \int_0^t C_A(t,s)a(s)\left[\frac{1}{m}h'(C_d(t,s))R_d(t,s) + \frac{m-1}{m}h'(C_o(t,s))R_o(t,s)\right]\mathrm{d}s\,,$$

$$\frac{\mathrm{d}\boldsymbol{v}}{\mathrm{d}t}(t) = -\nu(t)\boldsymbol{v}(t) + \frac{\overline{\alpha}}{m}\nabla\hat{\varphi}(\boldsymbol{v}(t))a(t)\int_0^t R_A(t,s)\,\mathrm{d}s \tag{C.33}$$

$$- \frac{1}{m}\int_0^t\left[M_R^{(d)}(t,s) + (m-1)M_R^{(o)}(t,s)\right]\boldsymbol{v}(s)\,\mathrm{d}s\,,$$

$$\partial_t C_d(t,t') = -\nu(t)C_d(t,t') + \frac{\overline{\alpha}}{m}\langle\nabla\hat{\varphi}'(\boldsymbol{v}(t)),\boldsymbol{v}(t')\rangle a(t)\int_0^t R_A(t,s)\,\mathrm{d}s \tag{C.34}$$

$$- \frac{1}{m}\int_0^t\left[M_R^{(d)}(t,s)C_d(t',s) + (m-1)M_R^{(o)}(t,s)C_o(t',s)\right]\mathrm{d}s$$

$$- \frac{1}{m}\int_0^{t'}\left[M_C^{(d)}(t,s)R_d(t',s) + (m-1)M_C^{(o)}(t,s)R_o(t',s)\right]\mathrm{d}s\,,$$

$$\partial_t C_o(t,t') = -\nu(t)C_o(t,t') + \frac{\overline{\alpha}}{m}\langle\nabla\hat{\varphi}(\boldsymbol{v}(t)),\boldsymbol{v}(t')\rangle a(t)\int_0^t R_A(t,s)\,\mathrm{d}s \tag{C.35}$$

$$- \frac{1}{m}\int_0^t\left[M_R^{(d)}(t,s)C_o(t',s) + M_R^{(o)}(t,s)C_d(t',s) + (m-2)M_R^{(o)}(t,s)C_o(t',s)\right]\mathrm{d}s$$

$$- \frac{1}{m}\int_0^{t'}\left[M_C^{(d)}(t,s)R_o(t',s) + M_C^{(o)}(t,s)R_d(t',s) + (m-2)M_C^{(o)}(t,s)R_o(t',s)\right]\mathrm{d}s\,,$$

$$\partial_t R_d(t,t') = -\nu(t)R_d(t,t') + \delta(t-t') \tag{C.36}$$

$$- \frac{1}{m}\int_{t'}^t\left[M_R^{(d)}(t,s)R_d(s,t') + (m-1)M_R^{(o)}(t,s)R_o(s,t')\right]\mathrm{d}s\,,$$

$$\partial_t R_o(t,t') = -\nu(t)R_o(t,t') - \frac{1}{m}\int_{t'}^t\left[M_R^{(d)}(t,s)R_o(s,t') + M_R^{(o)}(t,s)R_d(s,t')\right. \tag{C.37}$$

$$\left. + (m-2)M_R^{(o)}(t,s)R_o(s,t')\right]\mathrm{d}s\,.$$

**(2) Equations for auxiliary functions.** The memory kernels $M_R^{(s)}(t,s)$, $M_R^{(o)}(t,s)$ and $M_C^{(s)}(t,s)$, $M_C^{(o)}(t,s)$ are given by:

$$M_R^{(d)}(t,s) = \frac{\overline{\alpha}}{m}a(t)a(s)\left[R_A(t,s)h'(C_d(t,s)) + C_A(t,s)h''(C_d(t,s))R_d(t,s)\right]\,, \tag{C.38}$$

$$M_R^{(o)}(t,s) = \frac{\overline{\alpha}}{m}a(t)a(s)\left[R_A(t,s)h'(C_o(t,s)) + C_A(t,s)h''(C_o(t,s))R_o(t,s)\right]\,, \tag{C.39}$$

$$M_C^{(d)}(t,s) = \frac{\overline{\alpha}}{m}a(t)a(s)C_A(t,s)h'(C_d(t,s))\,, \tag{C.40}$$

$$M_C^{(o)}(t,s) = \frac{\overline{\alpha}}{m}a(t)a(s)C_A(t,s)h'(C_o(t,s))\,. \tag{C.41}$$

Further, $C_A(t,s)$, $R_A(t,s)$ are given by the same equations (C.19), where $\Sigma_C$, $\Sigma_R$ are simplified as follows:

$$\Sigma_C(t,s) = \tau^2 + \|\varphi\|^2 - a(t)\hat{\varphi}(\boldsymbol{v}(t)) - a(s)\hat{\varphi}(\boldsymbol{v}(s)) + \frac{a(t)a(s)}{m}h(C_d(t,s))$$

$$+ \frac{m-1}{m}a(t)a(s)h(C_o(t,s)) \tag{C.42}$$

$$\Sigma_R(t,s) = \frac{a(t)a(s)}{m}h'(C_d(t,s))R_d(t,s) + \frac{m-1}{m}a(t)a(s)h'(C_o(t,s))R_o(t,s)$$

Finally, the Lagrange multipliers are determined by

$$\nu(t) = \frac{\overline{\alpha}}{m}\langle\nabla\hat{\varphi}(\boldsymbol{v}(t)), \boldsymbol{v}(t)\rangle a(t)\int_0^t R_A(t,s)\,\mathrm{d}s$$

$$- \frac{1}{m}\int_0^t \left[M_R^{(s)}(t,s)C_d(t,s) + (m-1)M_R^{(o)}(t,s)C_o(t,s)\right]\mathrm{d}s \tag{C.43}$$

$$- \frac{1}{m}\int_0^t \left[M_C^{(s)}(t,s)R_d(t,s) + (m-1)M_C^{(o)}(t,s)R_o(t,s)\right]\mathrm{d}s\,.$$

**(3) Boundary conditions.** As anticipated the SymmDMFT is initialized as

$$\boldsymbol{v}(0) = \boldsymbol{0}, \qquad C_d(0,0) = 1 \qquad C_o(0,0) = 0\,. \tag{C.44}$$

### C.5 Expressions for train and test error under symmetric initialization

The general expression for train and test error given in Section C.2 specialize to:

$$e_{\mathrm{tr}}(t) = -\frac{1}{2}C_A(t,t)\,, \tag{C.45}$$

$$e_{\mathrm{ts}}(t) = \frac{1}{2}\left[\tau^2 + \|\varphi\|^2 - 2a(t)\hat{\varphi}(\boldsymbol{v}(t)) + \frac{1}{m}a^2(t)h(1) + \frac{m-1}{m}a^2(t)h(C_o(t,t))\right]\,. \tag{C.46}$$

## D  Numerical integration of the DMFT equations

### D.1  Integration technique

We integrate the SymmDMFT equations (C.32) to (C.37) using a standard Euler discretization. Namely, we discretize time on an equi-spaced grid $t \in \mathbb{T} := \{0, \eta, 2\eta, \dots\}$ and approximate derivatives by differences and integrals by sums on this grid. As an example, Eq. (C.32) is replaced by

$$\frac{a(t+\eta) - a(t)}{\eta} = \frac{\overline{\alpha}}{m}\hat{\varphi}(\boldsymbol{v}(t))\sum_{s\in\mathbb{T}, s\leq t}R_A(t,s)\,\eta \tag{D.1}$$

$$- \frac{\overline{\alpha}}{m}\sum_{s\in\mathbb{T}, s\leq t}R_A(t,s)a(s)\left[\frac{1}{m}h(C_d(t,s)) + \frac{m-1}{m}h(C_o(t,s))\right]\eta$$

$$- \frac{\overline{\alpha}}{m}\sum_{s\in\mathbb{T}, s\leq t}C_A(t,s)a(s)\left[\frac{1}{m}h'(C_d(t,s))R_d(t,s) + \frac{m-1}{m}h'(C_o(t,s))R_o(t,s)\right]\eta\,.$$

The discretization of Eq. (C.10) deserves an additional clarification because of the delta-function. For $t_a \geq t_b$, $t_a, t_b \in \mathbb{N}\eta$, we compute

$$\frac{R_{ij}(t_a+\eta, t_b) - R_{ij}(t_a, t_b)}{\eta} = -\nu_i(t_a)R_{ij}(t_a, t_b) - \frac{1}{\eta}\delta_{ij}\mathbf{1}_{t_a=t_b}$$

$$- \frac{1}{m}\sum_{l=1}^m \sum_{s\in[t_a,t_b]\cap\mathbb{N}\eta} M_{il}^R(t_a, s)\,R_{lj}(s, t_b)\,\eta\,,$$

with boundary condition

$$R_{ij}(t_b, t_b) = 0 \quad \forall i, j \leq m \,.$$

Of course, the solution of this system of difference equation does not coincide with the solution of the original equations (C.32) to (C.37), and in this section we will write $a(t; \eta)$, $C_o(t, s; \eta)$ and so on to emphasize the distinction.

Equations (C.42) can be directly interpreted as determining $\Sigma_C(t, s)$ and $\Sigma_R(t, s)$ on the grid $t, s \in \mathbb{T}$. Finally, we discretize Eq. (C.19) as

$$\sum_{s \in \mathbb{T}} [\mathbf{1}_{t=s} + \Sigma_R(t, s)\eta] \, R_A(s, t') \; = \frac{1}{\eta} \mathbf{1}_{t=t'} \,,$$

$$\sum_{s \in \mathbb{T}} [\mathbf{1}_{t=s} + \Sigma_R(t, s)\eta] \, C_A(s, t') \; + \sum_{s \in \mathbb{T}} \Sigma_C(t, s) R_A(t', s)\eta = 0 \,. \tag{D.2}$$

Note that we dropped the integration limits here, since they are enforced by the causality constraints implying $\Sigma_R(t, s) = 0$, $R_A(t, s) = 0$ for $t < s$. Defining the matrices $\boldsymbol{\Sigma}_R = (\Sigma_R(t, s) : t, s \in \mathbb{T})$, and similarly for $\boldsymbol{\Sigma}_C$, $\boldsymbol{C}_A$, $\boldsymbol{R}_A$, we can rewrite (D.2) as

$$[\boldsymbol{I} + \eta \boldsymbol{\Sigma}_R] \, \boldsymbol{R}_A = \frac{1}{\eta} \boldsymbol{I} \,, \tag{D.3}$$

$$[\boldsymbol{I} + \eta \boldsymbol{\Sigma}_R] \, \boldsymbol{C}_A \; + \eta \boldsymbol{\Sigma}_C \boldsymbol{R}_A = \boldsymbol{0} \,. \tag{D.4}$$

We truncate these matrices (which are infinite) to a maximum time $T$ (e.g., redefine $\boldsymbol{\Sigma}_R = (\Sigma_R(t, s) : t, s \in \mathbb{T}, s, t \leq T)$) and solve these equations by matrix inversion:

$$\boldsymbol{R}_A = \frac{1}{\eta} \big( \boldsymbol{I} + \eta \boldsymbol{\Sigma}_R \big)^{-1} \,, \tag{D.5}$$

$$\boldsymbol{C}_A = -\big( \boldsymbol{I} + \eta \boldsymbol{\Sigma}_R \big)^{-1} \boldsymbol{\Sigma}_C \big( \boldsymbol{I} + \eta \boldsymbol{\Sigma}_R \big)^{-1} \,. \tag{D.6}$$

We denote by $a(t; \eta)$, $\boldsymbol{v}(t; \eta)$, $C_o(t, s; \eta)$, $C_d(t, s; \eta)$, $R_o(t, s; \eta)$, $R_d(t, s; \eta)$, the functions obtained via the Euler integration scheme. We will assume that this solution is interpolated continuously for $t, s \notin \mathbb{T}$. For instance, for $i, j \in \mathbb{N}$ $a, b \in [0, 1)$, we let

$$C_d((i + a)\eta, (j + b)\eta; \eta) = (1 - a)(1 - b) \, C_d(i\eta, j\eta; \eta) + a(1 - b) \, C_d((i + 1)\eta, j\eta; \eta) \tag{D.7}$$
$$+ (1 - a)b \, C_d((i + 1)\eta, j\eta; \eta) + ab \, C_d((i + 1)\eta, (j + 1)\eta; \eta) \,.$$

Finally, while we described the discretization procedure for the SymmDMFT, the discussion above applies verbatimly for the full DMFT of Section C.1.

The DMFT equations and their symmetric specialization have a causal structure which means that they can be integrated by progressively by increasing $T$. Furthermore there is no self-consistency condition in the integration scheme at variance with the non-Gaussian settings, see for example [40]. This simplification allows to investigate the long time behavior of the dynamics in a numerical, rather efficient, way.

### D.2 Accuracy of the numerical integration scheme

The discretization of DMFT is expected to converge to the actual solution with errors of order $\eta$. Namely, we expect

$$C_d(t, t'; \eta) = C_d(t, t') + O(\eta) \,, \qquad C_o(t, t'; \eta) = C_o(t, t') + O(\eta) \,, \tag{D.8}$$

and similarly for the other functions. We refer to [13] for related examples in which the convergence was proved rigorously, and to [31] for an empirical study in a closely related model.

In order to test the accuracy of our approach, and the correctness of the DMFT equations, we simulated the gradient descent (GD) dynamics for the Gaussian model. Namely, we generate realizations of the process $\boldsymbol{f}^g(\boldsymbol{a}, \boldsymbol{W}) = (f_i^g(\boldsymbol{a}, \boldsymbol{W}) : i \leq n)$ with the prescribed covariance (A.4), and the vector $\boldsymbol{\varphi}^g = (\varphi_i^g : i \leq n)$ with same covariance as in Eq. (A.6) (see Section D.4.) We define $\widehat{\mathscr{R}}_n(\boldsymbol{a}, \boldsymbol{W})$ via Eq. (A.8) and implement the following GD iteration

$$\boldsymbol{a}^n(t + \eta_{\mathrm{GD}}) = \boldsymbol{a}^n(t) - \frac{\eta_{\mathrm{GD}} n}{d} \nabla_{\boldsymbol{a}} \widehat{\mathscr{R}}_n(\boldsymbol{a}^n(t), \boldsymbol{W}^n(t)) \,,$$

$$\boldsymbol{w}_i^n(t + \eta_{\mathrm{GD}}) = \boldsymbol{P}_{\mathbb{S}^{d-1}} \left( \boldsymbol{w}_i^n(t) - \frac{\eta_{\mathrm{GD}} n}{d} \nabla_{\boldsymbol{w}_i} \widehat{\mathscr{R}}_n(\boldsymbol{a}^n(t), \boldsymbol{W}^n(t)) \right) \,, \tag{D.9}$$

where $\boldsymbol{P}_{\mathbb{S}^{d-1}}$ is the projector to the unit sphere, i.e. $\boldsymbol{P}_{\mathbb{S}^{d-1}}(\boldsymbol{x}) = \boldsymbol{x}/\|\boldsymbol{x}\|$ if $\boldsymbol{x} \neq \boldsymbol{0}$ and $\boldsymbol{P}_{\mathbb{S}^{d-1}}(\boldsymbol{0}) = \boldsymbol{0}$. Note that the trajectories of Eq. (D.9) depend on the sample size $n$ (and hence the dimension $d = d_n$) and the stepsize $\eta_{\mathrm{GD}}$. To emphasize this dependence, we also use the notation $\boldsymbol{a}^n(t; \eta_{\mathrm{GD}})\, \boldsymbol{W}^n(t; \eta_{\mathrm{GD}})$.

We expect the GD trajectories defined by Eq. (D.9) approach the GF trajectories defined by Eq. (A.10) as $\eta_{\mathrm{GD}} \to 0$ uniformly in $n, d$. Namely,

$$\lim_{\eta_{\mathrm{GD}} \to 0} \limsup_{n,d \to \infty} \|\boldsymbol{W}^n(t; \eta_{\mathrm{GD}}) - \boldsymbol{W}^n(t)\|_F = 0\,, \tag{D.10}$$

$$\lim_{\eta_{\mathrm{GD}} \to 0} \limsup_{n,d \to \infty} \|\boldsymbol{a}^n(t; \eta_{\mathrm{GD}}) - \boldsymbol{a}^n(t)\|_2 = 0\,, \tag{D.11}$$

where the limits are understood to hold in probability for any fixed $t$. Informally, for fixed small $\eta_{\mathrm{GD}}$, GD dynamics is a good approximation to GF dynamics, irrespective of the dimension.

We generate several realizations of the processes $\boldsymbol{f}^g$, $\boldsymbol{\varphi}^g$, and of the gradient descent trajectories (D.9). We average observables of interest over these realizations and compare these with the Euler discretization of the DMFT equations. For instance, consider the correlation functions $C_{ij}(t, s)$. Then we can compare:

- $C_{ij}^n(t, s; \eta_{\mathrm{GD}}) = \mathbb{E}\langle \boldsymbol{w}_i^n(t; \eta_{\mathrm{GD}}), \boldsymbol{w}_j^n(s; \eta_{\mathrm{GD}})\rangle$ where the expectation is taken with respect to the GD process (D.9).

- $C_{ij}(t, s; \eta)$, the solution of the Euler discretization of the DMFT, described in the previous section.

Some results of this comparison are presented in the next subsection. This comparison allows us to gauge two types of systematic effects:

1. The effect of finite $n, d$. Indeed, the DMFT equations characterize the $n, d \to \infty$ limit of the GD dynamics (D.9).

2. The non-zero stepsize $\eta$. Note that the effect of discretization introduced in the DMFT equations are different from the ones in the gradient descent (D.9). Therefore the disagreement between the two is a measure of the nonzero-$\eta$ effects.

To clarify further the last point, we emphasize that, despite the notation, $C_{ij}(t, s; \eta)$ is not the $n, d \to \infty$ limit of $C_{ij}^n(t, s; \eta)$.

We note in passing that it is possible to derive DMFT equations for GD, hence characterizing $\lim_{n \to \infty} C_{ij}^n(t, s; \eta_{\mathrm{GD}})$. Similar characterizations were obtained for related (simpler) models in [40, 13, 41, 31, 32]. We defer the analysis of GD with large stepsizes to future work.

### D.3 Testing the numerical accuracy

Figures 6 and 7 we present examples of the numerical comparison described in the previous section, under two different settings, as described below.

**Setting 1.** We assume pure noise data with $\tau = 1$ and train a network with $m = 4$ neurons and covariance structure given by $h(z) = z/10 + z^2/2$. We simulate GD trajectories, according to Eq. (D.9) with $d = 100$, $n = 150$, and correspondingly evaluate the Euler discretization of DMFT, cf. Section D.1 for $\overline{\alpha} = n/d = 1.5$.

We choose an initialization that is not symmetric and therefore we have to use the full DMFT equations of Section C.1. More precisely, we initialize second layer weights as follows:

$$a_1(0) = a_2(0) = 1 \quad a_3(0) = a_4(0) = -1 \tag{D.12}$$

The weights of the first layer are instead initialized by generating two random vectors $\boldsymbol{y}_1, \boldsymbol{y}_2 \sim \mathrm{Unif}(\mathbb{S}^{d-1})$, and setting

$$\boldsymbol{w}_1(0) = \boldsymbol{w}_3(0) = \boldsymbol{y}_1 \quad \boldsymbol{w}_2(0) = \boldsymbol{w}_4(0) = \boldsymbol{y}_2 \tag{D.13}$$

This initialization results in initializing the DMFT equations with

$$\begin{aligned}
&C_{11}(0, 0) = C_{22}(0, 0) = C_{33}(0, 0) = C_{44}(0, 0) = 1\,, \\
&C_{13}(0, 0) = C_{24}(0, 0) = 1\,, \\
&C_{12}(0, 0) = C_{14}(0, 0) = C_{23}(0, 0) = C_{34}(0, 0) = 0\,.
\end{aligned} \tag{D.14}$$

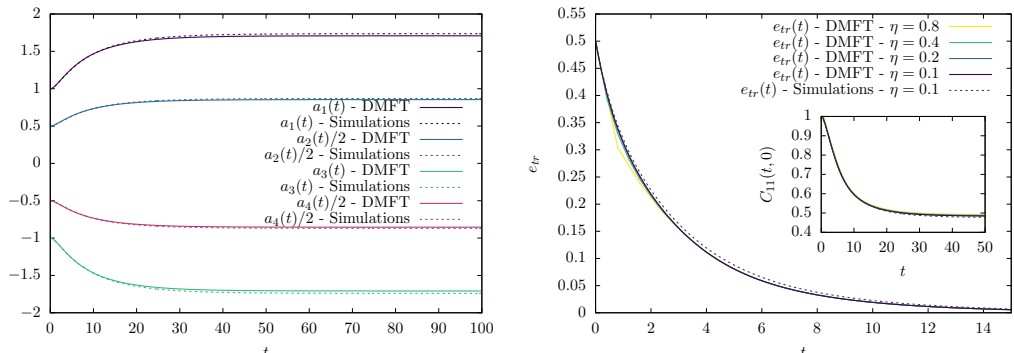

Figure 6: **Comparison between discretized DMFT and GD dynamics for the Gaussian model (labeled as 'Simulations').** GD results are averaged over $N = 10^4$ realizations of the Gaussian process, under Setting 1 described in the main text. Left frame: Second layer for DMFT and GD with $\eta_{\mathrm{GD}} = \eta = 0.1$. Right frame: Train error and correlation function for DMFT with a few values of $\eta$, and for GD.

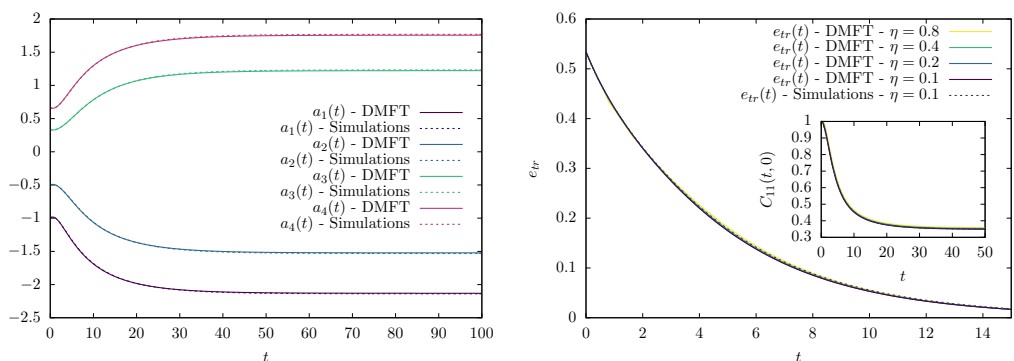

Figure 7: **Comparison between discretized DMFT and GD dynamics (labeled as 'Simulations').** GD results are averaged over $N = 10^4$ realizations of the Gaussian process, under Setting 2 described in the main text. Results for GD are averaged over $N = 10^4$ samples.

Both for the discretized DMFT and for GD for several values of the stepsize. The results of this analysis are plotted in Fig. 6.

**Setting 2.** We consider again pure noise with $\tau = 1$, a network with $m = 4$, input dimension $d = 100$ and sample size $n = 150$. We use hidden neurons with the same covariance structure as in the Setting 1.

However, we change the initialization with respect to Setting 1. First layer are initialized independently and uniformly at random. It follows that

$$C_{ij}(0,0) = \delta_{ij} \quad \forall i,j = 1, \ldots, 4 \tag{D.15}$$

Second layer weights are initialized according to

$$a_1(0) = -1, \quad a_2(0) = -\frac{1}{2}, \quad a_3(0) = \frac{1}{3}, \quad a_4(0) = \frac{2}{3}. \tag{D.16}$$

We use stepsize $\eta = 0.1$.

## D.4 Construction of the Gaussian process $f^g(\,\cdot\,)$

The Gaussian process $f^g$ can be constructed as follows. Define a sequence of independent Gaussian tensors $\boldsymbol{J}^{(k)} \in (\mathbb{R}^d)^{\otimes k}$, $k \geq 1$, with entries $(J^{(k)}_{i_1,\ldots,i_k} : i_j \leq d) \sim_{iid} \mathsf{N}(0,1)$. We then let

$$f^g(\boldsymbol{a}, \boldsymbol{W}) = \frac{1}{m} \sum_{i=1}^{m} a_i \sum_{k=0}^{\infty} c_k \sum_{i_1,\ldots,i_k=1}^{d} J^{(k)}_{i_1,\ldots,i_k} w_{i,i_1} \ldots w_{i,i_k} \tag{D.17}$$

It is easy to check that this stochastic process has the prescribed covariance, with

$$h(z) = \sum_{k=0}^{\infty} c_k^2 z^k \,, \tag{D.18}$$

has long as the series above has radius of convergence larger than 1. An analogous construction holds for $\varphi^g$.

# E   Dynamical regimes: General preliminaries

In the next two sections, we will study the SymmDMFT equations of Section C.4 and characterize different dynamical regimes in the large network limit. From a technical viewpoint, we develop a *singular perturbation theory* of the DMFT equations as $m \to \infty$ for fixed overparametrization ratio $\alpha = \overline{\alpha}/m$.

While singular perturbation theory is a classical domain of mathematics [9, 29], making this type of analysis rigorous is notoriously challenging. We will proceed heuristically as follows: $(i)$ Hypothesize a certain asymptotic behavior of the DMFT solution in a specific time-scale; $(ii)$ Check consistency with the DMFT equations; $(iii)$ Check that this behavior is observed in the numerical solution of the DMFT equations.

More precisely, a specific dynamical regime is identified by a scaling of the time variable, which in our case will take the form $t = t_{\#}(m) \cdot \hat{t}$ for a certain fixed function $t_{\#}(m)$ and $\hat{t}$ a scaled time of order one. The asymptotics of DMFT quantities in that regime takes the form (for instance)

$$\lim_{m \to \infty} \frac{1}{c_{\#}(m)} C_o\Big(t_{\#}(m) \cdot \hat{t}, t_{\#}(m) \cdot \hat{s}; m, \overline{\alpha} = \frac{\alpha}{m}\Big) = c_o(\hat{t}, \hat{s}; \alpha) \,, \tag{E.1}$$

where $c_{\#}(m)$, $c_o(\hat{t}, \hat{s}; \alpha)$ are two fixed functions, the limit is understood to hold at fixed $\hat{t}, \hat{s}, \alpha \in (0, \infty)$, and we made explicit the dependence of $C_o$ on $m, \overline{\alpha}$. More concisely, we will often write the above formula as

$$C_o\Big(t_{\#}(m) \cdot \hat{t}, t_{\#}(m) \cdot \hat{s}; m, \overline{\alpha} = \frac{\alpha}{m}\Big) = c_{\#}(m) c_o(\hat{t}, \hat{s}; \alpha) + o(c_{\#}(m)) \,, \tag{E.2}$$

and we will typically use $t$, $s$ instead of $\hat{t}, \hat{s}$ for the dummy variables.

The behavior of the DMFT equations depends in a crucial way in the initialization of the second layer weights:

- In Section F, we will consider the case of a 'lazy initialization,' i.e. we will assume $a(0) = \gamma_0 \sqrt{m}$ for some constant $\gamma_0 \in (0, \infty)$ independent of $m$.

- In Section G, we will consider the 'mean field initialization' i.e. assume $a(0) = a_0$ to be constant and independent of $m$.

# F   Dynamical regimes: Lazy initialization

As anticipated, in this section we study dynamical regimes under lazy initialization. In subsection F.1, we will consider the case of pure noise data and in subsection F.2 the $k$-index model.

Throughout this section, we let $\gamma(t) = a(t)/\sqrt{m}$ (in particular, $\gamma(0) = \gamma_0$).

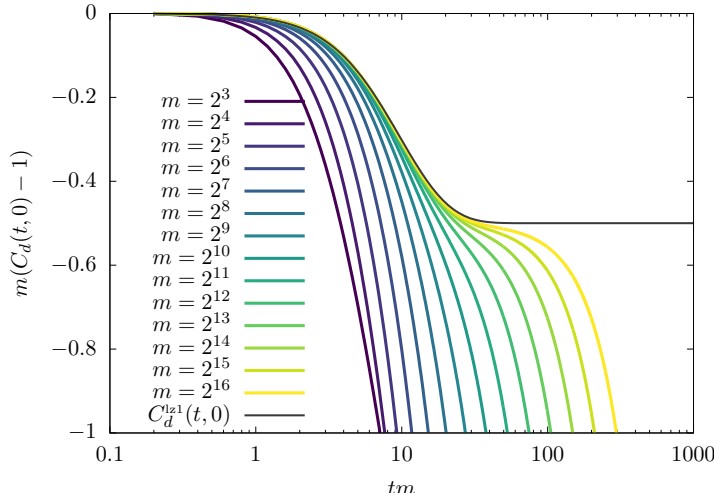

Figure 8: **Training of pure noise data: first dynamical regime.** Rescaled correlation function $m(C_d(t,s)-1)$ in the first dynamical regime as a function of the scaled time $tm$ for a model initialized with a lazy scaling and fixed second layer weights. Different curves correspond to the numerical integration of the SymmDMFT equations at various values of $m$. They appear to converge to the scaling solution in the large $m$ limit described by Eqs. (F.6). Here $\alpha = 0.5$, $\tilde{h}(z) = (3/10)z + z^2/2$ and $\tau = 1$.

### F.1 Pure noise model

Under the pure noise model, we have $\varphi = \hat{\varphi} = 0$. Further, the variable $\boldsymbol{v}(t)$ is not defined and can be dropped (equivalently, we can set $\boldsymbol{v}(t) = \boldsymbol{0}$).

We identify three dynamical regimes:

1. $t = O(1/m)$: $\gamma(t) = \gamma_0 + o_m(1)$, train error decreases, and the network approximates the null function (Section F.1.1).

2. $t = \Theta(1)$: $\gamma(t) = \gamma_0 + o_m(1)$, first-layer weights move significantly and train error converges to a limit $e_*(\gamma_0)$ (Section F.1.2). If $\gamma_0$ is larger than the interpolation threshold, then train error vanishes in this regime.

3. $t = \Theta(m)$: This regime emerges only if $\gamma_0$ is smaller than the interpolation threshold. (We discuss the identification of the interpolation transition of gradient flow in Section F.1.3.)

   If this is the case, $\gamma(t)$ grows on the time scale $t = \Theta(m)$ until it crosses the interpolation threshold. At that point the train error vanishes (Section F.1.4).

Since in the first two regimes $\gamma(t)$ does not change appreciably, the dynamics in these time scales is essentially equivalent to the one of a network in which second-layer weights are fixed and do not evolve by GF. In Section F.1.1 and F.1.2 we first consider this case.

We note that the pure noise model is unchanged if we rescale $\tau \to c\tau$, $\gamma_0 \to c\gamma_0$. More precisely, this results in a rescaling of the risk by $c^2$ and hence of time by the same factor. As a consequence quantities of interest often depend on $\gamma, \tau$ uniquely through their ratio $\gamma/\tau$.

#### F.1.1 First dynamical regime: $t = O(1/m)$

We first consider the case in which the (scaled) second layer weights are not updated and fixed to their initialization, i.e. $\gamma(t) = \gamma_0$.

It is possible to check that, up to higher-order terms, the SymmDMFT equations are solved by functions of the form (the first equation holds in weak sense, i.e. after integrating against a test function)

$$R_A(t/m, s/m) = m\,\delta(t-s) + o_m(m) \qquad C_A(t/m, s/m) = C_A^{\mathrm{lz1}}(t,s) + o_m(1) \qquad \text{(F.1)}$$

$$R_o(t/m, s/m) = \frac{1}{m}R_o^{\mathrm{lz1}}(t,s) + o_m(1/m) \quad C_o(t/m, s/m) = \frac{1}{m}C_o^{\mathrm{lz1}}(t,s) + o_m(1/m) \quad \text{(F.2)}$$

$$R_d(t/m, s/m) = \vartheta(t-s) + o_m(1) \qquad C_d(t/m, s/m) = 1 + \frac{1}{m}C_d^{\mathrm{lz1}}(t,s) + o_m(1/m) \tag{F.3}$$

$$\nu(t/m) = \nu^{\mathrm{lz1}}(t) + o_m(1)\,. \tag{F.4}$$

where $C_A^{\mathrm{lz1}}$, $C_d^{\mathrm{lz1}}$, $C_o^{\mathrm{lz1}}$, $\nu^{\mathrm{lz1}}$ and $R_o^{\mathrm{lz1}}$ are suitable functions independent of $m$. Here and below, we use the notation $\vartheta(t) = \mathbf{1}(t > 0)$.

Note that Eq. (F.3) implies that on this dynamical regime the weights of the first layer change by order $1/m$.

Plugging the asymptotic form in Eqs. (F.1) to (F.4) into the SymmDMFT equations and matching the leading orders for large $m$, we obtain that the functions $C_A^{\mathrm{lz1}}$, $C_d^{\mathrm{lz1}}$, $C_o^{\mathrm{lz1}}$, $\nu^{\mathrm{lz1}}$ and $R_o^{\mathrm{lz1}}$ must satisfy

$$\begin{aligned}
\nu^{\mathrm{lz1}}(t) &= -\alpha\gamma_0^2 h'(1) - \alpha\gamma_0^2 h'(0)C_o^{\mathrm{lz1}}(t,t)\,, \\
\partial_t R_o^{\mathrm{lz1}}(t,t') &= -\alpha\gamma_0^2 h'(0)\left(1 + R_o^{\mathrm{lz1}}(t,t')\right)\,, \\
\partial_t C_o^{\mathrm{lz1}}(t,t') &= -\alpha\gamma_0^2 h'(0)\left(1 + C_o^{\mathrm{lz1}}(t,t')\right)\,, \\
\partial_t C_d^{\mathrm{lz1}}(t,t') &= \alpha\gamma_0^2 h'(0)\left(C_o^{\mathrm{lz1}}(t,t) - C_o^{\mathrm{lz1}}(t,t')\right)\,, \\
C_A^{\mathrm{lz1}}(t,s) &= -\left[\tau^2 + \gamma_0^2 h(1) - \gamma_0^2 h(0)C_o^{\mathrm{lz1}}(t,s)\right]\,.
\end{aligned} \tag{F.5}$$

These are a set of ordinary differential equations that can be solved explicitly. We get

$$\begin{aligned}
R_o^{\mathrm{lz1}}(t,t') &= \vartheta(t-t')\left[e^{-\alpha\gamma_0^2 h'(0)(t-t')} - 1\right]\,, \\
C_o^{\mathrm{lz1}}(t,t) &= e^{-2\alpha\gamma_0^2 h'(0)t} - 1\,, \\
C_o^{\mathrm{lz1}}(t,t') &= -1 + e^{-\alpha\gamma_0^2 h'(0)(t-t')}(C_o^{\mathrm{lz1}}(t',t') + 1) \qquad \text{for } t > t'\,, \\
C_d^{\mathrm{lz1}}(t,t') &= 1 + e^{-\alpha\gamma_0^2 h'(0)(t+t')} - \frac{1}{2}\left(e^{-2\alpha\gamma_0^2 h'(0)t} + e^{-2\alpha\gamma_0^2 h'(0)t'}\right)\,, \qquad \text{for } t > t'\,.
\end{aligned} \tag{F.6}$$

In particular, Eqs. (F.6) imply

$$\begin{aligned}
\lim_{t\to\infty} C_o^{\mathrm{lz1}}(t,t) &= -1\,, \\
\lim_{t,t'\to\infty,\, t-t'\to\infty} R_o^{\mathrm{lz1}}(t,t') &= -1\,.
\end{aligned} \tag{F.7}$$

Recalling Eq. (F.2) we conclude that

$$\lim_{t\to\infty}\lim_{m\to\infty} m\, C_o(t/m, t/m) = -1\,, \tag{F.8}$$

or, using the interpretation of $C_o$,

$$\lim_{t\to\infty}\lim_{m\to\infty}\lim_{n\to\infty} m\cdot\langle \boldsymbol{w}_i(t/m), \boldsymbol{w}_j(t/m)\rangle = -1 \quad \forall i \neq j\,. \tag{F.9}$$

In other words, at the end of this dynamical regime, the first-layer weights form a regular simplex, with center $\overline{\boldsymbol{w}}(t/m) := m^{-1}\sum_{i=1}^m \boldsymbol{w}_i(t/m)$ satisfying $\|\overline{\boldsymbol{w}}(t/m)\|^2 = o_m(1)$.

Hence, at the end of the first dynamical regime, the first-layer weights are such that the linear component of the activation function $\sigma$ is removed. In other words, for $t$ a large constant, we have

$$f^g(\,\cdot\,; \boldsymbol{a}(t/m), \boldsymbol{W}(t/m)) = \frac{\gamma_0}{\sqrt{m}}\sum_{i=1}^m \sigma_G^{\mathrm{nl}}(\boldsymbol{w}_i(t/m)) + \mathrm{err}\,, \tag{F.10}$$

where $\sigma_G^{\mathrm{nl}}(\boldsymbol{w})$ is a Gaussian process with covariance structure given by $h(z) - zh'(0)$, and $\mathrm{err}$ is small in mean square.

Notice also that this is achieved by a $O(1/\sqrt{m})$ change in each of the first layer weights. Indeed, by Eq. (F.3), we have

$$\lim_{n\to\infty}\|\boldsymbol{w}_i(0) - \boldsymbol{w}_i(t/m)\|^2 = 2 - 2C_d(0, t/m) = -\frac{2}{m}C_d^{\mathrm{lz1}}(0,t) + o_m(1/m)\,. \tag{F.11}$$

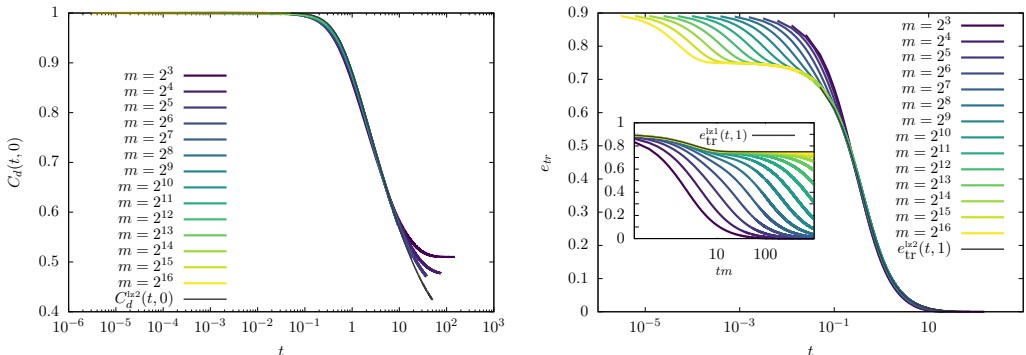

Figure 9: **Training with pure noise data under lazy initialization: second dynamical regime** $t = \Theta(1)$. Left panel: First-layer weights correlation function $C_d(t, 0)$ measuring the inner product between neurons at time $0$ and time $t$, plotted versus $t$ for several values of $m$, and compared with the large $m$-asymptotics $C_d^{\mathrm{lz2}}$. Right panel: training error a $e_{\mathrm{tr}}(t, \gamma_0, m)$ plotted versus $t$ for several values of $m$, and compared with the large-$m$ asymptotics in this regime $e_{\mathrm{tr}}^{\mathrm{lz2}}(t, 1)$. Notice the two-steps decrease of the training error, corresponding to the two regimes $t = O(1/m)$ and $t = \Theta(1)$. Inset: Same curves plotted versus $tm$, and compared with the asymptotic prediction $e_{\mathrm{tr}}^{\mathrm{lz1}}(\,\cdot\,, 1)$ in the first dynamical regime. For both panels we use $\alpha = 0.5$, $\tilde{h}(z) = (3/10)z + z^2/2$, $\gamma_0 = 1$ and $\tau = 1$.

Equations (F.1) to (F.4) can be used to compute the behavior of the train error in this dynamical regime:

$$\lim_{m \to \infty} e_{\mathrm{tr}}(t/m) = e_{\mathrm{tr}}^{\mathrm{lz1}}(t) . \tag{F.12}$$

Using Eqs. (F.6), we get the expression:

$$e_{\mathrm{tr}}^{\mathrm{lz1}}(t) = \frac{1}{2} \left[ \tau^2 + \gamma_0^2 h(1) + \gamma_0^2 h'(0) C_o^{\mathrm{lz1}}(t, t) \right] . \tag{F.13}$$

In particular, the train error at the end of this dynamical regime is

$$\lim_{t \to \infty} \lim_{m \to \infty} e_{\mathrm{tr}}(t/m) = \lim_{t \to \infty} e_{\mathrm{tr}}^{\mathrm{lz1}}(t) = \frac{1}{2} \left[ \tau^2 + \gamma_0^2 h(1) - \gamma_0^2 h'(0) \right] . \tag{F.14}$$

This is in agreement with (F.10). Indeed, note that

$$\widehat{\mathscr{R}}_n(\boldsymbol{a}, \boldsymbol{W}) = \frac{1}{2n} \|\boldsymbol{\varepsilon}\|^2 - \frac{1}{n} \langle \boldsymbol{\varepsilon}, \boldsymbol{f}^g(\boldsymbol{a}, \boldsymbol{W}) \rangle + \frac{1}{2n} \|\boldsymbol{f}^g(\boldsymbol{a}, \boldsymbol{W})\|^2 . \tag{F.15}$$

Training in this timescale attempts to minimize $\|\boldsymbol{f}^g(\boldsymbol{a}, \boldsymbol{W})\|^2$ without fitting the noise.

This picture is confirmed by the fact that Eqs. (F.6) depend on $h$ only through $h'(0)$. This means that the dynamics on timescales of order $1/m$ is controlled by the linear part of the covariance structure of the hidden layer.

In Fig.8 we test the correctness of the asymptotic ansatz of Eqs. (F.1) to (F.4). Namely, we compare the results of numerical integration of the SymmDMFT equations for various values of $m$, with the prediction of Eqs. (F.6). The match is excellent.

So far we assumed that second-layer weights are not optimized and $\gamma(t) = \gamma_0$. What happens if drop this constraint? It can be checked that the form given in Eqs. (F.1)-(F.4) still solves the SymmDMFT equations when $a(t)$ is allowed to evolve, and $\gamma(t/m) = \gamma_0 + o_m(1)$ for all fixed $t \in (0, \infty)$. In other words, second layer weights do not change significantly during this dynamical regime.

### F.1.2 Second dynamical regime: $t = \Theta(1)$

The second dynamical regime arises when $t = \Theta(1)$. Recall from the previous subsection that, for $t = o_m(1)$, the train error remains close (for large $m$) to the plateau characterized at the end of the first dynamical regime, see Eq. (F.14). When $t$ is of order one, the first layer weights start changing by an amount of order one as well, and the model starts to fit the noise.

As before, we begin by considering the simplified setting in which $\gamma(t) = \gamma_0$ is fixed and not optimized by GF.

We claim that the SymmDMFT equations are solved by the following ansatz, up to lower order terms as $m \to \infty$:

$$\nu(t) = \nu^{\mathrm{lz2}}(t) + o_m(1)\,, \tag{F.16}$$

$$C_d(t, t') = C_d^{\mathrm{lz2}}(t, t') + o_m(1)\,, \tag{F.17}$$

$$R_d(t, t') = R_d^{\mathrm{lz2}}(t, t') + o_m(1)\,, \tag{F.18}$$

$$C_o(t, t') = \frac{1}{m} C_o^{\mathrm{lz2}}(t, t') + o_m(1/m) = -\frac{1}{m} C_d^{\mathrm{lz2}}(t, t') + o_m(1/m)\,, \tag{F.19}$$

$$R_o(t, t') = \frac{1}{m} R_o^{\mathrm{lz2}}(t, t') + o_m(1/m) = -\frac{1}{m} R_d^{\mathrm{lz2}}(t, t') + o_m(1/m)\,. \tag{F.20}$$

Here $C_d^{\mathrm{lz2}}$, $R_d^{\mathrm{lz2}}$, $C_o^{\mathrm{lz2}}$, $R_o^{\mathrm{lz2}}$ and $\nu^{\mathrm{lz2}}$ are certain functions independent of $m$. Equations (F.19), (F.20) state in particular that $C_o^{\mathrm{lz2}}(t, t') = -C_d^{\mathrm{lz2}}(t, t')$ and $R_o^{\mathrm{lz2}}(t, t') = -R_d^{\mathrm{lz2}}(t, t')$, and the therefore we are left with the task of determining $C_d^{\mathrm{lz2}}(t, t')$, $R_d^{\mathrm{lz2}}(t, t')$. By substituting Eqs. (F.16) to (F.20) into the SymmDMFT equations and matching leading order terms, we get a set of two integral-differential equations for $C_d^{\mathrm{lz2}}(t, t')$, $R_d^{\mathrm{lz2}}(t, t')$, which we next state.

We first define

$$\Sigma_R^{\mathrm{lz2}}(t, s) := \gamma_0^2 \left( h'(C_d^{\mathrm{lz2}}(t, s)) - h'(0) \right) R_d^{\mathrm{lz2}}(t, s)\,,$$
$$\Sigma_C^{\mathrm{lz2}}(t, s) := \tau^2 + \gamma_0^2 h(C_d^{\mathrm{lz2}}(t, s)) - \gamma_0^2 h'(0) C_d^{\mathrm{lz2}}(t, s)\,, \tag{F.21}$$

then we define $R_A^{\mathrm{lz2}}$ and $C_A^{\mathrm{lz2}}$ as the solution of

$$\delta(t - t') = \int_{t'}^{t} \left[ \delta(t - s) + \Sigma_R^{\mathrm{lz2}}(t, s) \right] R_A^{\mathrm{lz2}}(s, t')\, \mathrm{d}s\,,$$
$$0 = \int_0^t \left[ \delta(t - s) + \Sigma_R^{\mathrm{lz2}}(t, s) \right] C_A^{\mathrm{lz2}}(s, t')\mathrm{d}s + \int_0^{t'} \Sigma_C^{\mathrm{lz2}}(t, s) R_A^{\mathrm{lz2}}(t', s)\, \mathrm{d}s\,. \tag{F.22}$$

We next define the asymptotic form for the memory kernels

$$M_R^{\mathrm{lz2}}(t, s) := \alpha \left[ R_A^{\mathrm{lz2}}(t, s)\tilde{h}'(C_d^{\mathrm{lz2}}(t, s)) + C_A^{\mathrm{lz2}}(t, s)\tilde{h}''(C_d^{\mathrm{lz2}}(t, s))R_d^{\mathrm{lz2}}(t, s) \right]\,,$$
$$M_C^{\mathrm{lz2}}(t, s) := \alpha \tilde{h}'(C_d^{\mathrm{lz2}}(t, s))C_A^{\mathrm{lz2}}(t, s)\,. \tag{F.23}$$

and we have defined

$$\tilde{h}(z) := h(z) - h'(0)z\,. \tag{F.24}$$

The equations for $\nu^{\mathrm{lz2}}$, $C_d^{\mathrm{lz2}}$ and $R_d^{\mathrm{lz2}}$ are then given by

$$\nu^{\mathrm{lz2}}(t) = -\int_0^t \left[ M_R^{\mathrm{lz2}}(t, s)C_d^{\mathrm{lz2}}(t, s) + M_C^{\mathrm{lz2}}(t, s)R_d^{\mathrm{lz2}}(t, s) \right]\, \mathrm{d}s\,, \tag{F.25}$$

$$\partial_t R_d^{\mathrm{lz2}}(t, t') = \delta(t - t') - \nu^{\mathrm{lz2}}(t)R_d^{\mathrm{lz2}}(t, t') - \int_{t'}^t M_R^{\mathrm{lz2}}(t, s)R_d^{\mathrm{lz2}}(s, t')\, \mathrm{d}s\,, \tag{F.26}$$

$$\partial_t C_d^{\mathrm{lz2}}(t, t') = -\nu^{\mathrm{lz2}}(t)C_d^{\mathrm{lz2}}(t, t') - \int_0^t M_R^{\mathrm{lz2}}(t, s)C_d^{\mathrm{lz2}}(t', s)\, \mathrm{d}s - \int_0^{t'} M_C^{\mathrm{lz2}}(t, s)R_d^{\mathrm{lz2}}(t', s)\, \mathrm{d}s\,. \tag{F.27}$$

(As before, in the second and last equation, it is understood that $t \geq t'$, and the last equation is understood to hold in weak sense.)

Given the constraints on $C_d$, $R_d$, we have the following constraints on $C_d^{\mathrm{lz2}}$, $R_d^{\mathrm{lz2}}$,

$$C_d^{\mathrm{lz2}}(t, t) = 1\,, \tag{F.28}$$

$$C_d^{\mathrm{lz2}}(t, s) = C_d^{\mathrm{lz2}}(s, t)\,, \tag{F.29}$$

$$R_d^{\mathrm{lz2}}(t, s) = 0 \quad \forall t \leq s\,. \tag{F.30}$$

In particular, the last condition, together with Eq. (F.26) implies $R_d^{\mathrm{lz2}}(t+, t) = 1$.

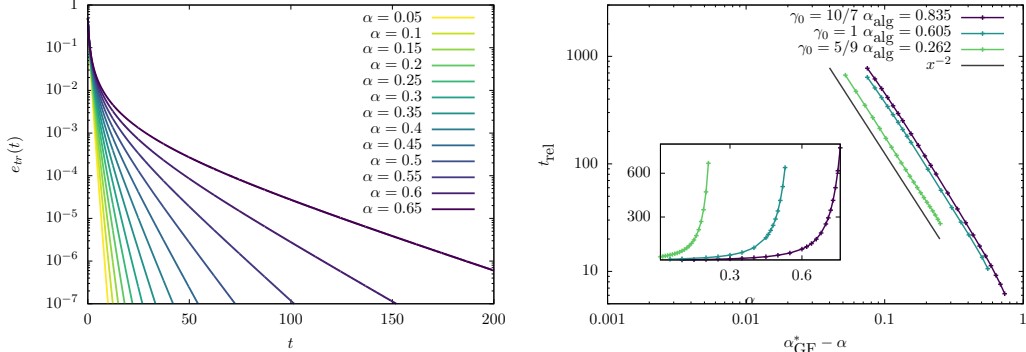

Figure 10: **Training with pure noise data under lazy initialization: algorithmic interpolation threshold.** Left Panel. We plot the train error as a function of time for different values of $\alpha$ at $\gamma_0 = 10/7$. The train error has an exponential decay to zero for $\alpha$ below the interpolation threshold. Right Panel. We plot the time for GF to converge to near-zero training error as a function of $\alpha$, for various values of $\gamma_0$, as computed using the theory of Section F.1.2. The divergence of $t_{\rm rel}$ signals the phase transition for GF interpolation $\alpha_{\rm GF}^*$. Inset: $\tau_{\rm rel}$ versus $\alpha$ in linear scale. Main panel: $t_{\rm rel}$ versus $\alpha_{\rm GF}^* - \alpha$ (with the fitted value of $\alpha_{\rm GF}^*$). Here we use $h(z) = (3/10)z + z^2/2$.

The evolution of the the train error in this second dynamical regime is given by

$$\lim_{m \to \infty} e_{\rm tr}(t) = e_{\rm tr}^{\rm lz2}(t, \gamma_0)\,, \tag{F.31}$$

$$e_{\rm tr}^{\rm lz2}(t, \gamma_0) = -\frac{1}{2} C_A^{\rm lz2}(t, t)\,, \tag{F.32}$$

where we have made explicit the dependence on the initialization of second-layer weights $\gamma_0$.

Note that Eq. (F.22) implies $C_A^{\rm lz2}(0,0) = -\Sigma_C^{\rm lz2}(0,0)$, and Eq. (F.21) yields $\Sigma_C^{\rm lz2}(0,0) = \tau^2 + \gamma_0^2 h(1) - \gamma_0^2 h'(0)$. Therefore

$$\lim_{t \to 0} e_{\rm tr}^{\rm lz2}(t, \gamma_0) = \frac{1}{2}\left[\tau^2 + \gamma_0^2 \tilde{h}(1)\right] = \lim_{t \to \infty} e_{\rm tr}^{\rm lz1}(t, \gamma_0)\,. \tag{F.33}$$

In other words, this second dynamical regime captures the decrease of the training error which starts at the plateau reached in the first regime, cf. Eq. (F.14). which coincides with the long time extrapolation of the first dynamical regime.

This second dynamical regime is fully non-linear and depends on the entire covariance function $\tilde{h}$. Further, the first order weights move by an amount $\|\boldsymbol{w}_i(t) - \boldsymbol{w}_i(0)\| = \Theta(1)$, as follows from the fact that $C_d^{\rm lz2}(t, 0) < 1$ strictly.

In order to confirm the ansatz (F.16) to (F.20), we compared the solution of the full SymmDMFT equations, with the solution of the asymptotic equations (F.25), (F.27). An example of such a comparison is presented in Fig. 9: the agreement is excellent.

The treatment above assumed the constraint $\gamma(t) = \gamma_0$. However, as in the first dynamical regime, if we let second layer weights evolve, they do not change appreciably. Namely, the asymptotic form given in Eqs. (F.16) to (F.20) still solves the SymmDMFT equations when $a(t)$ is allowed to evolve. We have $\gamma(t) = \gamma_0 + o_m(1)$ on this timescale.

### F.1.3 The algorithmic interpolation transition

For the discussion in this section, we denote by $e_{\rm tr}(t, \gamma_0, m, \alpha)$ the train error as a function of $t$, where we emphasized the dependence on the initial condition $\gamma_0$, on the number of neurons $m$, and on the overparametrization ratio $\alpha$. We further assume that second layer weigths are not evolved and therefore $\gamma(t) = \gamma_0$ for all $t$. We define the asymptotic train error achieved by GF as

$$e_{{\rm tr},\infty}(\gamma_0, m, \alpha) := \lim_{t \to \infty} e_{\rm tr}(t, \gamma_0, m, \alpha) \tag{F.34}$$

$$= \lim_{t \to \infty} \lim_{n \to \infty} \widehat{\mathscr{R}}_n(\boldsymbol{a}, \boldsymbol{W}(t))\,. \tag{F.35}$$

Again, in this definition $a_i = \gamma_0\sqrt{m}$ is kept fixed and does not evolve with time.

Notice that it is in principle possible that $\lim_{n\to\infty} \widehat{\mathscr{R}}_n(\boldsymbol{a}, \boldsymbol{W}(t_n))$ is strictly smaller than $e_{\text{tr},\infty}(\gamma_0, m, \alpha)$ if we let $t_n$ diverge with $n$ at sufficiently fast rate. However, based on results on related models in spin-glass theory we expect this not to be the case as long as $t_n$ is polynomial in $n$. Explicitly, we expect that, for any sequence $t_n \to \infty$

$$t_n \leq n^C \quad \Rightarrow \quad \lim_{n\to\infty} \widehat{\mathscr{R}}_n(\boldsymbol{a}, \boldsymbol{W}(t_n)) = e_{\text{tr},\infty}(\gamma_0, m, \alpha). \tag{F.36}$$

Using the reduced equations for $t = \Theta(1)$ timescale, i.e. Eqs. (F.25) to (F.27), we can also define

$$\begin{aligned} e^{\text{lz2}}_{\text{tr},\infty}(\gamma_0, \alpha) &:= \lim_{t\to\infty} e^{\text{lz2}}_{\text{tr}}(t, \gamma_0, \alpha) \tag{F.37} \\ &= \lim_{t\to\infty}\lim_{m\to\infty}\lim_{n\to\infty} \widehat{\mathscr{R}}_n(\boldsymbol{a}, \boldsymbol{W}(t)). \end{aligned}$$

A natural question is whether the large $m$ limit of $e_{\text{tr},\infty}(\gamma_0, m, \alpha)$ coincides with $e^{\text{lz2}}_{\text{tr},\infty}(\gamma_0, \alpha)$. This amounts to asking whether there exists dynamical regime with timescale $t(m)$ diverging with $m$ at which $e_{\text{tr}}(t(m), \gamma_0, m, \alpha)$ starts diverging significantly from the value at the end of the second dynamical regime namely $e^{\text{lz2}}_{\text{tr},\infty}(\gamma_0, \alpha)$. If $e^{\text{lz2}}_{\text{tr},\infty}(\gamma_0, \alpha) = 0$ then of course $\lim_{m\to\infty} e_{\text{tr}}(t(m), \gamma_0, m, \alpha) = 0$ as well.

If however $e^{\text{lz2}}_{\text{tr},\infty}(\gamma_0, \alpha) > 0$, then the answer depends upon whether the second layer weights are evolved with GF:

- In the constrained setting in which second-layer weights do not evolve, we observe (from numerical solutions of SymmDMFT ) that

$$\lim_{m\to\infty} e_{\text{tr},\infty}(\gamma_0, m, \alpha) = e^{\text{lz2}}_{\text{tr},\infty}(\gamma_0, \alpha). \tag{F.38}$$

- In the next section we will see that if $\gamma(t)$ evolves with GF then the train error achieved on a diverging timescale $t = \Theta(m)$ is strictly smaller than $e^{\text{lz2}}_{\text{tr},\infty}(\gamma_0, \alpha)$ and vanishes for large enough $t$.

Note that $e_{\text{tr},\infty}(\gamma_0, m, \alpha)$ and $e^{\text{lz2}}_{\text{tr},\infty}(\gamma_0, \alpha)$ also depend on the noise variance $\tau^2$. However, because of the invariance under rescaling discussed at the beginning of this section (adding $\tau$ as an argument):

$$e^{\text{lz2}}_{\text{tr},\infty}(\gamma_0, \alpha, \tau^2) = \tau^2 \cdot e^{\text{lz2}}_{\text{tr},\infty}(\gamma_0/\tau, \alpha, \tau^2 = 1), \tag{F.39}$$

and similarly for $e_{\text{tr},\infty}(\gamma_0, m, \alpha)$. Because of this relation, we can think that $\tau^2$ is fixed throughout, e.g. $\tau^2 = 1$.

We expect $e_{\text{tr},\infty}(\gamma_0, m, \alpha)$, $e^{\text{lz2}}_{\text{tr},\infty}(\gamma_0, \alpha)$ to be non-increasing in $\gamma_0$, and define the thresholds

$$\gamma_{\text{GF}}(\alpha, m) := \inf\left\{\gamma_0 : e_{\text{tr},\infty}(\gamma_0, m, \alpha) = 0\right\}, \tag{F.40}$$

$$\gamma^*_{\text{GF}}(\alpha) := \inf\left\{\gamma_0 : e^{\text{lz2}}_{\text{tr},\infty}(\gamma_0, \alpha) = 0\right\}. \tag{F.41}$$

(These definitions need to be modified if $\gamma_0 \mapsto e_{\text{tr},\infty}(\gamma_0, m, \alpha)$ is non-monotone.)

Of course, Eq. (F.38) implies

$$\lim_{m\to\infty} \gamma_{\text{GF}}(\alpha, m) = \gamma^*_{\text{GF}}(\alpha). \tag{F.42}$$

The numerical solution of the SymmDMFT equations imply that the curve $\gamma^*_{\text{GF}}(\alpha)$ is monotone increasing with $\alpha$, as also suggested by the Gaussian complexity bound (see Section 2.2 in the main text). Hence we can invert it to get a threshold $\alpha^*_{\text{GF}}(\gamma_0)$: the two descriptions are equivalent.

In order to determine $\alpha^*_{\text{GF}}(\gamma_0)$, we adopt a procedure already implemented in [31] for a simpler model. The procedure is based on the observation (from numerical solutions) that when $e^{\text{lz2}}_{\text{tr},\infty}(\gamma_0, \alpha) = 0$, $e^{\text{lz2}}_{\text{tr}}(t, \gamma_0, \alpha) = \exp(-t/t_{\text{rel}}(\alpha; \varepsilon) + o(t))$ for some $t_{\text{rel}}(\alpha) > 0$ which diverges as $\alpha \uparrow \alpha_{\text{GF}}$.

1. Define a grid of values of $\alpha$, $A_0 = \{\alpha_1, \alpha_2, \ldots, \alpha_K\}$, which are expected to be smaller than $\alpha^*_{\text{GF}}(\gamma_0)$.

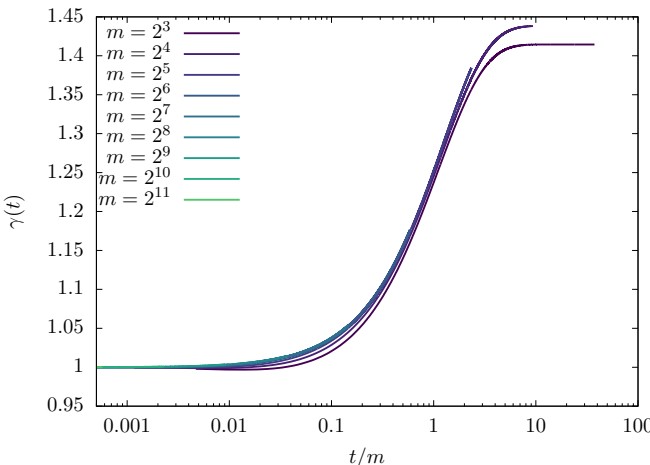

Figure 11: **Training with pure noise data under lazy initialization: second layer weights in the third dynamical regime.** Evolution of the (rescaled) weights of the second layer as a function of $t/m$. Here $\tau = 2.5$ and $\gamma_0 = 1$, $\alpha = 0.3$, and covariance structure for the neurons given by $h(z) = (9/10)z + z^2/2$.

2. For each value $\alpha \in A_0$, integrate numerically the reduced equations (F.25) to (F.27). Verify that $e_{\mathrm{tr}}^{\mathrm{lz2}}(t, \gamma_0, \alpha)$ appear to converge to $0$ with $t \to \infty$. Let $A \subseteq A_0$ be the subset of values for which this happens.

3. For each $\alpha \in A$, define $t_{\mathrm{rel}}(\alpha_i; \varepsilon) := \inf\{t : e_{\mathrm{tr}}^{\mathrm{lz2}}(t, \gamma_0, \alpha_i) < \varepsilon \cdot \tau^2\}$ where $\varepsilon$ is a small threshold value (we use $\varepsilon = 10^{-7}$).

4. Estimate parameters $\alpha_{\mathrm{GF}}^*(\gamma_0), c, \nu$ by fitting the relation $t_{\mathrm{rel}}(\alpha_i; \varepsilon) \sim c(\alpha_{\mathrm{GF}}^* - \alpha_i)^{-\nu}$.

Figure 10 illustrates the calculation of $\alpha_{\mathrm{GF}}^*(\gamma_0)$ for three values of $\gamma_0$. In the inset we plot $t_{\mathrm{rel}}$ for three values of $\gamma_0$ as a function of $\alpha$. In the main panel, we demonstrate the divergence of $t_{\mathrm{rel}}$ when $(\alpha_{\mathrm{GF}}^* - \alpha)$ vanishes. In practice, we observe $\nu = 2$ fit well the data across a variety of settings, suggesting this is the universal exponent for the divergence of $t_{\mathrm{rel}}$.

### F.1.4 Third dynamical regime: $t = \Theta(m)$

In the first two dynamical regimes, the large-$m$ behavior did not depend on whether we would let second layer evolve with GF or we kept them fixed, i.e. $\gamma(t) = \gamma_0$.

In contrast, the behavior on timescales diverging with $m$ depends significantly on the dynamics of second-layer weights.

- If second layer weights are fixed, no significant further evolution takes place. In particular, the training error does not decrease significantly below the value reached at the end of the second dynamical regime, i.e. $e_{\mathrm{tr},\infty}^{\ell}(\gamma_0, \alpha)$. This is stated formally in Eq. (F.38).

- If second layer weights evolve according to GF, then the dynamics on time-scales diverging with $m$ can be non-trivial and depends on the second-layer weights initialization $\gamma_0$. If $\gamma_0 > \gamma_{\mathrm{GF}}^*(\alpha)$, then GR reaches vanishing training error during the second dynamical regime, and no further evolution takes place.

  However, if $\gamma_0 < \gamma_{\mathrm{GF}}^*(\alpha)$, second layer weights start evolving when $t = \Theta(m)$, thus giving rise to a third dynamical regime. This is the object of the present subsection.

In Fig. 11, left frame, we plot the rescaled second layer weights $\gamma(t)$ (as predicted by numerical integration of the SymmDMFT equation) as a function of time for several values of $m$. Here, obviously, we do not constrain $\gamma(t) = \gamma(0)$.

We observe that $\gamma(t)$ changes only when $t = \Theta(m)$. Indeed, when plotted against $t/m$, curves obtained for different values of $m$ collapse onto each other. This suggests that, for $t = o(m)$ $\gamma(t) = \gamma(0) + o_m(1)$ (recall that $\gamma(0) = \gamma_0$ by definition). Further, the curve collapse suggests that,

for any fixed $\hat{t} \in (0, \infty)$:

$$\lim_{m \to \infty} \gamma(\hat{t}\,m, \gamma_0) = \gamma^{\mathrm{lz3}}(\hat{t}, \gamma_0)\,, \tag{F.43}$$

where we have made explicit the dependence on $\gamma_0$. Of course, the case $\gamma_0 > \gamma_{\mathrm{GF}}^*(\alpha)$ fits in this framework with $\gamma^{\mathrm{lz3}}(z, \gamma_0) = \gamma_0$ identically.

We next consider the evolution of the train error. In Fig. 12, left frame, we plot the train error (again, as predicted by numerical integration of the SymmDMFT equation) as a function of time for several values of $m$.

Again, when plotted as a function of $t/m$, curves for different values of $m$ reach a plateau, and collapse below the plateau. This suggests the following limit behavior, which is consistent with Eq. (F.43)

$$\lim_{m \to \infty} \tilde{e}_{\mathrm{tr}}(\hat{t}\,m, \gamma_0, m) = e_{\mathrm{tr}}^{\mathrm{lz3}}(\hat{t}, \gamma_0)\,. \tag{F.44}$$

(Here we use $\tilde{e}_{\mathrm{tr}}(\hat{t}\,m, \gamma_0)$ to denote the train error when second-layer weights evolve, in contrast with $e_{\mathrm{tr}}(\hat{t}\,m, \gamma_0)$ which we used for the setting in which second-layer weights are constrained.)

Matching the present dynamical regime ($t = \Theta(m)$) with previous one ($t = \Theta(1)$, cf. Section F.1.2), implies that

$$\lim_{\hat{t} \to 0+} e_{\mathrm{tr}}^{\mathrm{lz3}}(\hat{t}, \gamma_0) = \lim_{t \to \infty} e_{\mathrm{tr}}^{\mathrm{lz2}}(t, \gamma_0) = e_{\mathrm{tr},\infty}^{\mathrm{lz2}}(\gamma_0) \tag{F.45}$$

In other words, the function $e_{\mathrm{tr}}^{\mathrm{lz3}}$ describes the decrease of the train error below the level $e_{\mathrm{tr},\infty}^{\mathrm{lz2}}(\gamma_0)$ achieved during the second dynamical regime.

In order to characterize the scaling function $e_{\mathrm{tr}}^{\mathrm{lz3}}$, in Fig. 12, right frame, we plot parametrically the the train error for different values of $m$ as a function of the second layer weights $\gamma(t)$. We also plot the curve $(\gamma, e_{\mathrm{tr},\infty}^{\mathrm{lz2}}(\gamma))$. This plot is consistent with the following behavior as $m \to \infty$. In a first regimes (corresponding to $t = o(m)$) the train error has a drop that becomes vertical in the $m \to \infty$ limit, implying that $\gamma(t)$ does not evolve while the train error decreases until it reaches $e_{\mathrm{tr},\infty}^{\ell}(\gamma_0)$. In the last regime (corresponding to $t = \Theta(m)$), $\gamma(t)$ increases together with the decrease of the train error $e_{\mathrm{tr}}^{\ell}(t, \gamma_0)$. Remarkably, they follow the curve $(\gamma, e_{\mathrm{tr},\infty}^{\mathrm{lz2}}(\gamma))$.

In order to describe the last regime, we point out that $t \mapsto \gamma^{\mathrm{lz3}}(t)$ is monotone increasing. Therefore we can re-parametrize time by the value of the second layer weights. Namely, define $\tilde{\gamma}^{-1}$ the inverse function, so that

$$\hat{t} = \tilde{\gamma}^{-1}(\gamma^{\mathrm{lz3}}(\hat{t}, \gamma_0), \gamma_0)\,. \tag{F.46}$$

Using this reparametrization of time, the behavior in Fig. 12 can be formalized as

$$\lim_{t, m \to \infty : \gamma(t, \gamma_0, m) = \tilde{\gamma}} e_{\mathrm{tr}}^{\mathrm{lz3}}(t, \gamma_0, m) = e_{\mathrm{tr}}^{\mathrm{lz3}}(\tilde{\gamma}^{-1}(\tilde{\gamma}, \gamma_0), \gamma_0) =: \varepsilon(\tilde{\gamma}, \gamma_0)\,. \tag{F.47}$$

The collapse on finite $m$ curves in Fig. 12, right frame, onto the curve $(\gamma, e_{\mathrm{tr},\infty}^{\mathrm{lz2}}(\gamma))$ suggests that

$$\gamma > \gamma_0 \quad \Rightarrow \quad \varepsilon(\tilde{\gamma}, \gamma_0) = e_{\mathrm{tr},\infty}^{\mathrm{lz2}}(\gamma)\,. \tag{F.48}$$

In other words, the dynamics on timescales of order $m$ is *adiabatic*: at each increase of $\gamma(t)$ on timescales of order $m$, the train error relaxes to the the value it would have had if the second layer weights would have been fixed in time at the corresponding value of $\gamma$.

A remarkable consequence of Eq. (F.48) is that that

$$\lim_{\hat{t} \to \infty} \gamma^{\mathrm{lz3}}(\hat{t}) = \lim_{m \to \infty} \gamma_{\mathrm{GF}}(\alpha, m) = \gamma_{\mathrm{GF}}^*(\alpha)\,. \tag{F.49}$$

In words, in the large network limit, the norm of second-layer weights at the end of training is asymptotically the minimum norm that allows for interpolation.

## F.2 Multi-index model

In this section we generalize the computations of Section F.1 to the case in which the dataset has a structure produced via a $k$-index model. The weights of the second layer are set to $a(t) = \gamma(t)\sqrt{m}$ and evolve with GF. The initialization scale $\gamma(0) = \gamma_0$ is fixed and independent of $m$.

As in the pure noise case, we identify three dynamical regimes:

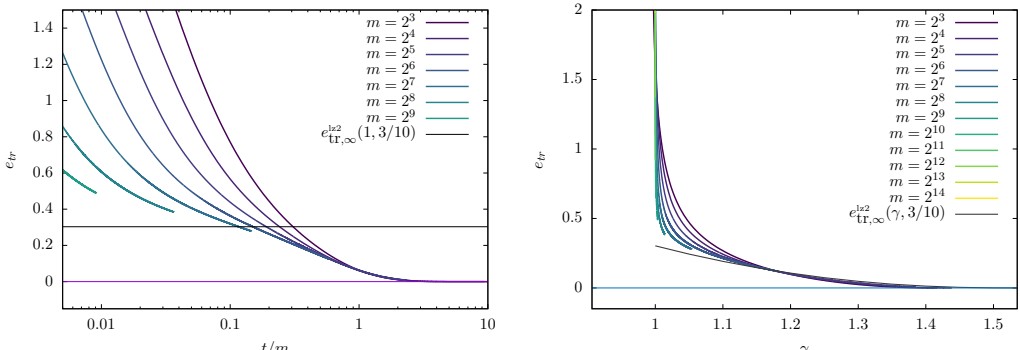

Figure 12: **Training with pure noise data under lazy initialization: third dynamical regime.** Left frame: Train error on timescales of order $m$. Right frame: GF trajectories in the plane $\gamma$ (second layer weights) — $e_{\text{tr}}$ (train error). Black dots represent pairs $(\gamma, e_{\text{tr},\infty}^{\text{lz2}}(\gamma))$, where $e_{\text{tr},\infty}^{\text{lz2}}(\gamma)$ is the train error achieved at the end of the first dynamical regime, cf. Section F.1.3. The data has been produced from the same model as in Fig. 11.

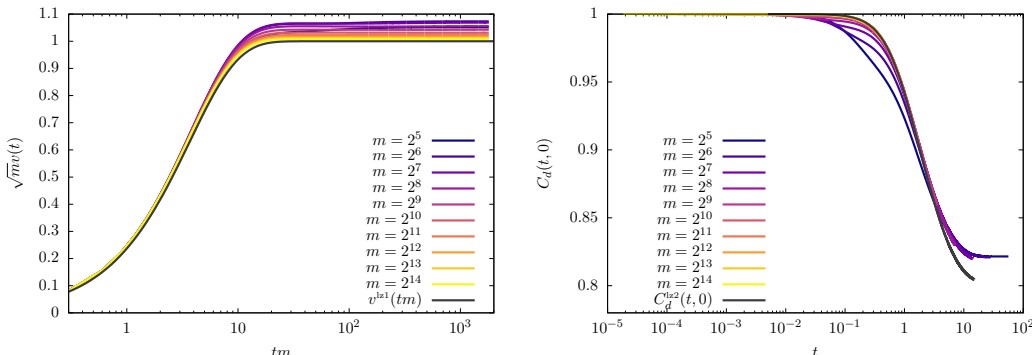

Figure 13: SymmDMFT **predictions and large network scaling for lazy training in a single index model.** Left: Projection $v(t)$ of the first layer weights onto the latent direction on timescales of the order $1/m$. The result for $m \to \infty$, $v^{\text{lz1}}$, has been obtained by integrating analytically Eq. (F.55). Right: The behavior of $C_d(t,0)$ on timescales $t = \Theta(1)$, compared with the scaling theory for $m \to \infty$, namely $C_d^{\text{lz2}}$. In both cases with $h(z) = \hat{\varphi}(z) = (9/10)z + z^2/2$, $\tau = 0.3$ and $\alpha = 0.3$, $\gamma_0 = 1$.

1. $t = O(1/m)$: $\gamma(t) = \gamma_0 + o_m(1)$, $\|\boldsymbol{w}_i(t) - \boldsymbol{w}_i(0)\| = \Theta(1/\sqrt{m})$. On this scale the network only learns a linear approximation of the target. Test and train error remain close to each other (Section F.2.1).

2. $t = \Theta(1)$: $\gamma(t) = \gamma_0 + o_m(1)$, $\|\boldsymbol{w}_i(t) - \boldsymbol{w}_i(0)\| = \Theta(1)$. Test error does not change but train error decreases significantly (Section F.2.2).

3. $t = \Theta(m)$: This regime only emerges if $\gamma_0$ is below a certain interpolation threshold, i.e. $\gamma_0 < \gamma_{\text{GF}}^*(\alpha, \varphi, \tau)$. In this regime $\gamma(t)$ grows until the threshold, and train error decreases to $0$ while test error decreases to $0$ (Section F.2.5).

### F.2.1 First dynamical regime: $t = O(1/m)$

On this timescale, the SymmDMFT equations are solved, up to higher order terms, by the following ansatz:

$$C_d(t/m, s/m) = 1 + o_m(1), \qquad\qquad R_d(t/m, s/m) = \vartheta(t-s) + o_m(1), \qquad \text{(F.50)}$$

$$C_o(t/m, s/m) = \frac{1}{m}C_o^{\text{lz1}}(t,s) + o_m(m^{-1}), \quad R_o(t/m, s/m) = \frac{1}{m}R_o^{\text{lz1}}(t,s) + o_m(m^{-1}),$$

$$\text{(F.51)}$$

$$\frac{1}{m}R_A(t/m, s/m) = \delta(t-s) + o_m(1)\,, \qquad C_A(t/m, s/m) = -\Sigma_C^{\text{lz1}}(t,s) + o_m(1)\,, \quad \text{(F.52)}$$

$$a(t/m)\sqrt{m} = \gamma_0 + o_m(1)\,, \qquad\qquad v(t/m) = \frac{1}{\sqrt{m}}\boldsymbol{v}^{\text{lz1}}(t) + o_m(m^{-1/2})\,, \tag{F.53}$$

with

$$\Sigma_C^{\text{lz1}}(t,s) = \tau^2 + \|\varphi\|^2 - \gamma_0\langle\nabla\hat\varphi(\mathbf{0}), \boldsymbol{v}^{\text{lz1}}(t)\rangle - \gamma_0\langle\nabla\hat\varphi(\mathbf{0}), \boldsymbol{v}^{\text{lz1}}(s)\rangle + \gamma_0^2\left(h(1) + h'(0)C_o^{\text{lz1}}(t,s)\right)\,. \tag{F.54}$$

In particular, Eq. (F.50) implies $\|\boldsymbol{w}_i(0) - \boldsymbol{w}_i(t/m)\| = o_m(1)$: weights of the first layer change by small amount.

The scaling functions defined in Eqs. (F.50)-(F.53) satisfy a set of equations that can be derived directly from the SymmDMFT equations:

$$\begin{aligned}
\partial_t \boldsymbol{v}^{\text{lz1}}(t) &= \alpha\gamma_0\nabla\hat\varphi(\mathbf{0}) - \alpha\gamma_0^2 h'(0)\boldsymbol{v}^{\text{lz1}}(t)\,,\\
\partial_t C_o^{\text{lz1}}(t,t') &= \alpha\gamma_0\langle\nabla\hat\varphi'(\mathbf{0}), \boldsymbol{v}^{\text{lz1}}(t')\rangle - \alpha\gamma_0^2 h'(0)\left(1 + C_o^{\text{lz1}}(t,t')\right)\\
\partial_t R_o^{\text{lz1}}(t,s) &= -\alpha\hat\gamma_0^2 h'(0)\left(1 + R_o^{\text{lz1}}(t,s)\right)\,.
\end{aligned} \tag{F.55}$$

Note that

$$\frac{\mathrm{d}C_o^{\text{lz1}}(t,t)}{\mathrm{d}t} = 2\lim_{t'\to t^-}\partial_t C_o^{\text{lz1}}(t,t')\,. \tag{F.56}$$

The solution of Eqs. (F.55) implies that

$$\begin{aligned}
\boldsymbol{v}_\infty^{\text{lz1}} &:= \lim_{t\to\infty}\boldsymbol{v}^{\text{lz1}}(t) = \frac{\nabla\hat\varphi(\mathbf{0})}{\gamma_0 h'(0)}\,,\\
\lim_{t\to\infty} C_o^{\text{lz1}}(t,t) &= -\left(1 - \|\boldsymbol{v}_\infty^{\text{lz1}}\|^2\right)\,.
\end{aligned} \tag{F.57}$$

Furthermore, on this timescale, the train and test error coincide and are given by

$$\lim_{m\to\infty} e_{\text{tr}}(t/m) = \lim_{m\to\infty} e_{\text{ts}}(t/m) = \frac{1}{2}\Sigma_C^{\text{lz1}}(t,t)\,. \tag{F.58}$$

The corresponding asymptotic value is given by

$$\lim_{t\to\infty}\lim_{m\to\infty} e_{\text{ts}}(t/m) = e_{\text{ts},\infty}^{\text{lz1}} = \frac{1}{2}\left(\tau^2 + \|\varphi\|^2 - \frac{1}{h_s'(0)}\|\nabla\hat\varphi(\mathbf{0})\|^2 + \gamma_0^2\tilde h(1)\right) \tag{F.59}$$

where

$$\tilde h(z) = h(z) - h'(0)\,. \tag{F.60}$$

The interpretation of this dynamical regime is analogous to the one of the same regime in the pure-noise setting, as confirmed by Eq. (F.59) : the network learns the linear component of the data distribution.

In the left panel of Fig. 13 we test the scaling theory in this dynamical regime, as given by Eqs. (F.50) to (F.53). We plot the solution of the SymmDMFT equations, versus $tm$, for increasing values of $m$: the curve collapse well on their conjectured $m \to \infty$ limit.

### F.2.2 Second dynamical regime: $t = \Theta(1)$

We next consider $t = \Theta(1)$. One can show that the SymmDMFT equations are solved, up to higher order terms as $m \to \infty$, by the following ansatz

$$\begin{aligned}
C_d(t,s) &= C_d^{\text{lz2}}(t,s) + o_m(1)\,, & R_d(t,s) &= R_d^{\text{lz2}}(t,s) + o_m(1)\,,\\
C_o(t,s) &= \frac{1}{m}C_o^{\text{lz2}}(t,s) + o_m(m^{-1})\,, & R_o(t,s) &= \frac{1}{m}R_o^{\text{lz2}}(t,s) + o_m(m^{-1})\,,\\
\boldsymbol{v}(t) &= \frac{1}{\sqrt{m}}\boldsymbol{v}_\infty^{\text{lz1}} + o_m(m^{-1/2})\,, & \nu(t) &= \nu^{\text{lz2}}(t) + o_m(1)\,,
\end{aligned} \tag{F.61}$$

with $\gamma(t) = \gamma_0 + o_m(1)$ and

$$C_o^{\text{lz2}}(t,s) = -C_d^{\text{lz2}}(t,s) + \|\boldsymbol{v}_\infty^{\text{lz2}}\|^2\,, \qquad R_o^{\text{lz2}}(t,s) = -R_d^{\text{lz2}}(t,s)\,. \tag{F.62}$$

In other words, on this time scale first layer weights move by order one $\|\boldsymbol{w}_i(t) - \boldsymbol{w}_i(0)\| = \Theta(1)$, but in a linear subspace that is orthogonal to the latent space. Second layer weights do not move appreciably. As a consequence, no additional learning takes place in this regime, but the model begins to overfit the data.

Note that the above scaling form is compatible with the long time limit of the previous dynamical regime.

In order to define the equations for the functions on the right-hand side of Eq. (F.61) we define $R_A^{\text{lz2}}$ and $C_A^{\text{lz2}}$ to be the solution of

$$\delta(t - t') = \int_{t'}^{t} \left[ \delta(t - s) + \Sigma_R^{\text{lz2}}(t, s) \right] R_A^{\text{lz2}}(s, t') \, \mathrm{d}s \,,$$

$$0 = \int_{0}^{t} \left[ \delta(t - s) + \Sigma_R^{\text{lz2}}(t, s) \right] C_A^{\text{lz2}}(s, t') \, \mathrm{d}s + \int_{0}^{t'} \Sigma_C^{\text{lz2}}(t, s) R_A^{\text{lz2}}(t', s) \, \mathrm{d}s \,,$$

(F.63)

where

$$\Sigma_R^{\text{lz2}}(t, s) = \gamma_0^2 \left( h'(C_d^{\text{lz2}}(t, s)) - h'(0) \right) R_d^{\text{lz2}}(t, s) \,,$$

$$\Sigma_C^{\text{lz2}}(t, s) = \tau^2 + \|\varphi\|^2 - 2\gamma_0 \langle \nabla \hat{\varphi}(\boldsymbol{0}), \boldsymbol{v}_\infty^{\text{lz1}} \rangle + \gamma_0^2 \left( h(C_d^{\text{lz2}}(t, s)) + h'(0) C_o^{\text{lz2}}(t, s) \right) \,.$$

(F.64)

Define the following memory kernels

$$M_{R,d}^{\text{lz2}}(t, s) = \alpha \gamma_0^2 \left[ R_A^{\text{lz2}}(t, s) h'(C_d^{\text{lz2}}(t, s)) + C_A^{\text{lz2}}(t, s) h''(C_d^{\text{lz2}}(t, s)) R_d^{\text{lz2}}(t, s) \right] \,,$$

$$M_{R,o}^{\text{lz2}}(t, s) = \alpha \gamma_0^2 h'(0) R_A^{\text{lz2}}(t, s) \,,$$

$$M_{C,d}^{\text{lz2}}(t, s) = \alpha \gamma_0^2 h'(C_d^{\text{lz2}}(t, s)) C_A^{\text{lz2}}(t, s) \,,$$

$$M_{C,o}^{\text{lz2}}(t, s) = \alpha \gamma_0^2 h'(0) C_A^{\text{lz2}}(t, s) \,.$$

(F.65)

Substituting the ansatz (F.61) into the SymmDMFT equations, and using Eqs. (F.62), we obtain the following equations for $C_d^{\text{lz2}}(t, t')$, $R_d^{\text{lz2}}(t, t')$, $\nu^{\text{lz2}}(t)$

$$\partial_t C_d^{\text{lz2}}(t, t') = -\nu^{\text{lz2}}(t) C_d^{\text{lz2}}(t, t') + \alpha \gamma_0 \langle \nabla \hat{\varphi}'(\boldsymbol{0}), \boldsymbol{v}_\infty^{\text{lz1}} \rangle \int_0^t R_A^{\text{lz2}}(t, s) \mathrm{d}s$$

$$- \int_0^t \mathrm{d}s \left[ M_{R,d}^{\text{lz2}}(t, s) C_d^{\text{lz2}}(t', s) + M_{R,o}^{\text{lz2}}(t, s) C_o^{\text{lz2}}(t', s) \right] \mathrm{d}s$$

(F.66)

$$- \int_0^{t'} \left[ M_{C,d}^{\text{lz2}}(t, s) R_d^{\text{lz2}}(t', s) + M_{C,o}^{\text{lz2}}(t, s) R_o^{\text{lz2}}(t', s) \right] \mathrm{d}s \,,$$

$$\partial_t R_d^{\text{lz2}}(t, t') = -\nu^{\text{lz2}}(t) R_d^{\text{lz2}}(t, t') + \delta(t - t')$$

(F.67)

$$- \int_{t'}^t \left[ M_{R,d}^{\text{lz2}}(t, s) R_d^{\text{lz2}}(s, t') + M_{R,o}^{\text{lz2}}(t, s) R_o^{\text{lz2}}(s, t') \right] \mathrm{d}s \,,$$

$$\nu^{\text{lz2}}(t) = \alpha \gamma_0 \langle \nabla \hat{\varphi}'(\boldsymbol{0}), \boldsymbol{v}_\infty^{\text{lz1}} \rangle \int_0^t R_A^{\text{lz2}}(t, s) \, \mathrm{d}s - \int_0^t \left[ M_{R,d}^{\text{lz2}}(t, s) C_d^{\text{lz2}}(t, s) + M_{R,o}^{\text{lz2}}(t, s) C_o^{\text{lz2}}(t, s) \right] \mathrm{d}s$$

$$- \int_0^t \left[ M_{C,d}^{\text{lz2}}(t, s) R_d^{\text{lz2}}(t, s) + M_{C,o}^{\text{lz2}}(t, s) R_o^{\text{lz2}}(t, s) \right] \mathrm{d}s \,.$$

(F.68)

Finally, the train and test errors converge to well defined limits for $t$ fixed and $m \to \infty$:

$$e_{\text{tr}}(t, \gamma_0) = e_{\text{tr}}^{\text{lz2}}(t, \gamma_0) + o_m(1) \,, \qquad e_{\text{ts}}(t, \gamma_0) = e_{\text{ts}}^{\text{lz2}}(t, \gamma_0) + o_m(1) \,.,$$

(F.69)

where

$$e_{\text{tr}}^{\text{lz2}}(t, \gamma_0) = -\frac{1}{2} C_A^{\text{lz2}}(t, t) \,, \quad e_{\text{ts}}^{\text{lz2}}(t, \gamma_0) = \frac{1}{2} \Sigma_C^{\text{lz2}}(t, t) \,.$$

(F.70)

Note that, using Eqs. (F.62), (F.64), and the fact that $C_d^{\text{lz2}}(t, t) = 1$ (because of the unit norm constraint on the first layer weights), we get

$$e_{\text{ts}}^{\text{lz2}}(t, \gamma_0) = \frac{1}{2} \left\{ \tau^2 + \|\varphi\|^2 - 2\gamma_0 \langle \nabla \hat{\varphi}(\boldsymbol{0}), \boldsymbol{v}_\infty^{\text{lz1}} \rangle + \gamma_0^2 \left( h(1) - h'(0) + h'(0) \|\boldsymbol{v}_\infty^{\text{lz1}}\|^2 \right) \right\} \,.$$

(F.71)

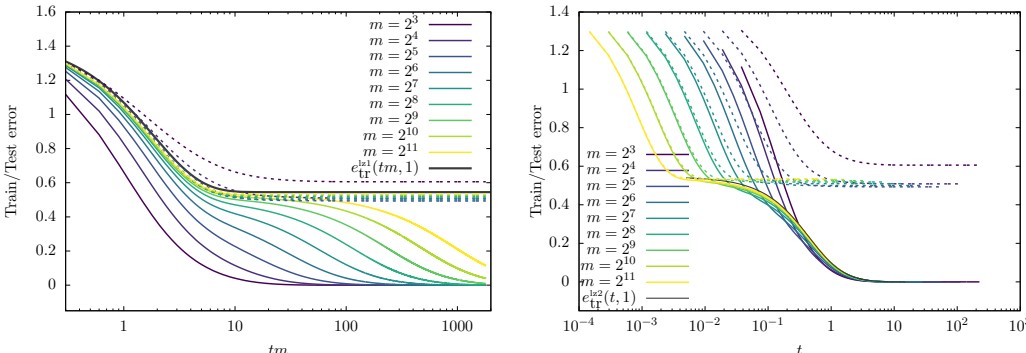

Figure 14: SymmDMFT **predictions and large network scaling for lazy training in a single index model: train and test error.** Left frame: train and test error on the time scale $t = \Theta(1/m)$ for several values of $m$, together with the asymptotic prediction as $m \to \infty$ on this time scale $e_{\text{tr}}^{\text{lz1}}(\hat{t}, \gamma_0) = e_{\text{ts}}^{\text{lz1}}(\hat{t}, \gamma_0)$. Right: train and test error on the time scale $t = \Theta(1)$ for several values of $m$, together with the asymptotic prediction as $m \to \infty$ on this time scale $e_{\text{tr}}^{\text{lz2}}(t, \gamma_0)$. Here $\gamma_0 = 1$, $h(z) = \hat{\varphi}(z) = (9/10)z + z^2/2$, $\tau = 0.3$, $\alpha = 0.3$.

Using Eq. (F.57), we obtain that the asymptotic test error in this dynamical regime is constant and equal to the test error achieved at the end of the previous regime, namely $e_{\text{ts}}^{\text{lz2}}(t, \gamma_0) = e_{\text{ts},\infty}^{\text{lz1}}$, cf. Eq. (F.59). As anticipated, no learning takes place on this timescale.

The predictions of Eqs. (F.61) are tested in the right panel of Fig. 13. We plot the correlation function $C_d(t, 0)$ for several values of $m$, as obtained by solving the SymmDMFT equations. We compare these results with the $m \to \infty$ prediction $C_d^{\text{lz2}}(t, 0)$ obtained by solving Eqs. (F.66) to (F.68). We observe collapse of finite $m$ curves on the large $m$ asymptotics supporting our conclusions.

In Fig. 14 we plot the behavior of the train and test error both on timescales $t = \Theta(1/m)$ (left frame, plotting $e_{\text{tr}}(t, \gamma_0)$, $e_{\text{ts}}(t, \gamma_0)$ versus $tm$) and $t = \Theta(1)$ (right frame, plotting $e_{\text{tr}}(t, \gamma_0)$, $e_{\text{ts}}(t, \gamma_0)$ versus $tm$). We the solutions of SymmDMFT equations at increasing values of $m$ with the theory scaling theory presented in the previous section (for $t = \Theta(1/m)$, left frame) and in this section (for $t = \Theta(1)$, right frame). As anticipated, we observe the following:

- On the time scale $t = \Theta(1/m)$ (left panel), test and train error collapse (as $m \to \infty$) on a common limiting curve $e_{\text{tr}}^{\text{lz1}}(\hat{t}, \gamma_0) = e_{\text{ts}}^{\text{lz1}}(\hat{t}, \gamma_0)$ which converges, for large $\hat{t}$, to the positive limiting value $e_{\text{ts},\infty}^{\text{lz1}}$ characterized in the previous section.

- On the time scale $t = \Theta(1)$ (right panel), test and train error collapse (as $m \to \infty$) on two distinct limiting curves. The first one is constant and equal to $e_{\text{ts},\infty}^{\text{lz1}}$. The second one decreases from $e_{\text{ts},\infty}^{\text{lz1}}$ to 0 and is predicted by the asymptotic theory in this section, cf. Eq. (F.70).

Note that, in the example of Fig. 14, the initialization $\gamma_0$ is sufficiently large that the train error decreases to zero on the time scale $\Theta(1)$, namely $\gamma_0 > \gamma_{\text{GF}}^*(\alpha, \varphi, \tau)$, for a suitable threshold $\gamma_{\text{GF}}^*(\alpha, \varphi, \tau)$. As we will see in the next section, a third dynamical regime emerges when $\gamma_0 < \gamma_{\text{GF}}^*(\alpha, \varphi, \tau)$.

### F.2.3  The algorithmic interpolation threshold

The asymptotic theory within the second dynamical regime, described in Section F.2.2, turns out to be equivalent to the one in the pure-noise model, Section F.1.2, up to a change of variables. Namely, defining

$$\tilde{C}_o(t, s) = C_o^{\text{lz2}}(t, s) + \|\boldsymbol{v}_\infty^{\text{lz1}}\|^2, \tag{F.72}$$

with initial condition $\tilde{C}_o(0, 0) = -1$, reduce the equations of Section F.2.2 to the ones of Section F.1.2 with noise level $\tau$ replaced by

$$\tau'^2 = \tau^2 + \|\varphi\|^2 - \frac{\|\nabla \hat{\varphi}(\boldsymbol{0})\|^2}{h'(0)}. \tag{F.73}$$

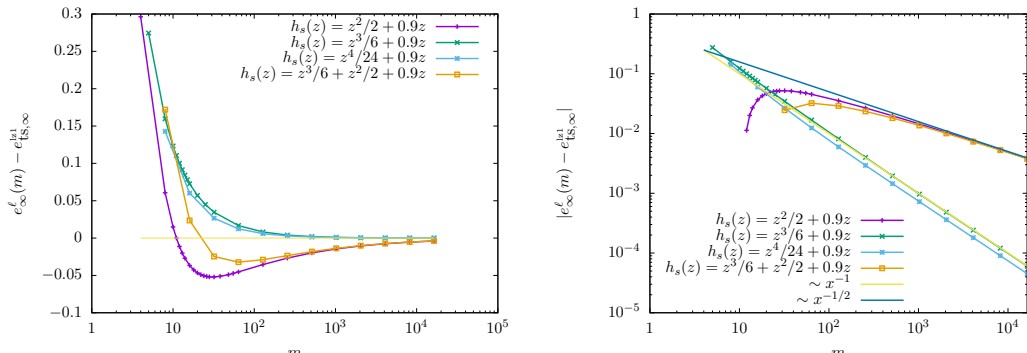

Figure 15: **The asymptotic behavior of the test error as a function of** $m$ for different $h(z) = \hat{\varphi}(z)$. We observe that soon as $h(z)$ contains a $z^2$ term, the NTK limit for $m \to \infty$ is approached from below (left panel). Furthermore the speed of the convergence to the limiting value depends crucially on whether a $z^2$ monomial is present in the Taylor expansion of $h(z)$ (right panel). The data has been produced with $\alpha = 0.3$ and $\tau = 0.6$.

The interpretation of this reduction is simple. On the time scale $t = \Theta(1)$, the first layer weights move orthogonally to the latent subspace spanned by $\boldsymbol{U}$. Hence, the dynamics on this timescale is not affected by the signal and only attempts to fit the labels noise. The noise is inflated as per Eq. (F.73), because the network is not able to fit beyond the linear part of the target distribution.

As a corollary of the above equivalence, the interpolation threshold of the $k$-index model coincides with with the interpolation threshold on pure noise data with noise level given by Eq. (F.73). Using the extended notation $\gamma_{\text{GF}}^*(\alpha, \varphi, \tau)$ to indicate the dependence on the underlying data distribution (which is parametrized by $\varphi, \tau$), we can write the stated relation as

$$\gamma_{\text{GF}}^*(\alpha, \varphi, \tau) = \left( \tau^2 + \|\varphi\|^2 - \frac{\|\nabla \hat{\varphi}(\mathbf{0})\|^2}{h'(0)} \right)^{1/2} \gamma_{\text{GF}}^*(\alpha, 0, 1) . \tag{F.74}$$

(Here we used the invariance under rescaling in the pure noise model, which implies $\gamma_{\text{GF}}^*(\alpha, 0, \tau^2) = \tau \gamma_{\text{GF}}^*(\alpha, 0, 1)$.)

### F.2.4 Dependence on $m$

Within NTK theory, it is normally assumed that optimal models are achieved at very large network sizes $m \to \infty$. Empirical results contradicting this expectation have been put forward in [54], but no theoretical analysis was provided either in [54] or in subsequent work. We can use the SymmDMFT theory to fill this gap and study the dependence of test error on the number of neurons $m$ under lazy initialization. We choose $\gamma_0 > \gamma_{\text{GF}}^*(\alpha, \varphi, \tau)$, and therefore vanishing training error is reached during the second dynamical regime, i.e. for $t = \Theta(1)$: this is therefore the last dynamical regime. Throughout this regime, we have $\gamma(t) = \gamma_0 + o_m(1)$.

Recalling that $e_{\text{ts}}(t, \gamma_0, m, \alpha)$ is the test error at time $t$ in this setting, as predicted by SymmDMFT we consider the limit

$$e_\infty^\ell(\gamma_0, m, \alpha) = \lim_{t \to \infty} e_{\text{ts}}(t, \gamma_0, m, \alpha) . \tag{F.75}$$

We note that, for $\gamma_0 > \gamma_{\text{GF}}^*(\alpha, \varphi, \tau)$, we expect

$$\lim_{m \to \infty} e_\infty^\ell(\gamma_0, m, \alpha) = e_{\text{ts}, \infty}^{\text{lzl}}, \tag{F.76}$$

to be given by Eq. (F.59).

In Fig. 15 we plot the SymmDMFT prediction for $e_\infty^\ell(\gamma_0, m, \alpha)$ as a function of $m$ for several choices of $h$ (we use $h = \hat{\varphi}$ here). The limit $m \to \infty$ of these curves matches $e_{\text{ts}, \infty}^{\text{lzl}}$ as expected. However we empirically observe that $e_\infty^\ell(\gamma_0, m, \alpha)$ approaches $e_{\text{ts}, \infty}^{\text{lzl}}$ in two qualitatively different ways:

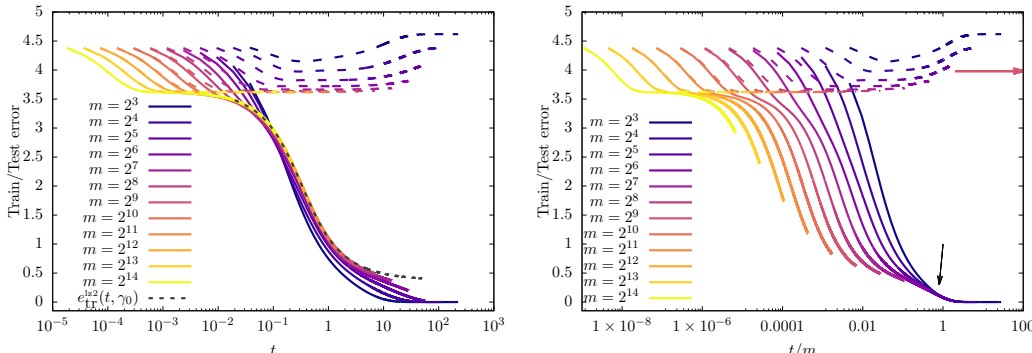

Figure 16: **Train and test error on different timescales when training on single index data and lazy initialization.** Train error (solid curves) and test error (dashed curves) for a model trained on a single index data with $h(z) = (9/10)z + z^2/2 = \hat{\varphi}(z)$. The noise level is $\tau = 2.5$ and initialization $a(0) = \gamma_0\sqrt{m}$, $\gamma_0 < \gamma_{\mathrm{GF}}^*(\alpha, \varphi, \tau)$. Left panel: timescales of order one. The grey dashed line corresponds to the scaling solution for $m \to \infty$ when the second layer does not evolve with GF. Right panel: same data plotted versus $t/m$, to explore timescales of order $m$. The arrows show scaling appearing and curves collapsing on a master curve.

- In the cases we consider that have $h''(0) \neq 0$, $e_{\mathrm{ts},\infty}^{\mathrm{lz1}}$ is approached from below as $m \to \infty$, and $e_\infty^\ell(\gamma_0, m, \alpha)$ is non-monotone. We also observe that, for the values of $m$ we consider, the approach to the asymptotic value is compatible with a rate $m^{-1/2}$: $e_\infty^\ell(\gamma_0, m, \alpha) = e_{\mathrm{ts},\infty}^{\mathrm{lz1}} - \Theta(m^{-1/2})$.

- In the cases we consider that have $h''(0) \neq 0$, then $e_{\mathrm{ts},\infty}^{\mathrm{lz1}}$ is approached from above as $m \to \infty$, and $e_\infty^\ell(\gamma_0, m, \alpha)$ is typically monotone. In this case the approach to the limiting behavior is compatible with a rate $m^{-1}$: $e_\infty^\ell(\gamma_0, m, \alpha) = e_{\mathrm{ts},\infty}^{\mathrm{lz1}} + \Theta(m^{-1})$.

The first scenario is the generic one, and similar to what is observed in [54] for actual neural networks. An intuitive explanation is that –at finite $m$– the projection of neurons onto the latent space $\|v_\infty^{\mathrm{lz1}}\| = \Theta(1/\sqrt{m})$ is sufficient for the network to partially learn the quadratic component of the target function. In order to establish on more solid grounds these empirical observations one should study the $1/m$ corrections to the scaling theory developed here. This is left for future work.

### F.2.5   Third dynamical regime: $t = \Theta(m)$

As for the pure noise case, beyond the time scale $t = \Theta(1)$, we distinguish two situations. If $\gamma_0 > \gamma_{\mathrm{GF}}^*(\alpha, \varphi, \tau)$, then vanishing training error is reached within the second dynamical regime $t = \Theta(1)$. If $\gamma_0 < \gamma_{\mathrm{GF}}^*(\alpha, \varphi, \tau)$, GF dynamics develops an additional regime for $t = \Theta(m)$. In this section, we study this third regime.

In Figure 16, we plot the SymmDMFT predictions for train and test errors as a function of time for several values of $m$, for a setting with $\gamma_0 < \gamma_{\mathrm{GF}}^*(\alpha, \varphi, \tau)$. In particular, in Fig.16-left we plot train and test error as a function of $t$. The curves for the train error for increasing value of $m$ collapse on limit curve given by $e_{\mathrm{tr}}^{\mathrm{lz2}}(t, \gamma_0)$ characterized in Section F.2.2. In other words, the dynamics on this timescales follows the scaling theory of Section F.2.2. However in this case $\gamma_0 < \gamma_{\mathrm{GF}}^*(\alpha, \varphi, \tau)$, whence by definition $e_{\mathrm{tr},\infty}^{\mathrm{lz2}} > 0$. This correspond to the limit curve in Fig. 16-left having a strictly positive asymptote.

Figure 16-right shows train and test error plotted against $t/m$. We observe that curves training error curves collapse on a common limit, that decreases from $e_{\mathrm{tr},\infty}^{\mathrm{lz2}}$ to 0, while test error curves increase above the plateau $e_{\mathrm{ts},\infty}^{\mathrm{lz1}}$. This suggests the following limit behavior

$$\lim_{m\to\infty} e_{\mathrm{tr}}(m\hat{t}, \gamma_0, m) = e_{\mathrm{tr}}^{\mathrm{lz3}}(\hat{t}, \gamma_0)$$
$$\lim_{m\to\infty} e_{\mathrm{ts}}(m\hat{t}, \gamma_0, m) = e_{\mathrm{ts}}^{\mathrm{lz3}}(\hat{t}, \gamma_0)\,.$$

(F.77)

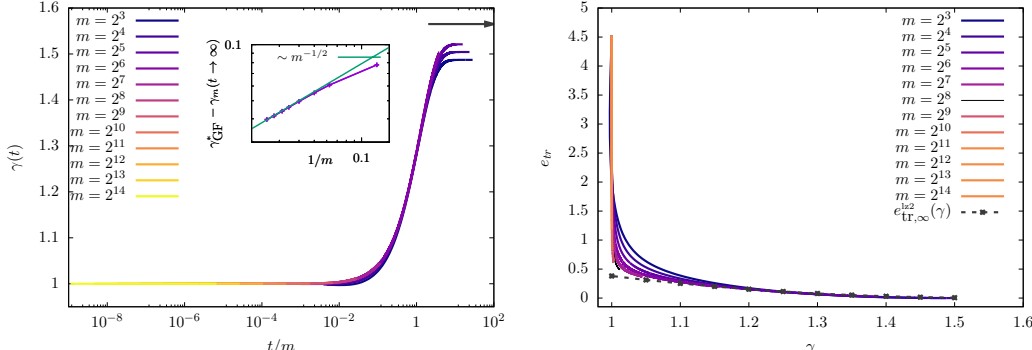

Figure 17: **Training a two layer network in the same setting of Figure 16.** Left panel: second layer weights on the timescale of order $m$. The black arrow corresponds to the interpolation threshold for a model, $\gamma^*_{GF}(\alpha, \tau)$ obtained by fitting the relaxation time as a function of the weights of an lazy initialized model for $\gamma_0 > \gamma^*_{GF}(\alpha, \tau)$. The second layer weights, at finite $m$ develop a plateau at long time. In the inset we show the approach of this plateaus to the limiting value given by $\gamma^*_{\text{GF}}(\alpha, \varphi, \tau)$. Right panel: parametric plot of the train error as a function of the scaled weights of the second layer. The dashed gray dashed line corresponds to the extrapolated train error for an network with second layer weights fixed to the corresponding value in the $m \to \infty$ (as extracted from the numerical integration of the scaling theory).

In order to further explore the GF dynamics in this regime, in Fig. 17-left we plot the evolution of the second layer rescaled weights against $t/m$. The curves for increasing values of $m$ collapse on a master curve, suggesting the existence of a limit

$$\lim_{m \to \infty} \gamma(m\hat{t}, \gamma_0) = \gamma^{\text{lz3}}(\hat{t}, \gamma_0) . \tag{F.78}$$

The limit curve $\gamma^{\text{lz3}}(\hat{t}, \gamma_0)$ increases from $\gamma_0$ to a limit value:

$$\lim_{\hat{t} \to \infty} \gamma^{\text{lz3}}(\hat{t}, \gamma_0) = \gamma^{\text{lz3}}_{\infty}(\gamma_0). \tag{F.79}$$

As in Section F.2.5 we consider the inverse function of $t \mapsto \gamma^{\text{lz3}}(t, \gamma_0)$, denoted by $\gamma \mapsto \tilde{\gamma}^{-1}(\gamma, \gamma_0)$. In Fig. 17-right we plot the train error as a function of the second layer weights $\gamma(t)$. Again, for increasing values of $m$ the curves collapse on a master curve which is given by

$$\varepsilon(\gamma, \gamma_0) = e^{\text{lz3}}_{\text{tr}}(\tilde{\gamma}^{-1}(\gamma, \gamma_0), \gamma_0) \tag{F.80}$$

We then also plot in Fig.17-right the asymptotic value of the train error for a network initialized with second layer weights blocked at an initialization scale $\gamma$, call it $e^{\text{lz2}}_{\text{tr},\infty}(\gamma)$.

The curves $\varepsilon(\gamma, \gamma_0)$ appear to have a vertical segment (corresponding to $t = o(m)$) in which the training error decreases, while $\gamma(t) = \gamma_0 + o_m(1)$ is nearly unchanged, and a continuously decreasing segment in which $\gamma(t)$ increases while $e_{\text{tr}}(t, \gamma_0)$ decreases to 0 (corresponding to $t = \Theta(m)$). In the second phase, the curves appear to converge to $e^{\text{lz2}}_{\text{tr},\infty}(\gamma)$ as $m \to \infty$. This suggests

$$\varepsilon(\gamma, \gamma_0) = e^{\text{lz2}}_{\text{tr},\infty}(\gamma) \quad \forall \gamma \geq \gamma_0 . \tag{F.81}$$

In other words the dynamics on timescales of order $m$ is adiabatic also in the multi index case. For a small change of the second layer weights on a scale of order $\sqrt{m}$, the train error relaxes to its asymptotic value on timescales of order one. This graph suggests that the limit value of $\gamma(t)$ coincides with the critical value for interpolation. Namely recalling the definition (F.79) for the asymptotic value of $\gamma(t)$, we have

$$\gamma^{\text{lz3}}_{\infty}(\gamma_0) = \gamma^*_{\text{GF}}(\alpha, \varphi, \tau) \tag{F.82}$$

where the interpolation threshold in the multi-index model $\gamma^*_{\text{GF}}(\alpha, \varphi, \tau)$ is related to the interpolation threshold in the pure noise model via Eq. (F.74).

# G    Dynamical regimes: Mean field initialization

In this section we assume the initialization of the weights of the second layer is kept of order one. To be definite, we set $a(0) = a_0$, independent of $m$. This corresponds to the mean field initialization studied in [38, 14, 45].

Specializing to the data distribution considered here, earlier work characterized the dynamics up to time $T$, under a few settings (which prove equivalent in this regime):

- One-pass SGD, with stepsize $\varepsilon \ll 1/d$ and therefore time horizons such that $T \ll d/n$ (the latter inequality follows from $T \le n\varepsilon$ for one-pass SGD). In this case, the dynamics is characterized by a set of ODEs for for the projections of the weights on the latent space and inner products between weights.

- Gradient flow in the population risk, which admits the same characterization and corresponds to the limit $n \to \infty$ of the above.

- The limit of the above regimes for large width $m \to \infty$. This is characterized by a partial differential equation for the distribution of projections of first layer weights onto the latent space, provided $T \le c_0 \log m$, for $c_0$ a sufficiently small constant.

We refer to [5, 21, 1, 6, 2, 10] for a few pointers to this literature. In all of these settings, the train error remains close to the test error. In contrast, the analysis presented here allow us to explore the overfitting regime.

Section G.1, we will focus on a pure noise data distribution, while Section G.2, considers a multi-index model. As in the case of lazy initializations, we consider first the limit $n, d \to \infty$ at $n/md = \alpha$ and $m$ fixed (hence characterized by SymmDMFT) and subsequently study dynamical regimes emerging as $m \to \infty$ at $n/md = \alpha$ fixed.

## G.1    Pure noise model

Under the pure noise model, we have $\varphi = \hat\varphi = 0$. We identify three distinct dynamical regimes:

- $t = O(1)$: $a(t) = a_0 + o_m(1)$, $e_{\mathrm{tr}}(t) = \tau^2/2 + o_m(1)$, and $\|\boldsymbol{w}_i(t) - \boldsymbol{w}_i(0)\| = o_m(1)$. In words, the weights change minimally and the train error remains close to the one of the null network $f(\boldsymbol{x}; \boldsymbol{\theta}) \approx 0$ (Section G.1.1).

- $t = \Theta(\sqrt{m})$: $a(t) = \Theta(1)$, $e_{\mathrm{tr}}(t) = \tau^2/2 + o_m(1)$, and $\|\boldsymbol{w}_i(t) - \boldsymbol{w}_i(0)\| = \Theta(1)$. Namely, weights change but the train error does not change significantly. (Section G.1.2).

- $t = \Theta(m)$. In this regime $a(t) = \sqrt{m}\gamma(t/m) + o_m(1)$, and therefore the network complexity becomes large enough for it to fit the noise. The dynamics on this timescale is closely related to the one under lazy initialization, studied in Section F.1.4. In particular, $\gamma(\hat{t})$ converge to the interpolation threshold $\gamma^*_{\mathrm{GF}}(\alpha, \tau)$ if $\hat{t} \to \infty$ (after $m \to \infty$). (Section G.1.3).

### G.1.1    First dynamical regime: $t = O(1)$

In this dynamical regime, the SymmDMFT equations are solved by the following scaling ansatz

$$C_d(t, s) = 1 + o_m(1) \qquad\qquad R_d(t, s) = \vartheta(t - s) + o_m(1), \qquad (\text{G.1})$$
$$mC_o(t, s) = C_o^{\mathrm{mfl}}(t, s) + o_m(1) \qquad mR_o(t, s) = R_o^{\mathrm{mfl}}(t, s) + o_m(1), \qquad (\text{G.2})$$
$$a(t) = a_0 + o_m(1) \qquad\qquad \nu(t) = o_m(1). \qquad (\text{G.3})$$

Furthermore we have

$$R_A^{\mathrm{mfl}}(t, s) = \delta(t - s) + o_m(1) \quad C_A^{\mathrm{mfl}}(t, s) = -\tau^2 + o_m(1), \qquad (\text{G.4})$$

Plugging the scaling ansatz in the SymmDMFT, we obtain equations determining the scaling functions $C_o^{\mathrm{mfl}}$, $R_o^{\mathrm{mfl}}$. Defining

$$\rho_0 := \alpha a_0^2 h'(0) \qquad (\text{G.5})$$

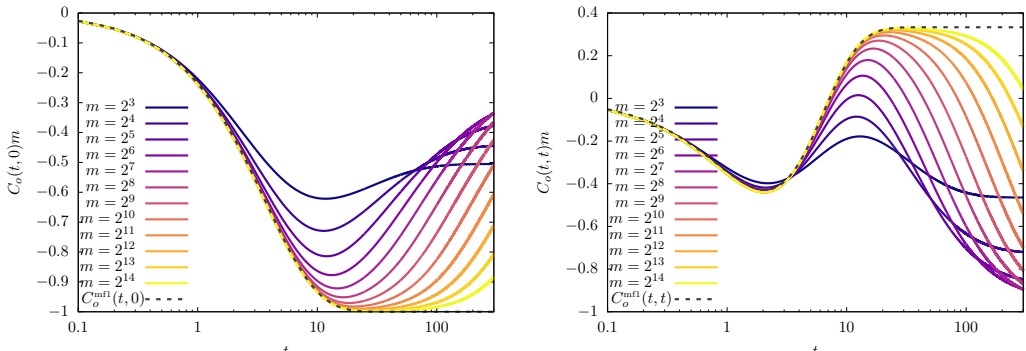

Figure 18: **Training on pure noise data under mean-field initialization:** $t = \Theta(1)$ **regime.** We plot $C_o(t,0)$ and $C_o(t,t)$ as given by solving the SymmDMFT equations for different values of $m$ and compare them with the asymptotic solution of Section G.1.1. Here we use $\tau = 0.6$, $\alpha = 0.3$ and $h(z) = (9/10)z + z^3/6$. Note that the vertical axis is multiplied by a factor $m$, in agreement with the prediction of Eq. (G.2).

we have

$$R_o^{\text{mf1}}(t,s) = \left[ e^{-\rho_0(t-s)} - 1 \right] \vartheta(t-s),$$

$$C_o^{\text{mf1}}(t,t') = \left[ \left[ \frac{2\tau^2}{\rho_0} - \frac{1}{\rho_0} \left( \tau^2 - \rho_0 \right) \right] e^{-2\rho_0 t'} - \frac{\tau^2}{\rho_0} e^{-\rho_0 t'} \right] e^{-\rho_0(t-t')}$$

$$+ \frac{\tau^2 - \rho_0}{\rho_0} - \frac{\tau^2}{\rho_0} e^{-\rho_0 t'}. \tag{G.6}$$

In particular

$$\lim_{t \to \infty} C_o^{\text{mf1}}(t,t) = \frac{\tau^2 - \rho_0}{\rho_0},$$

$$\lim_{t \to \infty} C_o^{\text{mf1}}(t,t') = \frac{\tau^2 - \rho_0}{\rho_0} - \frac{\tau^2}{\rho_0} e^{-\rho_0 t'}, \tag{G.7}$$

$$\lim_{t \to \infty, t' \to \infty, t-t' \geq 0} C_o^{\text{mf1}}(t,t') = \frac{\tau^2 - \rho_0}{\rho_0}.$$

The equations (G.4) imply that the train error is constant in this regime and equal to

$$e_{\text{tr}}(t) = \frac{\tau^2}{2} + o_m(1). \tag{G.8}$$

In other words, in this regime both first and second layer weights change minimally and the resulting error remains close to the one to the null function $f(\boldsymbol{x}; \boldsymbol{\theta}) \approx 0$. We will see that this regime is significantly more interesting for the case of data with a signal, see Section G.2. We note in passing that the limit value $\langle \boldsymbol{w}_j, \boldsymbol{w}_j \rangle \approx \frac{\tau^2 - \rho_0}{m\rho_0}$ for $i \neq j$ corresponds to minimizing the empirical risk under the linear approximation in which $\sigma(z)$ is replaced by $\sqrt{h'(0)}z$.

The above predictions are tested in Fig. 18 where we plot $C_o(t,t)$ and $C_o(t,0)$ for different values of $m$ and check their approach to the scaling functions $C_o^{\text{mf1}}(t,0)$ and $C_o^{\text{mf1}}(t,t)$.

### G.1.2 Second dynamical regime: $t = \Theta(\sqrt{m})$

We now consider the case in which time scales as $\sqrt{m}$. The following asymptotic forms can be checked to solve the SymmDMFT equations, up to higher order terms, for suitable choices of the scaling functions on the right-hand side:

$$C_d(t\sqrt{m}, s\sqrt{m}) = C_d^{\text{mf2}}(t,s) + o_m(1) \qquad\qquad R_d(t\sqrt{m}, s\sqrt{m}) = R_d^{\text{mf2}}(t,s) + o_m(1),$$

$$\tag{G.9}$$

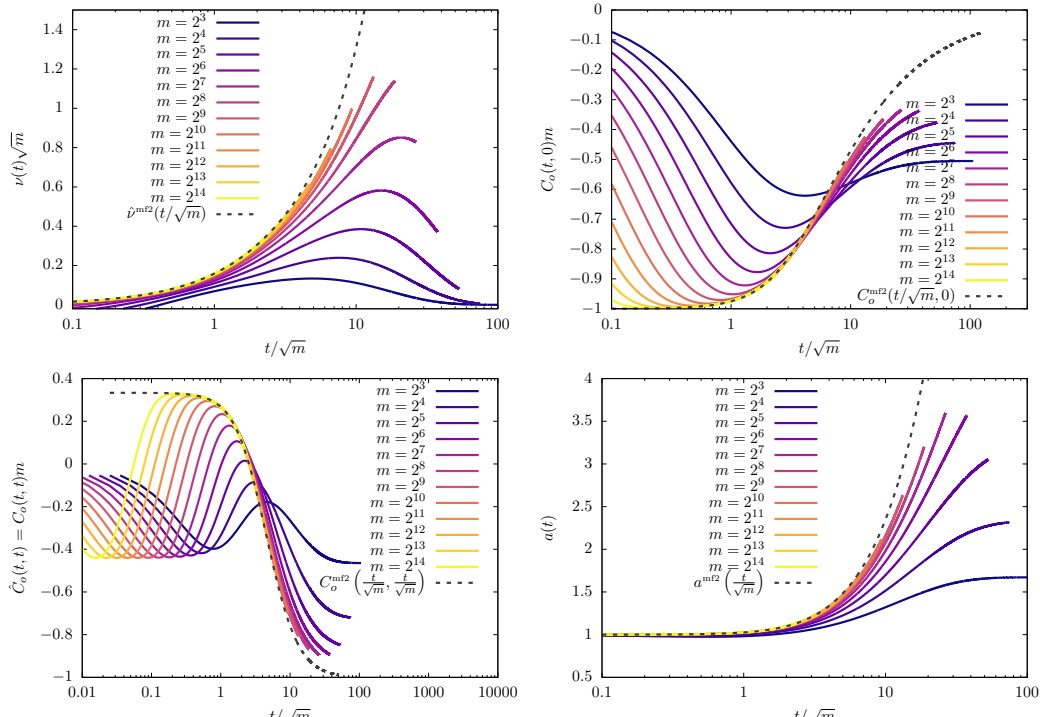

Figure 19: **Training on pure noise data under mean-field initialization:** $t = \Theta(\sqrt{m})$ **regime**, under the same setting as in Fig. 18. We plot the solutions of the SymmDMFT equations for several values of $m$ as a function of $t/\sqrt{m}$. We compare these to the $m \to \infty$ scaling theory of Section G.1.2, i.e. to numerical solutions of Eqs. (G.14) to (G.17).

$$C_o(t\sqrt{m}, s\sqrt{m}) = \frac{1}{m}C_o^{\text{mf2}}(t,s) + o_m(m^{-1}) \quad R_o(t\sqrt{m}, s\sqrt{m}) = \frac{1}{m}R_o^{\text{mf2}}(t,s) + o_m(m^{-1}),$$
(G.10)

$$\sqrt{m}R_A(t\sqrt{m}, s\sqrt{m}) = \delta(t-s) + o_m(1) \qquad C_A(t\sqrt{m}, s\sqrt{m}) = -\tau^2 + o_m(1), \quad \text{(G.11)}$$

$$\sqrt{m}\nu(t\sqrt{m}) = \nu^{\text{mf2}}(t) + o_m(1) \qquad\qquad a(t\sqrt{m}) = a^{\text{mf2}}(t) + o_m(1) . \text{(G.12)}$$

Plugging this scaling ansatz into the SymmDMFT equations we get the constraints

$$R_o^{\text{mf2}}(t,s) = -R_d^{\text{mf2}}(t,s),$$

$$C_o^{\text{mf2}}(t,s) = -C_d^{\text{mf2}}(t,s) + \frac{\tau^2}{\alpha h'(0)(a^{\text{mf2}}(t))^2}.$$
(G.13)

We also obtain that the following equations must be satisfied by $C_d^{\text{mf2}}(t,t')$, $R_d^{\text{mf2}}(t,t')$, $a^{\text{mf2}}(t)$, $\nu^{\text{mf2}}(t)$,

$$\partial_t C_d^{\text{mf2}}(t,t') = -\nu^{\text{mf2}}(t)C_d^{\text{mf2}}(t,t') + \alpha\tau^2 a^{\text{mf2}}(t)\int_0^t a^{\text{mf2}}(s)h''(C_d^{\text{mf2}}(t,s))R_d^{\text{mf2}}(t,s)C_d^{\text{mf2}}(t',s)\,\mathrm{d}s$$
(G.14)

$$+ \alpha\tau^2 a^{\text{mf2}}(t)\int_0^{t'} a^{\text{mf2}}(s)\left[h'(C_d^{\text{mf2}}(t,s)) - h'(0)\right]R_d^{\text{mf2}}(t',s)\,\mathrm{d}s,$$

$$\partial_t R_d^{\text{mf2}}(t,t') = \delta(t-t') - \nu^{\text{mf2}}(t)R_d^{\text{mf2}}(t,t')$$
(G.15)

$$+ \alpha\tau^2 a^{\text{mf2}}(t)\int_{t'}^t a^{\text{mf2}}(s)h''(C_d^{\text{mf2}}(t,s))R_d^{\text{mf2}}(t,s)R_d^{\text{mf2}}(s,t')\,\mathrm{d}s,$$

$$\nu^{\text{mf2}}(t) = \alpha\tau^2 a^{\text{mf2}}(t)\int_0^t \left[a^{\text{mf2}}(s)h''(C_d^{\text{mf2}}(t,s))R_d^{\text{mf2}}(t,s)C_d^{\text{mf2}}(t,s)\right]\,\mathrm{d}s$$
(G.16)

$$+ \alpha \tau^2 a^{\text{mf2}}(t) \int_0^t a^{\text{mf2}}(s) \left[ h'(C_d^{\text{mf2}}(t,s)) - h'(0) \right] R_d^{\text{mf2}}(t,s) \, \mathrm{d}s \,,$$

$$\frac{\mathrm{d}a^{\text{mf2}}(t)}{\mathrm{d}t} = \alpha \tau^2 \int_0^t a^{\text{mf2}}(s) \left[ h'(C_d^{\text{mf2}}(t,s)) - h'(0) \right] R_d^{\text{mf2}}(t,s) \, \mathrm{d}s \,, \tag{G.17}$$

with initial conditions given by

$$C_d^{\text{mf2}}(0,0) = 1 \qquad\qquad R_d^{\text{mf2}}(0+,0) = 1 \qquad\qquad a^{\text{mf2}}(0) = a_0 \,. \tag{G.18}$$

We test these predictions in Fig. 19. We plot several quantities in the solution of the Symm-mDMFT equations for increasing values of $m$ and compare them with the solution of the asymptotic equations (G.14) to (G.17). We observe convergence to the predicted asymptotic behavior.

Equations (G.14) to (G.17) can be further simplified. The right-hand side of Eq. (G.17) is a positive. Therefore $a^{\text{mf2}}(t)$ is a monotone increasing function. Define the time change

$$\tilde{t}(t) = \tau \sqrt{\alpha} \int_0^t a^{\text{mf2}}(s) \, \mathrm{d}s \,, \tag{G.19}$$

and the corresponding time-changed scaling functions

$$\tilde{\nu}(\tilde{t}(t)) = \frac{\nu^{\text{mf2}}(t)}{a^{\text{mf2}}(t) \tau \sqrt{\alpha}} \,,$$
$$\tilde{C}_d^{\text{mf}}(\tilde{t}(t), \tilde{t}(t')) = C_d^{\text{mf2}}(t,t') \,, \tag{G.20}$$
$$\tilde{R}_d^{\text{mf}}(\tilde{t}(t), \tilde{t}(t')) = R_d^{\text{mf2}}(t,t') \,.$$

Equations (G.14) to (G.17) imply that these time-changed function functions satisfy

$$\partial_t \tilde{C}_d^{\text{mf}}(t,t') = -\tilde{\nu}^{\text{mf}}(t) \tilde{C}_d^{\text{mf}}(t,t') + \int_0^t \tilde{h}''(\tilde{C}_d^{\text{mf}}(t,s)) \tilde{R}_d^{\text{mf}}(t,s) \tilde{C}_d^{\text{mf}}(t',s) \, \mathrm{d}s \tag{G.21}$$

$$+ \int_0^{t'} \tilde{h}'(\tilde{C}_d^{\text{mf}}(t,s)) \tilde{R}_d^{\text{mf}}(t',s) \, \mathrm{d}s \,,$$

$$\partial_t \tilde{R}_d^{\text{mf}}(t,t') = \delta(t-t') - \tilde{\nu}^{\text{mf}}(t) \tilde{R}_d^{\text{mf}}(t,t') + \int_{t'}^t \tilde{h}''(\tilde{C}_d^{\text{mf}}(t,s)) \tilde{R}_d^{\text{mf}}(t,s) \tilde{R}_d^{\text{mf}}(s,t') \, \mathrm{d}s \tag{G.22}$$

$$\tilde{\nu}^{\text{mf}}(t) = \int_0^t \tilde{h}''(\tilde{C}_d^{\text{mf}}(t,s)) \tilde{R}_d^{\text{mf}}(t,s) \tilde{C}_d^{\text{mf}}(t,s) \, \mathrm{d}s + \int_0^t \tilde{h}'(\tilde{C}_d^{\text{mf}}(t,s)) \tilde{R}_d^{\text{mf}}(t,s) \, \mathrm{d}s \,, \tag{G.23}$$

where again $\tilde{h}(z) = h(z) - h'(0)z$.

Equations (G.21), (G.23) are independent of the dynamics of the second layer weights. These equations are nothing but the DMFT equations describing gradient descent dynamics of the celebrated spherical mixed $p$-spin glass model [17, 20, 8, 23], whose definition we recall next. Consider a random cost function $H(\boldsymbol{x})$ indexed $\boldsymbol{x} \in \mathbb{S}^{d-1}$, which is a centered Gaussian process with covariance structure given by

$$\mathbb{E}\left(H(\boldsymbol{x})H(\boldsymbol{y})\right) = d\,\tilde{h}(\langle \boldsymbol{x}, \boldsymbol{y} \rangle) \,. \tag{G.24}$$

Define the gradient flow dynamics

$$\dot{\boldsymbol{x}}(t) = -\boldsymbol{P}_{\boldsymbol{x}(t)}^{\perp} \nabla H(\boldsymbol{x}(t)) \,, \tag{G.25}$$

where $\boldsymbol{P}_{\boldsymbol{x}(t)}^{\perp}$ is the projector orthogonal to $\boldsymbol{x}(t)$. Then the high-dimensional asymptotics of this dynamics is characterized by Eqs. (G.21), (G.23). In particular $\lim_{d\to\infty} \langle \boldsymbol{x}(t), \boldsymbol{x}(s) \rangle = \tilde{C}_d^{\text{mf}}(t,t')$ almost surely.

A particularly interesting quantity is the asymptotic energy value in the mixed $p$-spin model:

$$\mathscr{E} = \lim_{t\to\infty} \lim_{d\to\infty} \frac{1}{d} H(\boldsymbol{x}(t)) \,. \tag{G.26}$$

The DMFT analysis for this problem implies that

$$\mathscr{E} = -\lim_{t\to\infty} \int_0^t \tilde{h}'(\tilde{C}_d^{\text{mf}}(t,s)) \hat{R}_d^{\text{mf}}(t,s) \, \mathrm{d}s \,. \tag{G.27}$$

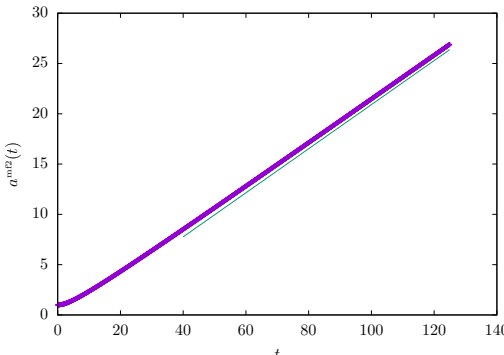

Figure 20: **Evolution of second layer weights, as predicted by the numerical solution of Eq.** (G.17).
Here we use $h(z) = (9/10)z + z^3/6$, $\alpha = 0.3$ and $\tau = 0.6$. The straight line is just a guide to the eyes to test the prediction of Eq. (G.29).

For $\tilde{h}(z) = c_k^2 z^k$, $k \geq 2$, we have the explicit expression [20, 19, 46]

$$\mathscr{E} = -2c_k \sqrt{\frac{k-1}{k}} \; . \tag{G.28}$$

An explicit expression for $\mathscr{E}$ for general covariance structure is an unknown [23].

The asymptotic energy $\mathscr{E}$ has an interesting interpretation for the dynamics of two-layer networks –within the SymmDMFT theory. Eq. (G.28) implies that

$$\lim_{t \to \infty} \frac{a^{\mathrm{mf2}}(t)}{t} = -\tau\sqrt{\alpha}\mathscr{E} =: A_\infty \; . \tag{G.29}$$

In Fig. 20 we test the prediction of Eq. (G.29) by integrating numerically Eqs. (G.14) to (G.17) and plotting the prediction for the second-layer weigths $a^{\mathrm{mf}}(t)$. We observe that at large $t$, $a^{\mathrm{mf2}}(t) \approx A_\infty t$, with $A_\infty$ given by Eq. (G.29)as predicted.

We also note that $C_A(t,t) = -\tau^2$ also in this timescale, and hence the train error does not change significantly. Namely , for any constant $t$, we have

$$e_{\mathrm{tr}}(t\sqrt{m}) = \frac{1}{2}\tau^2 + o_m(1) \; . \tag{G.30}$$

If we use heuristically Eq. (G.28) and Eq. (G.12) beyond the $\sqrt{t}$ time scale, we obtain

$$a(t) \approx a^{\mathrm{mf2}}\big(t/\sqrt{m}\big) \approx A_\infty \frac{t}{\sqrt{m}} \; . \tag{G.31}$$

This suggests that $a(t)$ becomes of order $\sqrt{m}$ on timescale of order $m$. When this happens, the network complexity is large enough to allow for interpolation, and hence we expect the dynamics to change. Indeed a new dynamical regime emerges for $t = \Theta(m)$, as we will study next.

### G.1.3 Third dynamical regime: $t = \Theta(m)$

As anticipated, an additional regime arises on timescales of order $m$. In Figure 21 we plot the evolution of the weights of the second layer as a function of $t/m$ for increasing values of the width $m$. The different curves collapse suggesting the following limit to exist

$$\lim_{m \to \infty} \frac{a(tm)}{\sqrt{m}} = \gamma^{\mathrm{mf3}}(t) \; . \tag{G.32}$$

The limit curve appears to grows linearly at small $t$, $\gamma^{\mathrm{mf3}}(t) = A_\infty t + o(t)$, where $A_\infty$ is the coefficient computed in the previous section, cf. Eq. (G.29). Hence, this third dynamical regime matches directly with the previous one. As can be seen from the right plot, there appear to be a finite limit $\lim_{t \to \infty} \gamma^{\mathrm{mf3}}(t) < \infty$.

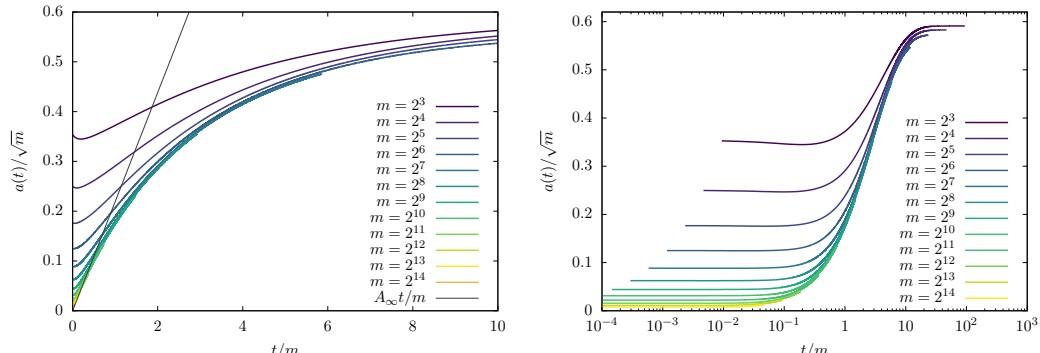

Figure 21: **Evolution of the second layer weigths when training on pure noise data under mean field initialization for** $t = \Theta(m)$. Rescaled second layer weights $a(t)/\sqrt{m}$ as a function of $t/m$. We plot solutions of the SymmDMFT equations for the setting of Fig. 18.

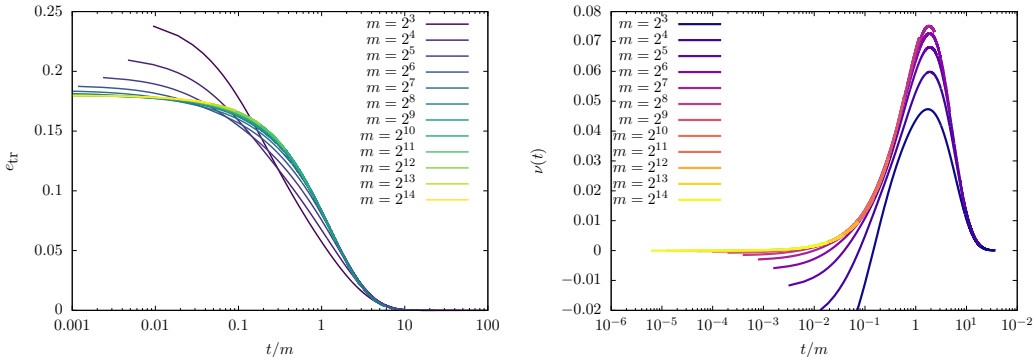

Figure 22: **Train error and Lagrange multiplier** $\nu(t)$ **on timescales of order** $m$ **under mean field initialization for pure noise data.** Solutions of the SymmDMFT equations for the setting of Fig. 18. Finite $m$ curves accumulate on master curves suggesting the existence of scaling functions.

We now turn to the analysis of the train error. Recall that on the previous timescales, the train error stays approximately constant, and equal to the train error of the null network, namely $e_{\text{tr}}(t\sqrt{m}) = \tau^2/2 + o_m(1)$ for any fixed $t$. In Fig. 22 we plot both the train error and the Lagrange multiplier $\nu$ as a function of $t/m$. Again, as $m$ grows, these curve converge to limit curves. This suggests the existence of the following limits

$$\lim_{m \to \infty} e_{\text{tr}}(tm) = e_{\text{tr}}^{\text{mf3}}(t), \tag{G.33}$$

$$\lim_{m \to \infty} \nu(tm) = \nu^{\text{mf3}}(t). \tag{G.34}$$

Note that in this case, differently from the lazy initialization setting, the corresponding scaling function do not depend on the initialization parameter $a_0$.

In order to characterize the limits in Eqs. (G.33)-(G.34), we proceed as in Sec. F.1.4. Namely, in Fig. 23 we plot the train error and the Lagrange multiplier $\nu$ as a function of the rescaled second layer weights $\gamma = a(t)/\sqrt{m}$. We also plot the asymptotic value of train error and Lagrange multiplier under the constrained GF dynamics in which second layer weigths are fixed to $a(t) = \gamma\sqrt{m}$ and do not evolve with time: $e_{\text{tr},\infty}^{\text{lz2}}(\gamma) := \lim_{t \to \infty} e_{\text{tr}}^{\text{lz2}}(t, \gamma)$ and $\nu_{\infty}^{\text{lz2}}(\gamma) := \lim_{t \to \infty} \nu^{\text{lz2}}(t, \gamma)$. These are computed by integration of the scaling theory in Section F.1.2.

The good collapse on these curves suggests to consider the the following construction, analogous to Sec. F.1.4. Define the inverse function of $t \mapsto \gamma^{\text{mf3}}(t)$, denoted by $(\gamma^{\text{mf3}})^{-1}$. Then, define

$$
\begin{aligned}
\varepsilon^{\text{mf3}}(\gamma) &= e_{\text{tr}}^{\text{mf3}}((\gamma^{\text{mf3}})^{-1}(\gamma)), \\
\nu_*^{\text{mf3}}(\gamma) &= \nu^{\text{mf3}}((\gamma^{\text{mf3}})^{-1}(\gamma)).
\end{aligned}
\tag{G.35}
$$

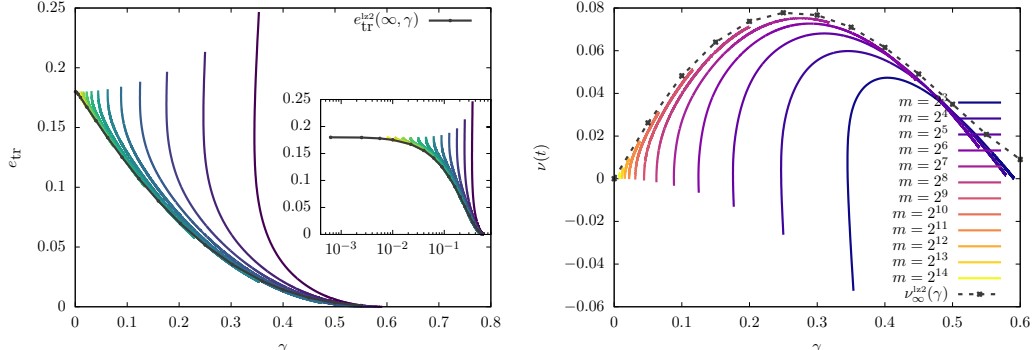

Figure 23: **Train error, rescaled second-layer weights and the Lagrange multiplier $\nu(t)$ on timescales of order $m$ under mean field initialization.** Left Panel: parametric plot of the train error as a function of the weights of the second layer on the scale $\sqrt{m}$, namely $\gamma = a(t)/\sqrt{m}$. The inset shows the same data on a logarithmic scale. Right Panel: same plot for the Lagrange multiplier $\nu$. Data is the same as in Fig. 18.

Figure 23 suggests that

$$\varepsilon^{\text{mf3}}(\gamma) \approx e_{\text{tr},\infty}^{\text{lz2}}(\gamma)\,, \tag{G.36}$$

$$\nu_*^{\text{mf3}}(\gamma) \approx \nu_{\text{tr},\infty}^{\text{lz2}}(\gamma)\,. \tag{G.37}$$

Equations (G.36), (G.36) imply that on timescales of order $m$ the dynamics is adiabatic. For each incremental change of $a$ on a the scale $\sqrt{m}$, all one-time quantities relax to the asymptotic value which turns out to be the same as a constrained model with $a(t)/\sqrt{m} = \gamma$ fixed.

The consequence of Eqs. (G.36)-(G.36) is that

$$\lim_{t \to \infty} \gamma^{\text{mf3}}(t) \approx \gamma_{\text{GF}}^*(\alpha, \tau)\,. \tag{G.38}$$

where $\gamma_{\text{GF}}^*(\alpha, \tau)$ corresponds to the interpolation value of the initialization scale of a lazy model.

## G.2   Multi-index model

In this section we consider the case in which the dataset is distributed according to a multi-index model. For time scales beyond $t = O(1)$, we will assume that $h(z) = \hat{\varphi}(z)$. This simplifies the asymptotics for $t$ large but of order one.

We identify two dynamical regimes emerging as $m \to \infty$:

- $t = O(1)$: $a(t) = O(1)$ but is not constant. Also, the projection $\boldsymbol{v}(t)$ of first layer weights onto the latent space evolve as well as do train and test error. We further have $e_{\text{tr}}(t) = e_{\text{ts}}(t) + o_m(1)$: there is no overfitting. This evolution is captured the mean field theory of [38, 14] which we recover as $m \to \infty$ limit of SymmDMFT.

- $t = \Theta(m)$: $a(t) = \Theta(\sqrt{m})$, $\boldsymbol{v}(t)$ decreases towards 0 and train and test error diverge. In this dynamical regime the network unlearns to a large extent the latent structure of the data and overfit it.

### G.2.1   First dynamical regime: $t = \Theta(1)$

For timescales of order one, the SymmDMFT equations are solved, up to subleading terms as $m \to \infty$, by the following ansatz

$$C_d(t, s) = C_d^{\text{mf1}}(t, s) + o_m(1)\,, \qquad C_o(t, s) = C_o^{\text{mf1}}(t, s) + o_m(1)\,, \tag{G.39}$$

$$R_d(t, s) = R_d^{\text{mf1}}(t, s) + o_m(1)\,, \qquad m R_o(t, s) = R_o^{\text{mf1}}(t, s) + o_m(1) \tag{G.40}$$

$$v(t) = v^{\text{mf1}}(t) + o_m(1)\,, \qquad a(t) = a^{\text{mf1}}(t)\,. \tag{G.41}$$

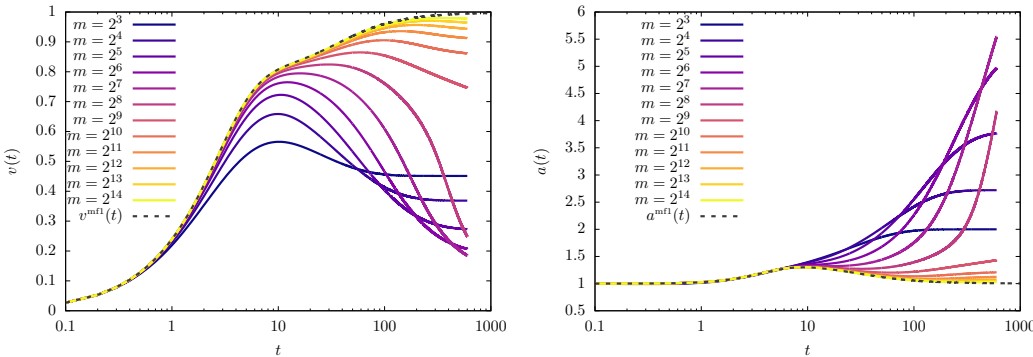

Figure 24: **Gradient flow dynamics under mean field initialization in the first dynamical regime**
$t = O(1)$**.** for data distributed according to a single index model. Curves are numerical solutions of
the SymmDMFT equations: we plot $v(t)$ and $a(t)$ for different values of $m$ and compare them to the
mean field predictions. Data is distributed according to a single index model with $h_t(z) = \hat{\varphi}(z) =$
$h(z) = (9/10)z + z^3/6$ with $\tau = 0.6$ and $\alpha = 0.3$.

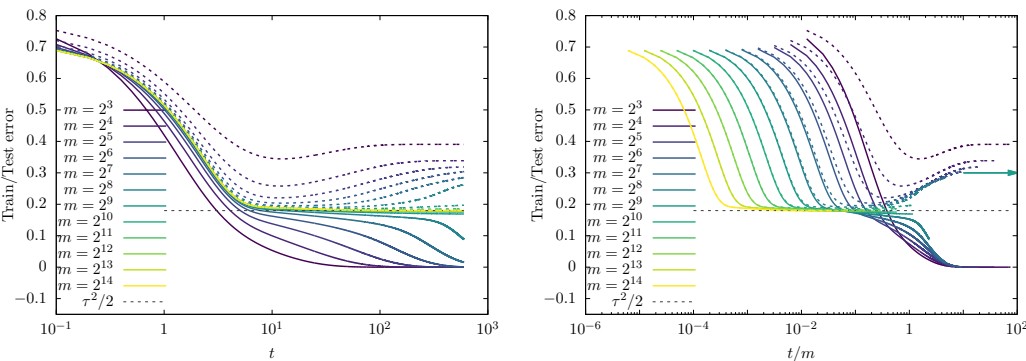

Figure 25: **Evolution of the train and test error on different timescales under mean field
initialization** $a(0) = 1$**.** The train (solid curves) and test (dashed curves) errors as a function
of time $t$ (left panel) and scaled time $t/m$ (right panel). Curves are numerical solutions of the
SymmDMFT equations for $h(z) = (9/10)z + z^3/6$, $\hat{\varphi}(z) = h(z)$, $\tau = 0.6$ and $\alpha = 0.3$. The arrow
on the right panel corresponds to the asymptotic test error for a model with second layer weights
fixed to the corresponding interpolation threshold.

The corresponding scaling equations are then given by

$$
\begin{aligned}
&\partial_t R_o^{\mathrm{mfl}}(t,t') = -\nu^{\mathrm{mfl}}(t) R_o^{\mathrm{mfl}}(t,t') - \alpha a^{\mathrm{mfl}}(t)^2 h'(C_o^{\mathrm{mfl}}(t,t)) \left( R_d^{\mathrm{mfl}}(t,t') + R_o^{\mathrm{mfl}}(t,t') \right) , \\
&\partial_t C_o^{\mathrm{mfl}}(t,t') = -\nu^{\mathrm{mfl}}(t) C_o^{\mathrm{mfl}}(t,t') + \alpha \langle \nabla \hat{\varphi}(\boldsymbol{v}^{\mathrm{mfl}}(t)), \boldsymbol{v}^{\mathrm{mfl}}(t') \rangle a^{\mathrm{mfl}}(t) - \alpha a^{\mathrm{mfl}}(t)^2 h'(C_o^{\mathrm{mfl}}(t,t)) C_o^{\mathrm{mfl}}(t,t') , \\
&\qquad \nu^{\mathrm{mfl}}(t) = \alpha \langle \nabla \hat{\varphi}(\boldsymbol{v}^{\mathrm{mfl}}(t)), \boldsymbol{v}^{\mathrm{mfl}}(t) \rangle a^{\mathrm{mfl}}(t) - \alpha a^{\mathrm{mfl}}(t)^2 h'(C_o^{\mathrm{mfl}}(t,t)) C_o^{\mathrm{mfl}}(t,t) \\
&\partial_t C_d^{\mathrm{mfl}}(t,t') = -\nu^{\mathrm{mfl}}(t) C_d^{\mathrm{mfl}}(t,t') + \alpha \langle \nabla \hat{\varphi}(\boldsymbol{v}^{\mathrm{mfl}}(t)), \boldsymbol{v}^{\mathrm{mfl}}(t') \rangle a^{\mathrm{mfl}}(t) - \alpha a^{\mathrm{mfl}}(t)^2 h'(C_o^{\mathrm{mfl}}(t,t)) C_o^{\mathrm{mfl}}(t,t') , \\
&\partial_t R_d^{\mathrm{mfl}}(t,t') = -\nu^{\mathrm{mfl}}(t) R_d^{\mathrm{mfl}}(t,t') + \delta(t - t') , \\
&\quad \partial_t \boldsymbol{v}^{\mathrm{mfl}}(t) = -\nu^{\mathrm{mfl}}(t) \boldsymbol{v}^{\mathrm{mfl}}(t) + \alpha \nabla \hat{\varphi}(\boldsymbol{v}^{\mathrm{mfl}}(t)) a^{\mathrm{mfl}}(t) - \alpha a^{\mathrm{mfl}}(t)^2 h'(C_o^{\mathrm{mfl}}(t,t)) \boldsymbol{v}^{\mathrm{mfl}}(t) , \\
&\quad \partial_t a^{\mathrm{mfl}}(t) = \alpha \left( \hat{\varphi}(\boldsymbol{v}^{\mathrm{mfl}}(t)) - a^{\mathrm{mfl}}(t) h(C_o^{\mathrm{mfl}}(t,t)) \right) .
\end{aligned}
\tag{G.42}
$$

These equations are solved by setting:

$$
C_o^{\mathrm{mfl}}(t,t') = \langle \boldsymbol{v}^{\mathrm{mfl}}(t), \boldsymbol{v}^{\mathrm{mfl}}(t') \rangle
\tag{G.43}
$$

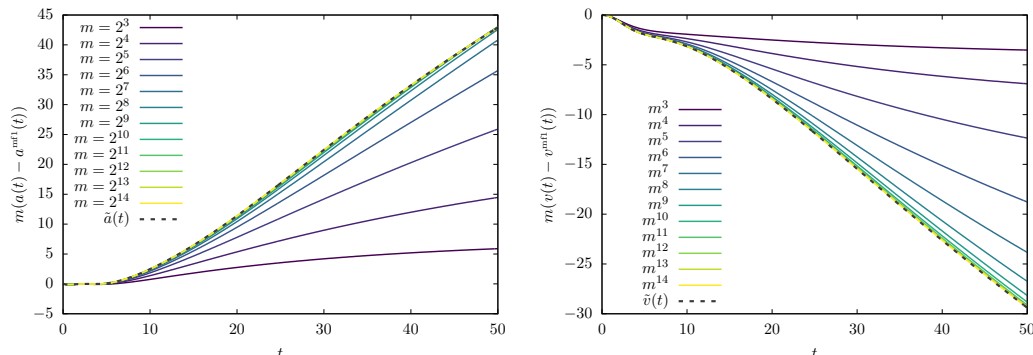

Figure 26: **Finite width corrections.** The $1/m$ corrections to the second layer weights and the projection on the latent space of the single index model on timescales of order 1. Dashed lines are obtained by integrating numerically Eqs. (G.56) to (G.59) determining the limits $m \to \infty$. Here, $\hat{\varphi}(z) = h(z) = (9/10)z + z^3/6$ with $\tau = 0.6$ and $\alpha = 0.3$.

with $\boldsymbol{v}^{\mathrm{mfl}}(t)$, $a^{\mathrm{mfl}}(t)$ the solution of

$$
\partial_t \boldsymbol{v}^{\mathrm{mfl}}(t) = \alpha a^{\mathrm{mfl}}(t)\big(\boldsymbol{I}_k - \boldsymbol{v}^{\mathrm{mfl}}(t)\boldsymbol{v}^{\mathrm{mfl}}(t)^{\mathsf{T}}\big)\big(\nabla\hat{\varphi}(\boldsymbol{v}^{\mathrm{mfl}}(t)) - a^{\mathrm{mfl}}(t)h'(\|\boldsymbol{v}^{\mathrm{mfl}}(t)\|^2)\boldsymbol{v}^{\mathrm{mfl}}(t)\big) \,,
$$
$$
\partial_t a^{\mathrm{mfl}}(t) = \alpha\big(\hat{\varphi}(\boldsymbol{v}^{\mathrm{mfl}}(t)) - a^{\mathrm{mfl}}(t)h(\|\boldsymbol{v}^{\mathrm{mfl}}(t)\|^2)\big) \,,
$$

$$\tag{G.44}$$

with initial conditions given by $\boldsymbol{v}^{\mathrm{mfl}}(0) = \boldsymbol{0}$ and $a^{\mathrm{mfl}}(0) = a_0$.

Equations (G.44) coincide with the mean field theory of [38, 14, 45], when the latter are specialized to the multi-index model studied here, under symmetric initializations [10]. (See also [2].) Using the ansatz of Eqs. (G.39) to (G.41) in the formulas for training and test error (C.45), (C.46), we get

$$
\lim_{m \to \infty} e_{\mathsf{tr}}(t) = \lim_{m \to \infty} e_{\mathsf{ts}}(t) = e^{\mathrm{mfl}}(t) \,,
\tag{G.45}
$$

with

$$
e^{\mathrm{mfl}}(t) = \frac{1}{2}\left[\tau^2 + \|\varphi\|^2 - 2a^{\mathrm{mfl}}(t)\hat{\varphi}(\boldsymbol{v}^{\mathrm{mfl}}(t)) + a^{\mathrm{mfl}}(t)^2 h(\|\boldsymbol{v}^{\mathrm{mfl}}(t)\|^2)\right] \,.
\tag{G.46}
$$

A particularly simple case is the one in which $k = 1$ (single index model) and $\varphi = \sigma$ (whence $\hat{\varphi} = h$). For a class of such activations with $h'(0) > 0$, we have $a^{\mathrm{mfl}}(t), v^{\mathrm{mfl}}(t) \to 1$ as $t \to \infty$ and therefore

$$
\lim_{t \to \infty} e^{\mathrm{mfl}}(t) = \frac{\tau^2}{2} \,.
\tag{G.47}
$$

In other words, neurons align perfectly with latent direction, the generalization error vanishes, and and train and test error converge for large constant $t$ to the Bayes error $\tau^2/2$.

In Fig. 24 we compare the solution of Eqs. (G.44) with the numerical integrations of the SymmDMFT equations for a range of values of $m$. As $m$ increases, the SymmDMFT solutions converge to the asymptotic predictions $v^{\mathrm{mfl}}(t)$, $a^{\mathrm{mfl}}(t)$, confirming the above ansatz.

Similarly, in Fig. 25-left panel we compute the train and test error by solving the SymmDMFT equations and compare the results to the asymptotic prediction provided by Eq. (G.46). We observe that –as predicted– train and test error match on an increasingly long time interval. At a certain point, they diverge: we will next characterize the timescale on which this happens.

### G.2.2 Escape from the mean field dynamical regime

In order to understand on which time scale the dynamics diverges from mean field theory described above, we will study small deviations from this theory. We expect that these deviations will diverge with time. Characterizing this divergence will allow to determine time scale on which we exit the mean field regime.

We focus on the case of a single index model $k = 1$, with $\hat{\varphi} = h$, and set $a(0) = 1$. We believe that the qualitative conclusions obtained in this case apply more generally. We also assume $h$ to be such

that the long time asymptotics of mean field dynamical solutions is

$$\lim_{t\to\infty} a^{\text{mfl}}(t) = 1\,, \qquad \lim_{t\to\infty} v^{\text{mfl}}(t) = 1\,. \tag{G.48}$$

As mentioned in the previous section, this holds for a broad class of activations. In other words, for time $t$ large and yet of order one, the neurons are very well aligned.

We next study the corrections to the mean field solution. We claim that such corrections are of order $1/m$ and define the functions $\tilde{a}(t)$, $\tilde{v}(t)$, dots ,$\tilde{R}_o(t, t')$, via

$$m(a(t) - a^{\text{mfl}}(t)) = \tilde{a}(t) + o_m(1)\,, \tag{G.49}$$

$$m(v(t) - v^{\text{mfl}}(t)) = \tilde{v}(t) + o_m(1)\,, \tag{G.50}$$

$$m(C_d(t, t') - C_d^{\text{mfl}}(t, t')) = \tilde{C}_d(t, t') + o_m(1)\,, \tag{G.51}$$

$$m(C_o(t, t') - C_o^{\text{mfl}}(t, t')) = \tilde{C}_o(t, t') + o_m(1)\,, \tag{G.52}$$

$$m(R_d(t, t') - R_d^{\text{mfl}}(t, t')) = \tilde{R}_d(t, t') + o_m(1)\,, \tag{G.53}$$

$$m(R_o(t, t') - R_o^{\text{mfl}}(t, t')) = \tilde{R}_o(t, t') + o_m(1)\,, \tag{G.54}$$

$$m(\nu(t) - \nu^{\text{mfl}}(t)) = \tilde{\nu}(t) + o_m(1)\,. \tag{G.55}$$

Substituting the above form into the SymmDMFT equations and matching the next-to-leading order in $m$ we can obtain the equations for the $1/m$ corrections. It turns out that equations for $\tilde{a}$, $\tilde{v}$, $\tilde{C}_o$ and $\tilde{\nu}$ decouple from the equations for $\tilde{C}_d$, $\tilde{R}_d$ and $\tilde{R}_o$. Given that we are interested in the former quantities we only report the corresponding equations:

$$\frac{\mathrm{d}\tilde{a}(t)}{\mathrm{d}t} = \alpha\hat{\varphi}'(v^{\text{mfl}}(t))\tilde{v}(t) - \alpha\hat{\varphi}(v^{\text{mfl}}(t))\int_0^t \Sigma_R^{(1)}(t, s)\,\mathrm{d}s - \alpha a^{\text{mfl}}(t)\left[h(1) - h(C_o^{\text{mfl}}(t, t))\right] \tag{G.56}$$

$$+ \alpha\int_0^t \Sigma_R^{(1)}(t, s)a^{\text{mfl}}(s)h(C_o^{\text{mfl}}(t, s))\mathrm{d}s - \alpha\tilde{a}(t)h(C_o^{\text{mfl}}(t, t)) - \alpha a^{\text{mfl}}(t)h'(C_o^{\text{mfl}}(t, t))\tilde{C}_o(t, t)$$

$$- \alpha\int_0^t C_A^{\text{mfl}}(t, s)a^{\text{mfl}}(s)\left[h'(C_d^{\text{mfl}}(t, s))R_d^{\text{mfl}}(t, s) + h'(C_o^{\text{mfl}}(t, s))R_o^{\text{mfl}}(t, s)\right]\mathrm{d}s\,,$$

$$\frac{\mathrm{d}\tilde{v}(t)}{\mathrm{d}t} = -\nu^{\text{mfl}}(t)\tilde{v}(t) - \tilde{\nu}(t)v^{\text{mfl}}(t) + \alpha\hat{\varphi}(v^{\text{mfl}}(t))\tilde{a}(t) + \alpha\hat{\varphi}''(v^{\text{mfl}}(t))\tilde{v}(t)a^{\text{mfl}}(t) \tag{G.57}$$

$$- \alpha\hat{\varphi}'(v^{\text{mfl}}(t))a^{\text{mfl}}(t)\int_0^t \Sigma_R^{(1)}(t, s)\,\mathrm{d}s - \int_0^t \left[\tilde{M}_R^{(d)}(t, s) - M_{R,o}^{(0)}(t, s)\right]v^{\text{mfl}}(s)\,\mathrm{d}s$$

$$- \int_0^t \left[M_{R,o}^{(1)}(t, s)v^{\text{mfl}}(s) + M_{R,o}^{(0)}(t, s)\tilde{v}(s)\right]\mathrm{d}s\,,$$

$$\tilde{\nu}(t) = \alpha\hat{\varphi}'(v^{\text{mfl}}(t))\tilde{v}(t)a^{\text{mfl}}(t) + \alpha\hat{\varphi}''(v^{\text{mfl}}(t))v^{\text{mfl}}(t)\tilde{v}(t)a^{\text{mfl}}(t) \tag{G.58}$$

$$+ \alpha\hat{\varphi}'(v^{\text{mfl}}(t))v^{\text{mfl}}(t)\tilde{a}(t) - \alpha\hat{\varphi}'(v^{\text{mfl}}(t))v^{\text{mfl}}(t)\int_0^t \Sigma_R^{(1)}(t, s)\,\mathrm{d}s$$

$$- \int_0^t \left[\tilde{M}_R^{(d)}(t, s)C_d^{\text{mfl}}(t, s) - M_{R,o}^{(0)}(t, s)C_o^{\text{mfl}}(t, s)\right]\mathrm{d}s$$

$$- \int_0^t \left[M_{R,o}^{(1)}(t, s)C_o^{\text{mfl}}(t, s) + M_{R,o}^{(0)}(t, s)\tilde{C}_o(t, s)\right]\mathrm{d}s$$

$$- \int_0^t \left[\tilde{M}_C^{(d)}(t, s)R_d^{\text{mfl}}(t, s) + M_{C,o}^{(0)}(t, s)R_o^{\text{mfl}}(t, s)\right]\mathrm{d}s\,,$$

$$\frac{\partial\tilde{C}_o(t, t')}{\partial t} = -\nu^{\text{mfl}}(t)\tilde{C}_o(t, t') - \tilde{\nu}(t)C_o^{\text{mfl}}(t, t') + \alpha\hat{\varphi}''(v^{\text{mfl}}(t))\tilde{v}(t)v^{\text{mfl}}(t')a^{\text{mfl}}(t) \tag{G.59}$$

$$+ \alpha\hat{\varphi}'(v^{\text{mfl}}(t))\tilde{v}(t')a^{\text{mfl}}(t) + \alpha\hat{\varphi}'(v^{\text{mfl}}(t))v^{\text{mfl}}(t')\tilde{a}(t)$$

$$- \alpha\hat{\varphi}'(v^{\text{mfl}}(t))v^{\text{mfl}}(t')a^{\text{mfl}}(t)\int_0^t \Sigma_R^{(1)}(t, s)\,\mathrm{d}s$$

$$- \int_0^t \left[\tilde{M}_R^{(d)}(t, s)C_o^{\text{mfl}}(t', s) + M_{R,o}^{(0)}(t, s)C_d^{\text{mfl}}(t', s) - 2M_{R,o}^{(0)}(t, s)C_o^{\text{mfl}}(t's)\right]\mathrm{d}s$$

$$-\int_0^t \left[ M_{R,o}^{(0)}(t,s)\tilde{C}_o(t',s) + M_{R,o}^{(1)}(t,s)C_o^{\mathrm{mfl}}(t',s) \right] ds$$

$$-\int_0^{t'} \left[ \tilde{M}_C^{(d)}(t,s)R_d^{\mathrm{mfl}}(t',s) + M_{C,o}^{(0)}(t,s)R_o^{\mathrm{mfl}}(t',s) \right] ds .$$

Here, we used the following auxiliary functions

$$\Sigma_R^{(1)}(t,s) = a^{\mathrm{mfl}}(t)a^{\mathrm{mfl}}(s) \left[ h'(C_d^{\mathrm{mfl}}(t,s))R_d^{\mathrm{mfl}}(t,s) + h'(C_o^{\mathrm{mfl}}(t,s))R_o^{\mathrm{mfl}}(t,s) \right] , \tag{G.60}$$

$$C_A^{\mathrm{mfl}}(t,s) = -\left[ \tau^2 + h_t(1) - a^{\mathrm{mfl}}(t)\varphi(v^{\mathrm{mfl}}(t)) - a^{\mathrm{mfl}}(s)\varphi(v^{\mathrm{mfl}}(s)) + a^{\mathrm{mfl}}(t)a^{\mathrm{mfl}}(s)h(C_o^{\mathrm{mfl}}(t,s)) \right] , \tag{G.61}$$

$$\tilde{M}_R^{(d)}(t,s) = \alpha a^{\mathrm{mfl}}(t)a^{\mathrm{mfl}}(s) \left[ h'(1)\delta(t-s) + C_A^{\mathrm{mfl}}(t,s)h''(C_d^{\mathrm{mfl}}(t,s))R_d^{\mathrm{mfl}}(t,s) \right] , \tag{G.62}$$

$$M_{R,o}^{(0)}(t,s) = \alpha(a^{\mathrm{mfl}}(t))^2 h'(C_o^{\mathrm{mfl}}(t,s))\delta(t-s) , \tag{G.63}$$

$$M_{R,o}^{(1)}(t,s) = \alpha \left[ 2a^{\mathrm{mfl}}(t)\tilde{a}(t)h'(C_o^{\mathrm{mfl}}(t,t)) + (a^{\mathrm{mfl}}(t))^2 h''(C_o^{\mathrm{mfl}}(t,t))\tilde{C}(t,t) \right] \delta(t-s) \tag{G.64}$$

$$- \alpha a^{\mathrm{mfl}}(t)a^{\mathrm{mfl}}(s)\Sigma_R^{(1)}(t,s)h'(C_o^{\mathrm{mfl}}(t,s)) \tag{G.65}$$

$$+ \alpha a^{\mathrm{mfl}}(t)a^{\mathrm{mfl}}(s)C_A^{\mathrm{mfl}}(t,s)h''(C_o^{\mathrm{mfl}}(t,s))R_o^{\mathrm{mfl}}(t,s) , \tag{G.66}$$

$$\tilde{M}_C^{(d)}(t,s) = \alpha a^{\mathrm{mfl}}(t)a^{\mathrm{mfl}}(s)C_A^{\mathrm{mfl}}(t,s)h'(C_d^{\mathrm{mfl}}(t,s)) , \tag{G.67}$$

$$M_{C,o}^{(0)}(t,s) = \alpha a^{\mathrm{mfl}}(t)a^{\mathrm{mfl}}(s)C_A^{\mathrm{mfl}}(t,s)h'(C_o^{\mathrm{mfl}}(t,s)) . \tag{G.68}$$

Note that Eqs. (G.56) to (G.59) are a set of four integral-differential equations for the four functions $\tilde{a}(t), \tilde{v}(t), \tilde{\nu}(t), \tilde{C}_o(t,t')$. The original SymmDMFT equations involve three other functions: $\tilde{C}_d(t,t'), \tilde{R}_d(t,t'), \tilde{R}_o(t,t')$? We also remark that: $(i)$ These equations are linear in the unknowns $\tilde{a}(t), \tilde{v}(t), \tilde{\nu}(t), \tilde{C}_o(t,t')$; $(ii)$ They can be integrated numerically with the same strategy used to integrate the SymmDMFT equations.

In Fig. 26 we plot the deviations from the mean field limit $m(a(t) - a^{\mathrm{mfl}}(t))$ and $m(v(t) - v^{\mathrm{mfl}}(t))$ as a function of time $t$, as obtained by solving the SymmDMFT equations[1], for several values of $m$. We also plot the predicted limits $\tilde{a}(t), \tilde{v}(t)$, which are obtained by integrating Eqs. (G.56) to (G.59) As $m$ gets large, the finite-$m$ curves appear to converge to the predictions $\tilde{a}(t), \tilde{v}(t)$.

In Figure 27 we plot the result of integrating Eqs. (G.56) to (G.59) over a wider time window. We observe that $\tilde{v}, \tilde{a}, \tilde{\nu}$ and $\tilde{C}_o(t,t)$ diverge linearly with $t$.

This suggests the following asymptotics for these corrections

$$\lim_{t\to\infty} \frac{\tilde{a}(t)}{t} = a_* , \qquad\qquad \lim_{t\to\infty} \frac{\tilde{v}(t)}{t} = v_* , \tag{G.69}$$

$$\lim_{t\to\infty} \frac{\tilde{\nu}(t)}{t} = \nu_* , \qquad\qquad \lim_{t\to\infty} \frac{\tilde{C}_o(t,t)}{t} = c_* . \tag{G.70}$$

The values of the constant $a_*, v_*, \nu_*$ and $c_*$ can be obtained analytically by using the above ansatz in Eqs. (G.56) to (G.59). We obtain that they solve the following linear equations

$$0 = \hat{\varphi}'(1)v_* + \hat{\varphi}(1)a_* , \tag{G.71}$$

$$0 = \hat{\varphi}'(1)c_* + 2\hat{\varphi}(1)a_* , \tag{G.72}$$

$$0 = -\hat{\varphi}'(1)\nu_* - \hat{\varphi}(1)\left(\alpha\hat{\varphi}''(1) - \alpha\hat{\varphi}'(1) - \alpha(\hat{\varphi}'(1))^2\right)a_* + 2\alpha\hat{\varphi}(1)\hat{\varphi}''(1) , \tag{G.73}$$

$$0 = -\frac{1}{2}c_* - \nu_1 c_* - 2\nu_* v_1 + 4v_1\alpha\tau^2 , \tag{G.74}$$

where

$$v_1 := \lim_{t\to\infty} (v(t) - 1)t , \tag{G.75}$$

$$\nu_1 := \lim_{t\to\infty} \tilde{\nu}(t)t . \tag{G.76}$$

The asymptotic linear behavior predicted by Eqs. (G.69), (G.70), with the coefficients determined by Eqs. (G.71)-(G.74) is plotted in Fig. 27. We observe good agreement with the numerical integration of Eqs. (G.56) to (G.59).

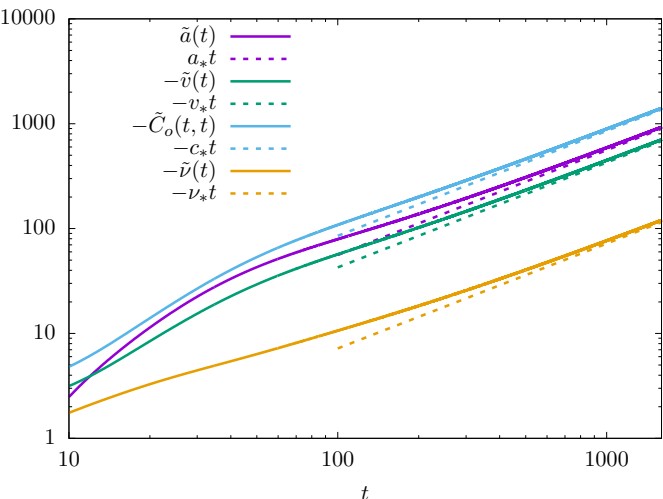

Figure 27: **Finite width corrections to dynamical observables under mean field initialization.** The $1/m$ corrections to $v(t)$, $a(t)$, $C_o(t,t)$ and $\tilde{\nu}(t)$ as a function of time as extracted from the numerical integration of the corresponding equations. The dashed lines are the asymptotic predictions for $t \to \infty$ which show that the divergence of all quantities is linear with time. Here, $\hat{\varphi}(z) = h(z) = (9/10)z + z^3/6$ with $\tau = 0.6$ and $\alpha = 0.3$.

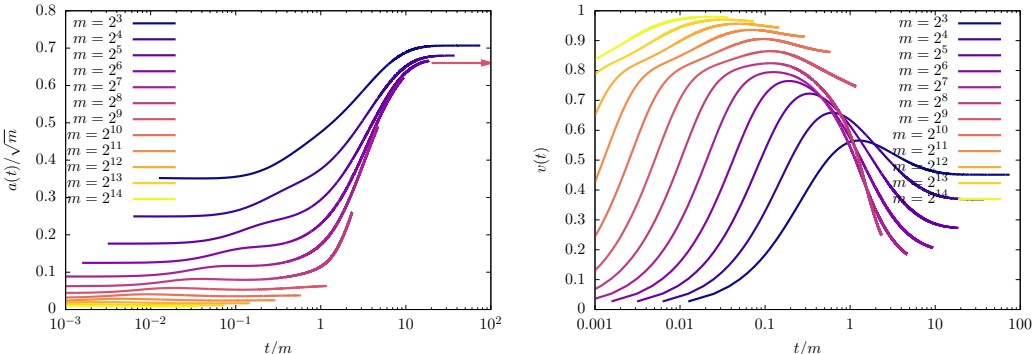

Figure 28: **econd layer weights and projection on of the first layer weigths onto the latent structure of the data for gradient flow under mean field initialization on timescales of order $m$.** Left: rescaled second layer weights $a(t)/m$ as a function of the rescaled time $t/m$. The arrow on the right points at the threshold $\gamma_{\text{GF}}^*(\alpha, \varphi, \tau)$ for interpolation under gradient flow, see Section F.2.3. Right: projection of the first layer weights on the latent space in the single index model as a function of rescaled time $t/m$. Here, $\hat{\varphi}(z) = h(z) = (9/10)z + z^3/6$ with $\tau = 0.6$ and $\alpha = 0.3$. $v = 1/\gamma$ in (F.57).

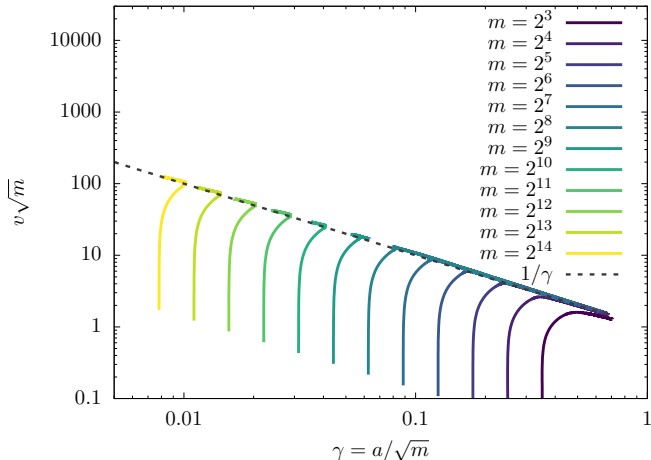

Figure 29: **Parametric plot of the rescaled projection onto the latent direction** $v\sqrt{m}$ **against rescaled second layer weights** $\gamma = a/\sqrt{m}$. Same data as in Fig. 28. Dashed line is $v\sqrt{m} = 1/\gamma$.

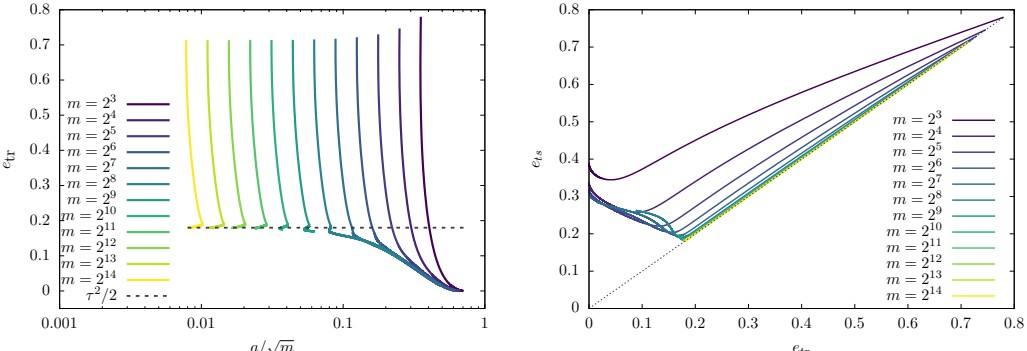

Figure 30: **Train and test error of gradient flow under mean field initialization, for increasing values of** $m$. Left: train error as a function of rescaled weights $a(t)/\sqrt{m}$. Dashed line is the Bayes error $\tau^2/2$. Curves are traversed in time from top to bottom. Right: test error versus train error. Curves are traversed in time from right to left. Here $\hat{\varphi}(z) = h(z) = (9/10)z + z^3/6$ with $\tau = 0.6$ and $\alpha = 0.3$.

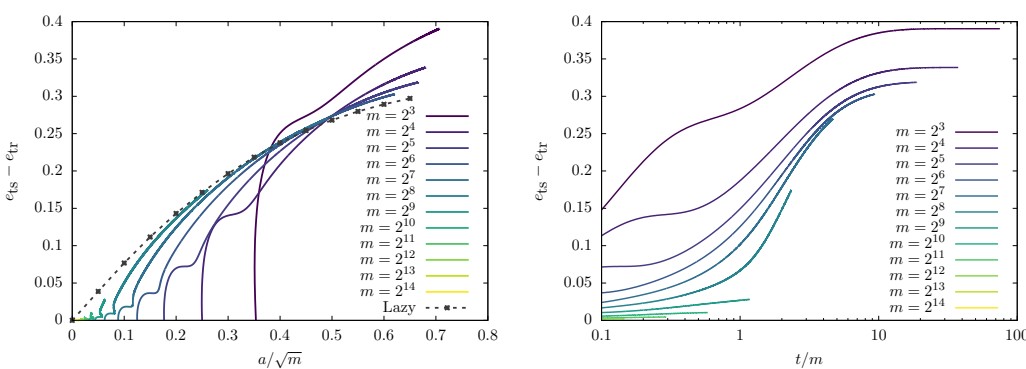

Figure 31: **The difference between test and train error for the single index data.** Left panel: the difference between test and train error plotted as a function of $a/\sqrt{m}$ and compared to what is obtained from a model with fixed second layer weights initialized with Lazy scaling. Right panel: the difference between test and train error on timescales of order $m$. Here, $\hat{\varphi}(z) = h(z) = (9/10)z + z^3/6$ with $\tau = 0.6$ and $\alpha = 0.3$.

The above analysis implies that (considering to be definite second layer weights, and projection of first layer weigths onto the latent direction), for $m \gg 1$, $t \gg 1$,

$$a(t) = a^{\text{mfl}}(t) + \frac{1}{m}\left(a_* t + o(t)\right) + \frac{1}{m}\Delta_a(t, m), \tag{G.77}$$

$$v(t) = v^{\text{mfl}}(t) + \frac{1}{m}\left(v_* t + o(t)\right) + \frac{1}{m}\Delta_v(t, m), \tag{G.78}$$

where $\lim_{m\to\infty} \Delta_{a/v}(t, m) = 0$. If we neglect the error terms, and assume that this expression holds for $t$ larger than $O(1)$ in $m$, then it indicates that $a(t)$, $v(t)$ differ significantly from the mean field prediction when $t/m$ becomes of order one. We expect therefore a third dynamical regime for $t = \Theta(m)$, which will be the object of the next section.

### G.2.3 Second dynamical regime: $t = \Theta(m)$ and beyond

As pointed out at the end of the previous section, we expect a third dynamical regime when $t = \Theta(m)$. By this time, the stability calculation in the previous section indicates that second layer weights become of order $\sqrt{m}$. Figure 28 confirms this, and shows that, in the same regime $v(t)$ becomes small. In fact, numerical solution of the SymmDMFT equations are consistent with $a(t) = \Theta(\sqrt{m})$, $v(t) = \Theta(1/\sqrt{m})$, and $a(t)v(t) \approx 1$ for $t = \Theta(m)$.

For a small constant $c$ denote by $t_0(m; c)$ the time at which $a(t_0(m; c)) = c\sqrt{m}$. We then expect that the following exists

$$\lim_{m\to\infty} \frac{a(t_0(m; c) + \theta\, w(m))}{\sqrt{m}} = \gamma^{\text{mf3}}(\theta), \tag{G.79}$$

$$\lim_{m\to\infty} \boldsymbol{v}(t_0(m; c) + \theta\, w(m))\sqrt{m} = \boldsymbol{v}^+(\theta), \tag{G.80}$$

provided $w(m)$ is a suitable function (with $w(m) = O(t_0(m; c))$). The stability analysis in the previous section suggests that $t_0(m; c) \le t_*(c)m + o(m)$. Our numerical solutions do not cover a large enough range of values of $m$ to verify this ansatz, and determine the scaling of $w(m)$ with $m$. On the other hand, they indicate that indeed $t_0(m; c) = \Theta(m)$.

Since the second layer weights become of order $\sqrt{m}$ in this dynamical regime, train and test error start to differ significantly. We expect

$$\lim_{m\to\infty} e_{\text{tr}}(t_0(m; c) + \theta\, w(m)) = e_{\text{tr}}^{\text{mf3}}(\theta), \tag{G.81}$$

$$\lim_{m\to\infty} e_{\text{tr}}(t_0(m; c) + \theta\, w(m)) = e_{\text{ts}}^{\text{mf3}}(\theta). \tag{G.82}$$

This picture is confirmed by Fig. 30, which reports train and test error as predicted by numerical solutions of the SymmDMFT equations for increasing values of $m$. On the left, we plot the train error as a function of the rescaled second layer weights $\gamma = a/\sqrt{m}$. We observe that curves for different values of $m$ decrease until they reach the Bayes error $\tau^2$. On this phase however different curves do not collapse corresponding to the fact that $\gamma$ vanishes. In the second phase, $\gamma$ grows to be of order one and correspondingly the train error decreases below the Bayes error: this is the third dynamical regime. Overfitting takes place at this point.

In the right frame of Fig. 30, we plot test error versus train error. We observe, again, the two phases emerging for large $m$. In the first phase train error and test error are closely matched. In the second phase, train error decreases and test error correspondingly increases. Again, this takes place when $t = \Theta(m)$.

Finally, in Fig. 31, we repeat similar plots for the generalization error (difference between test and train error).

When $t/m$ is large, the train error vanishes. We observe from Figure 28, left frame that, as $t \to \infty$, rescaled second layer weights reach a finite limit that is close to the interpolation threshold characterized in Section F.2.3. Namely

$$\lim_{\tau\to\infty} \gamma^+(\theta) \approx \gamma_{\text{GF}}^*(\alpha, \varphi, \tau). \tag{G.83}$$

---

[1]We note that solving the SymmDMFT equations accurately enough to capture these corrections requires either to use very fine discretization, or a higher-order integration method.

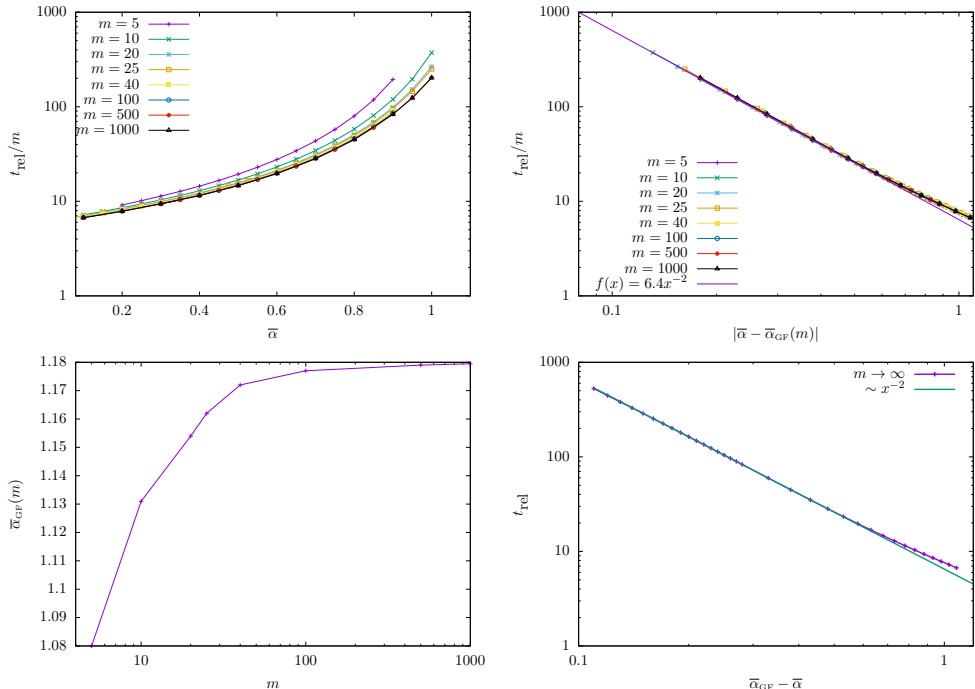

Figure 32: **The interpolation transition for pure noise data** and a network with second layer weights that do not evolve with time, fixed at $a = 1$, see Section H. The noise level is fixed to $\tau = 1$ and we considered $h(z) = (3/10)z + z^2/2$. Top left panel: relaxation time ((rate for convergence to vanishing error) for different values of $m$. Top right panel: logarithmic plot of the relaxation time. The value of the algorithmic threshold for different values of $m$ is a fitting parameter. Bottom left panel: values of the algorithmic thresholds as a function of $m$. Bottom right panel: the relaxation time as extracted from the scaling limit of the SymmDMFT equations in the $m \to \infty$ limit. The algorithmic threshold is in this case $\overline{\alpha}_{\mathrm{GF}}(\infty) \approx 1.18$ which fits well the behavior plotted in the left bottom plot.

## H   Dynamics under mean field initialization for $n/d = \overline{\alpha}$ fixed

### H.1   Interpolation threshold at fixed $a(t) = a_0$

In this section, we consider an alternative scaling in the large width limit. As before, we use the SymmDMFT equations, and therefore study the limit $n, d \to \infty$ with $n/d \to \overline{\alpha}$. In the previous sections we studied the large width limit $m \to \infty$ with $\alpha = \overline{\alpha}/m$ fixed. In that setting interpolation is only possible when the network complexity scales, i.e. second-layer weights are $a = \Theta(\sqrt{m})$

Here instead we keep $a(t) = 1$ and do not let evolve second-layer weights with GF. We consider pure noise data, and show that interpolation takes place if $\overline{\alpha} < \overline{\alpha}_{\mathrm{GF}}(m)$, while the train error remains bounded away from zero for $\overline{\alpha} > \overline{\alpha}_{\mathrm{GF}}(m)$. As expected from Gaussian complexity considerations, the threshold $\overline{\alpha}_{\mathrm{GF}}(m)$ has a finite limit as $m \to \infty$. In particular, for any $\alpha > 0$, a network with $a$ bounded cannot interpolate pure noise data.

As thorough in Sec.F we fix $\overline{\alpha}$ and integrate numerically the SymmDMFT equations for finite but increasing values of $m$. We fix the initialization scale $a_0$ and the noise level $\tau$ and change only $\overline{\alpha}$.

We observe that for $\overline{\alpha}$ small enough the train error decreases exponentially fast to zero. Namely, recalling that $e_{\mathrm{tr}}(t; \overline{\alpha}) := \lim_{n,d \to \infty} \widehat{\mathscr{R}}_n(\boldsymbol{a}(t), \boldsymbol{W}(t))$, we have that

$$\overline{\alpha} < \overline{\alpha}_{\mathrm{GF}}(m) \quad \Rightarrow \quad e_{\mathrm{tr}}(t; \overline{\alpha}) = \exp\{-t/t^*_{\mathrm{rel}}(\overline{\alpha}, m) + o(t)\}. \tag{H.1}$$

However, the relaxation time time $t^*_{\mathrm{rel}}(\overline{\alpha}, m)$ increases as $\overline{\alpha} \uparrow \overline{\alpha}_{\mathrm{GF}}(m)$. Concretely, we define $t_{\mathrm{rel}}(\overline{\alpha}, m, c)$ as the infimum time such that $e_{\mathrm{tr}}(t; \overline{\alpha}) \leq c$, where $c$ is some small constant. In practice, we set $c = 10^{-7}$. The results are plotted as a function of $\overline{\alpha}$ for several values of $m$ in Fig. 32, top left plot.

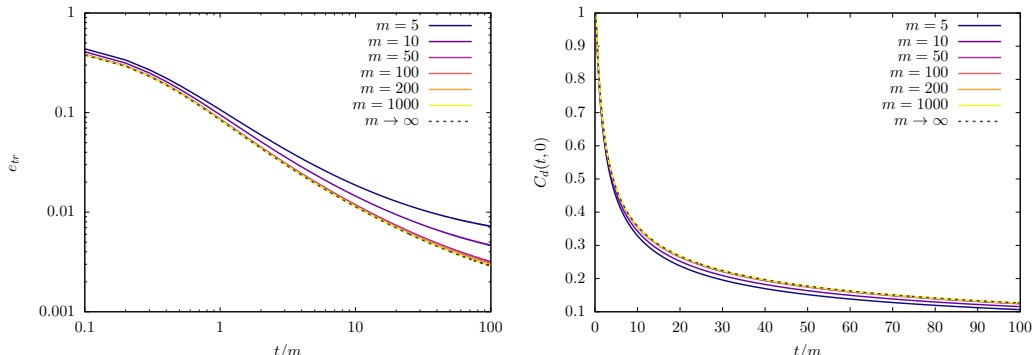

Figure 33: **heck of the convergence of the numerical solution of the** SymmDMFT **for** $\overline{\alpha}$ **fixed to the scaling solution for** $m \to \infty$**.** The left panel shows the behavior of the train error while the right panel shows the behavior of the correlation $C_d(t,0)$. Both panels refer to a model where the teacher is pure noise with $\tau = 1$ and the student is made of of neurons whose covariance structure is given by $h(z) = (3/10)z + z^2/2$.

For each value of $m$ the relaxation time appears to diverge at the critical point $\overline{\alpha}_{\mathrm{GF}}(m)$ as an inverse power of $\overline{\alpha}_{\mathrm{GF}}(m) - \overline{\alpha}$, namely:

$$\overline{\alpha} \uparrow \overline{\alpha}_{\mathrm{GF}}(m) \quad \Rightarrow \quad t_{\mathrm{rel}}(\overline{\alpha}, m, c) = \frac{L(m,c)}{(\overline{\alpha}_{\mathrm{GF}}(m) - \overline{\alpha})^\nu} \left(1 + o(1)\right). \tag{H.2}$$

The exponent $\nu$ appears to be independent of $m$. We fit this form to our data and extract the interpolation thresholds $\overline{\alpha}_{\mathrm{rel}}(m)$. In Fig. 32, top right, we plot $t_{\mathrm{rel}}(\overline{\alpha}, m, c)/m$ as a function of the gap to this threshold. This plot confirms the form (H.2), with exponent $\nu \approx 2$. Also, the fact that different curves superimpose indicate that $L(m,c) \approx L_*(c)m$.

The estimated interpolation thresholds $\overline{\alpha}_{\mathrm{GF}}(m)$ are plotted as a function of $m$ in the bottom left of Fig. 32. These data are consistent with the existence of a finite limit

$$\overline{\alpha}_{\mathrm{GF}}(\infty) = \lim_{m \to \infty} \overline{\alpha}_{\mathrm{GF}}(m), \tag{H.3}$$

and numerically $\overline{\alpha}_{\mathrm{GF}}(\infty) \approx 1.18$.

In the next subsection, we derive equations describing the $m \to \infty$ limit for $\overline{\alpha} = O(1)$, $a = O(1)$ fixed. Studying these equations yields further support to Eq. (H.3).

### H.2   Infinite width limit at fixed $\overline{\alpha}$

In order to study the limit $m \to \infty$ at fixed $\overline{\alpha}$, we discuss the limit of the SymmDMFT equations when $m \to \infty$. As we have seen previously, the relaxation time of the train error is proportional to $m$. This is clearly visible in Fig. 32-top/left. This suggests that for $m \to \infty$, dynamics takes place on timescales of order $m$. Therefore we propose the following asymptotic ansatz

$$mR_o(tm, sm) = \tilde{R}_o^{\overline{\alpha}}(t,s) + o_m(1), \qquad C_o(tm, sm) = \tilde{C}_o^{\overline{\alpha}}(t,s) + o_m(1), \tag{H.4}$$

$$R_d(tm, sm) = \tilde{R}_d^{\overline{\alpha}}(t,s) + o_m(1), \qquad C_d(tm, sm) = \tilde{C}_d^{\overline{\alpha}}(t,s) + o_m(1), \tag{H.5}$$

$$m\nu(tm) = \tilde{\nu}^{\overline{\alpha}}(t) + o_m(1), \tag{H.6}$$

which defines a set of functions, $\tilde{R}_d^{\overline{\alpha}}, \tilde{C}_d^{\overline{\alpha}}, \tilde{R}_o^{\overline{\alpha}}, \tilde{C}_o^{\overline{\alpha}}$ and $\tilde{\nu}^{\overline{\alpha}}$. We now describe the equations that these scaling functions satisfy satisfy. First we define $\tilde{C}_A^{\overline{\alpha}}$ and $\tilde{R}_A^{\overline{\alpha}}$ as the solution of

$$\delta(t - t') = \int_{t'}^{t} \left[\delta(t-s) + \tilde{\Sigma}_R(t,s)\right] \tilde{R}_A^{\overline{\alpha}}(s, t'\mathrm{d}s, )$$

$$0 = \int_0^t \left[\delta(t-s) + \tilde{\Sigma}_R(t,s)\right] \tilde{C}_A^{\overline{\alpha}}(t',s) \, \mathrm{d}s + \int_0^{t'} \mathrm{d}s \tilde{\Sigma}_C(t,s) \tilde{R}_A^{\overline{\alpha}}(t',s) \, \mathrm{d}s, \tag{H.7}$$

where

$$\tilde{\Sigma}_R(t,s) = h'(\tilde{C}_d^{\overline{\alpha}}(t,s))\tilde{R}_d^{\overline{\alpha}}(t,s) + h'(\tilde{C}_o^{\overline{\alpha}}(t,s))\tilde{R}_o^{\overline{\alpha}}(t,s)$$
$$\tilde{\Sigma}_C(t,s) = \tau^2 + h(\tilde{C}_o^{\overline{\alpha}}(t,s)).$$
(H.8)

Then we define the limit memory kernels:

$$\tilde{M}_R^{(d)}(t,s) = \overline{\alpha}\tilde{C}_A(t,s)h''(\tilde{C}_d^{\overline{\alpha}}(t,s))\tilde{R}_d^{\overline{\alpha}}(t,s),$$
$$\tilde{M}_C^{(d)}(t,s) = \overline{\alpha}\tilde{C}_A^{\overline{\alpha}}(t,s)h'(\tilde{C}_d^{\overline{\alpha}}(t,s)),$$
$$\tilde{M}_R^{(o)}(t,s) = \overline{\alpha}\left[\tilde{C}_A(t,s)h''(\tilde{C}_o^{\overline{\alpha}}(t,s))\tilde{R}_o^{\overline{\alpha}}(t,s) + \tilde{R}_A^{\overline{\alpha}}(t,s)h'(\tilde{C}_o^{\overline{\alpha}}(t,s))\right],$$
$$\tilde{M}_C^{(o)}(t,s) = \overline{\alpha}\tilde{C}_A^{\overline{\alpha}}(t,s)h'(\tilde{C}_o^{\overline{\alpha}}(t,s)).$$
(H.9)

Substituting the above ansatz in the SymmDMFT equations and matching the leading order terms, we get the following equations that determine $\tilde{R}_d^{\overline{\alpha}}$, $\tilde{C}_d^{\overline{\alpha}}$, $\tilde{R}_o^{\overline{\alpha}}$, $\tilde{C}_o^{\overline{\alpha}}$ and $\tilde{\nu}^{\overline{\alpha}}$:

$$\partial_t \tilde{C}_d^{\overline{\alpha}}(t,t') = -\tilde{\nu}^{\overline{\alpha}}(t)\tilde{C}_d^{\overline{\alpha}}(t,t') - \int_0^t \left[\tilde{M}_R^{(d)}(t,s)\tilde{C}_d^{\overline{\alpha}}(t',s) + \tilde{M}_R^{(o)}(t,s)\tilde{C}_o^{\overline{\alpha}}(t',s)\right]\,\mathrm{d}s$$
$$-\int_0^{t'} \left[\tilde{M}_C^{(d)}(t,s)\tilde{R}_d^{\overline{\alpha}}(t',s) + \tilde{M}_C^{(o)}(t,s)\tilde{R}_o^{\overline{\alpha}}(t',s)\right]\,\mathrm{d}s,$$
(H.10)

$$\partial_t \tilde{C}_o^{\overline{\alpha}}(t,t') = -\tilde{\nu}^{\overline{\alpha}}(t)\tilde{C}_o^{\overline{\alpha}}(t,t') - \int_0^t \left[\tilde{M}_R^{(d)}(t,s) + \tilde{M}_R^{(o)}(t,s)\right]\tilde{C}_o^{\overline{\alpha}}(t',s)\,\mathrm{d}s$$
$$-\int_0^{t'} \left[\tilde{R}_o^{\overline{\alpha}}(t',s) + \tilde{R}_d^{\overline{\alpha}}(t',s)\right]\tilde{M}_C^{(o)}(t,s)\,\mathrm{d}s,$$
(H.11)

$$\partial_t \tilde{R}_d^{\overline{\alpha}}(t,t') = -\tilde{\nu}^{\overline{\alpha}}(t)\tilde{R}_d^{\overline{\alpha}}(t,t') + \delta(t-t') - \int_{t'}^t \mathrm{d}s\,\tilde{M}_R^{(d)}(t,s)\tilde{R}_d^{\overline{\alpha}}(s,t'),$$
(H.12)

$$\partial_t \tilde{R}_o^{\overline{\alpha}}(t,t') = -\tilde{\nu}^{\overline{\alpha}}(t)\tilde{R}_o^{\overline{\alpha}}(t,t') - \int_{t'}^t \left[\tilde{M}_R^{(d)}(t,s)\tilde{R}_o^{\overline{\alpha}}(s,t') + \tilde{M}_R^{(o)}(t,s)\tilde{R}_d^{\overline{\alpha}}(s,t')\right.$$
$$\left.+\tilde{M}_R^{(o)}(t,s)\tilde{R}_o^{\overline{\alpha}}(s,t')\right]\,\mathrm{d}s,$$
(H.13)

$$\tilde{\nu}^{\overline{\alpha}}(t) = -\int_0^t \mathrm{d}s\left[\tilde{M}_R^{(d)}(t,s)\tilde{C}_d(t,s) + \tilde{M}_R^{(o)}(t,s)\tilde{C}_o^{\overline{\alpha}}(t,s)\right]\,\mathrm{d}s$$
$$-\int_0^t \left[\tilde{M}_C^{(d)}(t,s)\tilde{R}_d^{\overline{\alpha}}(t,s) + \tilde{M}_C^{(o)}(t,s)\tilde{R}_o^{\overline{\alpha}}(t,s)\right]\,\mathrm{d}s.$$
(H.14)

These are to be solved with boundary condition

$$\tilde{C}_o^{\overline{\alpha}}(0,0) = 0,\qquad\qquad\qquad \tilde{R}_o^{\overline{\alpha}}(0,0) = 0,\qquad\qquad (\text{H.15})$$
$$\tilde{C}_d^{\overline{\alpha}}(0,0) = 1,\qquad\qquad\qquad \tilde{R}_d^{\overline{\alpha}}(0^+,0) = 1.\qquad\qquad (\text{H.16})$$

The scaling behavior of the train error is then given by

$$\lim_{m\to\infty} e_{\mathrm{tr}}(t) = -\frac{1}{2}\tilde{C}_A^{\overline{\alpha}}(t,t) =: e_{\mathrm{tr}}^{\overline{\alpha}}(t).$$
(H.17)

In order to test the accuracy of the asymptotic analysis developed in this sections, we solved numerically the SymmDMFT equations for increasing values of $m$ and compare the results to the numerical integration of Eqs. (H.10), (H.14) presented in this section. Some results of this comparison are presented in Fig. 33, which shows good agreement between finite-$m$ curves and $m \to \infty$ limit.

The solution of Eqs. (H.10), (H.14) provides another route to estimate the large-$m$ interpolation threshold $\overline{\alpha}_{\mathrm{GF}}(\infty)$ at fixed $a(t) = 1$. Namely, we solve the equations numerically and extract the $t_{\mathrm{rel}}(\overline{\alpha}, \infty, c)$, which is defined analogously to above. We then fit the divergence of $t_{\mathrm{rel}}(\overline{\alpha}, \infty, c)$ at $\overline{\alpha}_{\mathrm{GF}}(\infty)$ according to Eq. (H.2). We obtain $\overline{\alpha}_{\mathrm{GF}}(\infty) \approx 1.18$, in agreement with the threshold obtained by extrapolating the finite-$m$ thresholds $\overline{\alpha}_{\mathrm{GF}}(m)$. In the bottom right plot of Fig. 32 we plot $t_{\mathrm{rel}}(\overline{\alpha}, \infty, c)$ as function of $\overline{\alpha}_{\mathrm{GF}}(\infty) - \overline{\alpha}$. This confirms the behavior of Eq. (H.2) with $\nu \approx 2$.

We conclude by emphasizing that, throughout this section $\alpha(t) = 1$ and $\tau = 1$ were fixed. If we generalize to arbitrary $\alpha(t) = a_0$ and arbitrary $\tau > 0$, the threshold $\overline{\alpha}_{\mathrm{GF}}(m)$ will of course on these quantities through the ratio $a_0/\tau$.

# I Details about SGD simulations

In this appendix we provide some details about the numerical simulations with stochastic gradient descent (SGD) presented in Figures 2, 4.

We generate data according to the pure noise model $y_i = \varepsilon_i$ (Fig. 2), $y_i = \varphi(\boldsymbol{w}_*^\mathsf{T}\boldsymbol{x}_i) + \varepsilon_i$ (Fig. 4), $i \leq n$. We learn the two-layer network of Eq. (1.1), see below for the class definition.

```
class Net(nn.Module):
    def __init__(self, a, m, d):
        super().__init__()
        self.m = m
        self.lin1 = nn.Linear(d,m,bias=False)
        self.lin1.weight.data = (1/np.sqrt(d))*torch.randn((m,d))
        self.lin2 = nn.Linear(m,1,bias=False)
        self.lin2.weight.data[0,:] = a
        self.act = Myact()
        self.project()
    def forward(self, x ):
        x1 = self.act(self.lin1(x))
        return self.lin2(x1)/self.m
    def project(self, epsilon):
        row_norms = torch.norm(self.lin1.weight.data, dim=1, keepdim=
            True)
        row_norms = torch.clamp(row_norms, min=epsilon)
        self.lin1.weight.data = self.lin1.weight.data/row_norms
```

As shown in this code, we use the initialization

$$\big(\boldsymbol{a}_0, \boldsymbol{W}_0\big) = \boldsymbol{P}_\mathsf{B}\big(\overline{\boldsymbol{a}}_0, \overline{\boldsymbol{W}}_0\big) , \tag{I.1}$$

$$\overline{\boldsymbol{a}}_0 = (a_0, \ldots, a_0) , \quad (W_{0,ij})_{i \leq m, j \leq d} \sim \mathsf{N}(0, 1/d) . \tag{I.2}$$

where $\boldsymbol{P}_\mathsf{B}$ projects first layer weights to the unit ball:

$$\boldsymbol{P}_\mathsf{B}\Big(\boldsymbol{a}, (\boldsymbol{w}_1, \ldots, \boldsymbol{w}_m)\Big) = \Big(\boldsymbol{a}, \Big(\frac{\boldsymbol{w}_1}{\|\boldsymbol{w}_1\| \wedge 1}, \ldots, \frac{\boldsymbol{w}_m}{\|\boldsymbol{w}_m\| \wedge 1}\Big)\Big) . \tag{I.3}$$

We use the standard SGD iteration without weight decay and constant stepsize $\eta$, and batch size $b$:

$$\overline{\boldsymbol{\theta}}_{k+1} = \boldsymbol{\theta}_k - \eta \nabla \widehat{\mathscr{R}}_{S(k)}(\boldsymbol{\theta}_k) , \quad \widehat{\mathscr{R}}_S(\boldsymbol{\theta}) = \frac{1}{2|S|} \sum_{i \in S} \big(y_i - f(\boldsymbol{x}_i; \boldsymbol{\theta})\big)^2 , \tag{I.4}$$

$$\boldsymbol{\theta}_{k+1} = \boldsymbol{P}_\mathsf{B}(\overline{\boldsymbol{\theta}}_{k+1}) . \tag{I.5}$$

The optimizer is defined in the code below

```
optimizer = optim.SGD(net.parameters(), lr=lr, momentum=0.,
    weight_decay=0.)
lambda_step = lambda epoch: 1
scheduler = torch.optim.lr_scheduler.LambdaLR(optimizer, lr_lambda=
    lambda_step)
```

In the simulations of Figures 2, and 4 we use batch size $b = 100$ and step size $\eta = 0.1$. Each symbol reports the average of $N_{\text{sim}} = 10$ simulations.

# J Lower bounding the overfitting timescale

Throughout this appendix we use $t$ to denote the rescaled time $\hat{t}$ introduced in Section 3.

## J.1 Proof of Theorem 3.1

By computing the derivative $\partial_{a_i}\widehat{\mathscr{R}}_n(\boldsymbol{a}(t), \boldsymbol{W}(t))$, we get

$$\frac{\mathrm{d}}{\mathrm{d}t}\big|a_\ell(t)\big| \leq \left|\frac{1}{n}\sum_{i=1}^n \big(y_i - f(\boldsymbol{x}_i; \boldsymbol{a}(t), \boldsymbol{W}(t))\sigma(\boldsymbol{w}_\ell(t)^\mathsf{T}\boldsymbol{x}_i)\right|$$

$$\leq \sqrt{2\widehat{\mathscr{R}}_n(\boldsymbol{a}(t), \boldsymbol{W}(t))} \cdot \sqrt{\frac{1}{n}\sum_{i=1}^{n}\sigma(\boldsymbol{w}_\ell^\mathsf{T}\boldsymbol{x}_i)^2}$$

$$\leq 4L\sqrt{2\widehat{\mathscr{R}}_n(\boldsymbol{a}(0), \boldsymbol{W}(0))} \cdot \sqrt{\frac{1}{n}\sum_{i=1}^{n}(1 + (\boldsymbol{w}_\ell(t)^\mathsf{T}\boldsymbol{x}_i)^2)}$$

$$\leq 10L\sqrt{2\widehat{\mathscr{R}}_n(\boldsymbol{a}(0), \boldsymbol{W}(0))}\,,$$

where, for $n \geq d$, the last inequality holds with probability at least $1 - 2\exp(-cn)$ (for some universal $c > 0$) by standard upper bounds on the norm of random matrices [53]. Further

$$\sqrt{2n\widehat{\mathscr{R}}_n(\boldsymbol{a}(0), \boldsymbol{W}(0))} = \left\|\boldsymbol{y} - \frac{1}{m}\sum_{i=1}^{m}a_i\sigma(\boldsymbol{X}\boldsymbol{w}_i)\right\|$$

$$\stackrel{(a)}{\leq} \|\boldsymbol{y}\| + a_0 \max_{\ell \leq m}\left\|\sigma(\boldsymbol{X}\boldsymbol{w}_\ell(0))\right\|$$

$$\stackrel{(b)}{\leq} \tau\|\boldsymbol{g}\| + \|\varphi(\boldsymbol{X}\boldsymbol{U})\| + a_0 \max_{\ell \leq m}\left\|\sigma(\boldsymbol{X}\boldsymbol{w}_\ell(0))\right\|$$

$$\leq \tau\|\boldsymbol{g}\| + L\|\boldsymbol{X}\boldsymbol{U}\| + \sqrt{n}|\varphi(0)| + a_0 L\|\boldsymbol{X}\|_{\mathrm{op}} + a_0\sqrt{n}|\sigma(0)|\,,$$

where in $(a)$ it is understood that $\sigma$ is applied entrywise to $\boldsymbol{X}\boldsymbol{w}_i \in \mathbb{R}^n$ and in $(b)$ we have $\boldsymbol{g} \sim \mathsf{N}(\boldsymbol{0}, \boldsymbol{I}_n)$, and $\varphi$ is applied row-wise to $\boldsymbol{X}\boldsymbol{U} \in \mathbb{R}^{n \times k}$. By using standard concentration on the norm of random matrices, also with probability $1 - \exp(-cn)$, we have (for $m \leq n$)

$$\sqrt{2\widehat{\mathscr{R}}_n(\boldsymbol{a}(0), \boldsymbol{W}(0))} = C(\tau + \sqrt{k} + a_0 L)\,.$$

Summarizing the above bounds, we have

$$\frac{\mathrm{d}}{\mathrm{d}t}|a_\ell(t)| \leq a_1\,,$$

which implies the first claim by integration.

To prove the second claim, we consider the following sets of parameters $\mathcal{W}_{m,d}^\infty(\overline{a}) \subseteq \mathcal{W}_{m,d}(\overline{a})$ (which will also prove useful in the next section)

$$\mathcal{W}_{m,d}(\overline{a}) := \left\{(\boldsymbol{a}, \boldsymbol{W}) \in \mathbb{R}^m \times \mathbb{R}^{m \times d} : \frac{\|\boldsymbol{a}\|_1}{m} \leq \overline{a},\ \|\boldsymbol{w}_i\|_2 = 1\ \forall i \leq m\right\}, \qquad \text{(J.1)}$$

$$\mathcal{W}_{m,d}^\infty(\overline{a}) := \left\{(\boldsymbol{a}, \boldsymbol{W}) \in \mathbb{R}^m \times \mathbb{R}^{m \times d} : \|\boldsymbol{a}\|_\infty \leq \overline{a},\ \|\boldsymbol{w}_i\|_2 = 1\ \forall i \leq m\right\}. \qquad \text{(J.2)}$$

The second claim follows in turn if we prove that there exists a universal constant $C$ such that

$$\sup_{(\boldsymbol{a}, \boldsymbol{W}) \in \mathcal{W}_{m,d}(\overline{a})}\left|\widehat{\mathscr{R}}_n(\boldsymbol{a}, \boldsymbol{W}) - \mathscr{R}(\boldsymbol{a}, \boldsymbol{W})\right| \leq C(L^2\overline{a}^2 + \tau^2)\sqrt{\frac{d}{n}}\,. \qquad \text{(J.3)}$$

This is a standard estimate, that we reproduce for the readers' convenience.

We begin by bounding the expectation of the supremum by symmetrization and contraction inequalities. Letting $(\xi_i)_{i \leq n} \sim_{iid} \mathsf{Unif}(\{+1, -1\})$, we have

$$\mathbb{E}\sup_{(\boldsymbol{a}, \boldsymbol{W}) \in \mathcal{W}_{m,d}(\overline{a})}\left|\widehat{\mathscr{R}}_n(\boldsymbol{a}, \boldsymbol{W}) - \mathscr{R}(\boldsymbol{a}, \boldsymbol{W})\right| \leq \mathbb{E}\sup_{(\boldsymbol{a}, \boldsymbol{W}) \in \mathcal{W}_{m,d}(\overline{a})}\frac{1}{n}\sum_{i=1}^{n}\xi_i\big(y_i - f(\boldsymbol{x}_i; \boldsymbol{a}, \boldsymbol{W})\big)^2$$

$$\leq 2\mathbb{E}\sup_{(\boldsymbol{a}, \boldsymbol{W}) \in \mathcal{W}_{m,d}(\overline{a})}\frac{1}{n}\sum_{i=1}^{n}\xi_i y_i f(\boldsymbol{x}_i; \boldsymbol{a}, \boldsymbol{W}) + \mathbb{E}\sup_{(\boldsymbol{a}, \boldsymbol{W}) \in \mathcal{W}_{m,d}(\overline{a})}\frac{1}{n}\sum_{i=1}^{n}\xi_i f(\boldsymbol{x}_i; \boldsymbol{a}, \boldsymbol{W})^2$$

$$=: 2E_1 + E_2\,.$$

We begin by bounding $E_1$:

$$E_1 = \mathbb{E} \sup_{(\boldsymbol{a},\boldsymbol{W}) \in \mathcal{W}_{m,d}(\overline{a})} \sum_{j=1}^{m} \frac{a_j}{m} \frac{1}{n} \sum_{i=1}^{n} \xi_i y_i \sigma(\boldsymbol{w}_j^{\mathsf{T}} \boldsymbol{x}_i)$$

$$\leq \overline{a}\, \mathbb{E} \sup_{\|\boldsymbol{w}\|=1} \frac{1}{n} \sum_{i=1}^{n} \xi_i y_i \sigma(\boldsymbol{w}^{\mathsf{T}} \boldsymbol{x}_i)$$

$$\overset{(a)}{\leq} \overline{a}\, L \mathbb{E} \sup_{\|\boldsymbol{w}\|=1} \frac{1}{n} \sum_{i=1}^{n} \xi_i (1+|y_i|) \boldsymbol{w}^{\mathsf{T}} \boldsymbol{x}_i$$

$$\leq \overline{a}\, L \mathbb{E} \Big\{ \Big\| \frac{1}{n} \sum_{i=1}^{n} \xi_i (1+|y_i|) \boldsymbol{x}_i \Big\| \Big\}$$

$$\leq C \overline{a}\, L (L+\tau) \sqrt{\frac{d}{n}}\,,$$

where in $(a)$ we applied the contraction inequality of [36] to the function $\psi_i(t) = y_i \sigma(t/(|y_i|+1))$. We next bound term $E_2$:

$$E_2 = \mathbb{E} \sup_{(\boldsymbol{a},\boldsymbol{W}) \in \mathcal{W}_{m,d}(\overline{a})} \sum_{j,l=1}^{m} \frac{a_j a_l}{m^2} \frac{1}{n} \sum_{i=1}^{n} \xi_i \sigma(\boldsymbol{w}_j^{\mathsf{T}} \boldsymbol{x}_i) \sigma(\boldsymbol{w}_l^{\mathsf{T}} \boldsymbol{x}_i)$$

$$\leq \overline{a}^2 \mathbb{E} \sup_{\boldsymbol{w},\tilde{\boldsymbol{w}} \in \mathbb{S}^{d-1}} \frac{1}{n} \sum_{i=1}^{n} \xi_i \sigma(\boldsymbol{w}^{\mathsf{T}} \boldsymbol{x}_i) \sigma(\tilde{\boldsymbol{w}}^{\mathsf{T}} \boldsymbol{x}_i)$$

$$\overset{(b)}{\leq} C L^2 \overline{a}^2 \mathbb{E} \sup_{\boldsymbol{w} \in \mathbb{S}^{d-1}} \frac{1}{n} \sum_{i=1}^{n} \xi_i \boldsymbol{w}^{\mathsf{T}} \boldsymbol{x}_i$$

$$\leq C L^2 \overline{a}^2 \sqrt{\frac{d}{n}}\,,$$

where inequality $(b)$ follows by applying the contraction inequality of [36] to $\psi(t_1,t_2) = \sigma(t_1)\sigma(t_2)$ which is $CL^2$-Lipschitz because $\|\sigma\|_{\mathrm{Lip}}, \|\sigma\|_\infty \leq L$.

Summarizing, we proved that

$$\mathbb{E} \sup_{(\boldsymbol{a},\boldsymbol{W}) \in \mathcal{W}_{m,d}(\overline{a})} \big| \widehat{\mathscr{R}}_n(\boldsymbol{a},\boldsymbol{W}) - \mathscr{R}(\boldsymbol{a},\boldsymbol{W}) \big| \leq C(L^2 \overline{a}^2 + \tau^2) \sqrt{\frac{d}{n}}\,. \tag{J.4}$$

In order to complete the proof of Eq. (J.3), we will show that the supremum concentrates around its expectation. For fixed $(\boldsymbol{a},\boldsymbol{W}) \in \mathcal{W}_{m,d}(\overline{a})$, we have

$$\big| f(\boldsymbol{x};\boldsymbol{a},\boldsymbol{W}) - f(\boldsymbol{x}';\boldsymbol{a},\boldsymbol{W}) \big| \leq L\overline{a} \|\boldsymbol{x} - \boldsymbol{x}'\|_2\,,$$

$$\big| \varphi(\boldsymbol{U}^{\mathsf{T}} \boldsymbol{x}) - \varphi(\boldsymbol{U}^{\mathsf{T}} \boldsymbol{x}') \big\| \leq L \|\boldsymbol{x} - \boldsymbol{x}'\|_2\,.$$

We write $\widehat{\mathscr{R}}_n(\boldsymbol{X};\boldsymbol{a},\boldsymbol{W})$ to emphasize the dependence of the risk on $\boldsymbol{X}$ Letting $r(\boldsymbol{x}_i;\boldsymbol{a},\boldsymbol{W}) = \varphi(\boldsymbol{U}^{\mathsf{T}}\boldsymbol{x}) - f(\boldsymbol{x};\boldsymbol{a},\boldsymbol{W})$, we have

$$\nabla_{\boldsymbol{x}_i} \widehat{\mathscr{R}}_n(\boldsymbol{X};\boldsymbol{a},\boldsymbol{W}) = \frac{1}{n}\big(\varepsilon_i + r(\boldsymbol{x}_i;\boldsymbol{a},\boldsymbol{W})\big) \nabla_{\boldsymbol{x}_i} r(\boldsymbol{x}_i;\boldsymbol{a},\boldsymbol{W})\,,$$

$$\Rightarrow \big\| \nabla_{\boldsymbol{x}_i} \widehat{\mathscr{R}}_n(\boldsymbol{X};\boldsymbol{a},\boldsymbol{W}) \big\| \leq \frac{C}{n}(|\varepsilon_i| + L\overline{a})L\overline{a}\,.$$

Hence

$$\big\| \nabla_{\boldsymbol{X}} \widehat{\mathscr{R}}_n(\boldsymbol{X};\boldsymbol{a},\boldsymbol{W}) \big\| \leq \frac{C}{\sqrt{n}} L\overline{a}\Big( L\overline{a} + \frac{\|\varepsilon\|}{\sqrt{n}} \Big)$$

$$\leq \frac{C'}{\sqrt{n}} L\overline{a}(L\overline{a} + \tau)\,,$$

where the last inequality holds on an event that has probability at least $1-e^{-n}$. Defining $Z_{n,d,m}(\overline{a}) :=$ $\sup_{(\boldsymbol{a},\boldsymbol{W})\in\mathcal{W}_{m,d}(\overline{a})}\big|\widehat{\mathscr{R}}_n(\boldsymbol{a},\boldsymbol{W}) - \mathscr{R}(\boldsymbol{a},\boldsymbol{W})\big|$, Borell inequality yields

$$\mathbb{P}\Big\{\big|Z_{n,d,m}(\overline{a}) - \mathbb{E}Z_{n,d,m}(\overline{a})\big| \geq B\,t\Big\} \leq 2\,e^{-nt^2} + e^{-n}\,,$$
$$B := C''L\overline{a}(L\overline{a}+\tau)\,.$$

Together with Eq. (J.4), we thus obtain that the following holds with probability $1 - 2e^{-t} - e^{-n}$

$$\mathbb{E}\sup_{(\boldsymbol{a},\boldsymbol{W})\in\mathcal{W}_{m,d}(\overline{a})}\big|\widehat{\mathscr{R}}_n(\boldsymbol{a},\boldsymbol{W}) - \mathscr{R}(\boldsymbol{a},\boldsymbol{W})\big| \leq C(L^2\overline{a}^2+\tau^2)\sqrt{\frac{d}{n}} + C(L^2\overline{a}^2+\tau^2)\sqrt{\frac{t}{n}}\,.$$

This yields the desired claim.

## J.2  Proof of Theorem 3.2

We introduce the notations:

$$\boldsymbol{g}_{n,\ell}^{\boldsymbol{w}}(\boldsymbol{a},\boldsymbol{W}) := \frac{m}{|a_\ell|}\big[\nabla_{\boldsymbol{w}_\ell}\widehat{\mathscr{R}}_n(\boldsymbol{a},\boldsymbol{W}) - \nabla_{\boldsymbol{w}_\ell}\mathscr{R}(\boldsymbol{a},\boldsymbol{W})\big]\,, \tag{J.5}$$

$$g_{n,\ell}^{a}(\boldsymbol{a},\boldsymbol{W}) := m\big[\nabla_{a_\ell}\widehat{\mathscr{R}}_n(\boldsymbol{a},\boldsymbol{W}) - \nabla_{a_\ell}\mathscr{R}(\boldsymbol{a},\boldsymbol{W})\big]\,. \tag{J.6}$$

We begin by establishing a uniform convergence lemma.

**Lemma J.1.** *Under the data distribution of Section A, assume $\|\varphi\|_\infty \leq L$ and the activation function to be bounded differentiable with Lipschitz continuous first derivative $\|\sigma\|_\infty, \|\sigma'\|_\infty, \|\sigma'\|_{\mathrm{Lip}} \leq L$. Then there exists a universal constant $C_1$, and a constant $c_0 > 0$ dependent on $L, \tau, \alpha$ such that, with probability at least $1 - 2\exp(-nc_0)$,*

$$\sup_{(\boldsymbol{a},\boldsymbol{W})\in\mathcal{W}_{m,d}(\overline{a})}\max_{\ell\leq m}\big\|\boldsymbol{g}_{n,\ell}^{\boldsymbol{w}}(\boldsymbol{a},\boldsymbol{W})\big\| \leq C(L^2\overline{a}+\tau^2)\sqrt{\frac{d}{n}\log(ne/d)}\,, \tag{J.7}$$

$$\sup_{(\boldsymbol{a},\boldsymbol{W})\in\mathcal{W}_{m,d}(\overline{a})}\max_{\ell\leq m}\big|g_{n,\ell}^{a}(\boldsymbol{a},\boldsymbol{W})\big| \leq C(L^2\overline{a}+\tau^2)\sqrt{\frac{d}{n}}\,. \tag{J.8}$$

*Proof.* **Gradient with respect to $\boldsymbol{w}_\ell$.** By a concentration argument, it is sufficient to consider the expected supremum. Writing the formula for $\nabla_{\boldsymbol{w}_\ell}\widehat{\mathscr{R}}_n$ and using a standard symmetrization argument, we get

$$\mathbb{E}\sup_{(\boldsymbol{a},\boldsymbol{W})\in\mathcal{W}_{m,d}(\overline{a})}\big\|\boldsymbol{g}_{n,\ell}^{\boldsymbol{w}}(\boldsymbol{a},\boldsymbol{W})\big\| = \mathbb{E}\sup_{(\boldsymbol{a},\boldsymbol{W})\in\mathcal{W}_{m,d}(\overline{a}),\|\boldsymbol{u}\|\leq 1}\langle\boldsymbol{u}, \boldsymbol{g}_{n,\ell}^{\boldsymbol{w}}(\boldsymbol{a},\boldsymbol{W})\rangle$$

$$\leq 2\,\mathbb{E}\sup_{\boldsymbol{w},\boldsymbol{u}}\frac{1}{n}\sum_{i=1}^{n}\xi_i y_i\sigma'(\boldsymbol{w}^\mathsf{T}\boldsymbol{x}_i)\boldsymbol{u}^\mathsf{T}\boldsymbol{x}_i + 2\,\overline{a}\mathbb{E}\sup_{\boldsymbol{w},\overline{\boldsymbol{w}},\boldsymbol{u}}\frac{1}{n}\sum_{i=1}^{n}\xi_i\sigma'(\boldsymbol{w}^\mathsf{T}\boldsymbol{x}_i)\sigma(\overline{\boldsymbol{w}}^\mathsf{T}\boldsymbol{x}_i)\boldsymbol{u}^\mathsf{T}\boldsymbol{x}_i$$

$$=: B_1 + B_2\,,$$

where the $\xi_i$ are i.i.d. Radamacher random variables and in the last two lines it is understood that the supremum is over $\|\boldsymbol{w}\|, \|\overline{\boldsymbol{w}}\|, \|\boldsymbol{u}\| \leq 1$. Consider the second term in the last expression. Defining $\eta(x) = x\mathbf{1}_{|x|\leq M} + M(\mathbf{1}_{x>M} - \mathbf{1}_{x<-M})$, and $\overline{\eta}(x) = x - \eta(x)$, we have

$$B_2 = 2\overline{a}\,\mathbb{E}\sup_{\boldsymbol{w},\overline{\boldsymbol{w}},\boldsymbol{u}\in\mathsf{B}^d(1)}\frac{1}{n}\sum_{i=1}^{n}\xi_i\sigma'(\boldsymbol{w}^\mathsf{T}\boldsymbol{x}_i)\sigma(\overline{\boldsymbol{w}}^\mathsf{T}\boldsymbol{x}_i)\boldsymbol{u}^\mathsf{T}\boldsymbol{x}_i$$

$$\leq 2\overline{a}\,\mathbb{E}\sup_{\boldsymbol{w},\overline{\boldsymbol{w}},\boldsymbol{u}\in\mathsf{B}^d(1)}\frac{1}{n}\sum_{i=1}^{n}\xi_i\sigma'(\boldsymbol{w}^\mathsf{T}\boldsymbol{x}_i)\sigma(\overline{\boldsymbol{w}}^\mathsf{T}\boldsymbol{x}_i)\eta(\boldsymbol{u}^\mathsf{T}\boldsymbol{x}_i)$$

$$\qquad + 2\overline{a}\,\mathbb{E}\sup_{\boldsymbol{w},\overline{\boldsymbol{w}},\boldsymbol{u}\in\mathsf{B}^d(1)}\frac{1}{n}\sum_{i=1}^{n}\xi_i\sigma'(\boldsymbol{w}^\mathsf{T}\boldsymbol{x}_i)\sigma(\overline{\boldsymbol{w}}^\mathsf{T}\boldsymbol{x}_i)\overline{\eta}(\boldsymbol{u}^\mathsf{T}\boldsymbol{x}_i)$$

$$=: B_{2,1} + B_{2,2}\,.$$

Further defining $\phi(t_1, t_2, t_3) := \sigma(t_1)\sigma'(t_2)\eta(t_3)$ (which is $CL^2M$-Lipschitz for $M \geq 1$), we have

$$B_{2,1} = \overline{a}\,\mathbb{E}\sup_{\boldsymbol{w},\overline{\boldsymbol{w}},\boldsymbol{u}\in\mathsf{B}^d(1)}\frac{1}{n}\sum_{i=1}^{n}\xi_i\phi(\boldsymbol{w}^\mathsf{T}\boldsymbol{x}_i, \overline{\boldsymbol{w}}^\mathsf{T}\boldsymbol{x}_i, \boldsymbol{u}^\mathsf{T}\boldsymbol{x}_i). \tag{J.9}$$

Using the contraction inequality of [36], we get

$$B_{2,1} \leq CL^2M\overline{a}\left\{\mathbb{E}\sup_{\boldsymbol{w}\in\mathsf{B}^d(1)}\frac{1}{n}\sum_{i=1}^{n}\xi_i\boldsymbol{w}^\mathsf{T}\boldsymbol{x}_i + \mathbb{E}\sup_{\overline{\boldsymbol{w}}\in\mathsf{B}^d(1)}\frac{1}{n}\sum_{i=1}^{n}\xi_i\overline{\boldsymbol{w}}^\mathsf{T}\boldsymbol{x}_i + \mathbb{E}\sup_{\boldsymbol{u}\in\mathsf{B}^d(1)}\frac{1}{n}\sum_{i=1}^{n}\xi_i\boldsymbol{u}^\mathsf{T}\boldsymbol{x}_i\right\}$$

$$\leq CL^2M\overline{a}\sqrt{\frac{d}{n}}.$$

Next consider $B_{2,2}$:

$$B_{2,2} \leq 2\overline{a}L^2\mathbb{E}\sup_{\boldsymbol{u}\in\mathsf{B}^d(1)}\frac{1}{n}\sum_{i=1}^{n}|\overline{\eta}(\boldsymbol{u}^\mathsf{T}\boldsymbol{x}_i)|$$

$$\leq 2\overline{a}L^2\sup_{\boldsymbol{u}\in\mathsf{B}^d(1)}\mathbb{E}|\overline{\eta}(\boldsymbol{u}^\mathsf{T}\boldsymbol{x}_i)| + 2\overline{a}L^2\mathbb{E}\sup_{\boldsymbol{u}\in\mathsf{B}^d(1)}\frac{1}{n}\sum_{i=1}^{n}\xi_i|\overline{\eta}(\boldsymbol{u}^\mathsf{T}\boldsymbol{x}_i)|$$

$$\leq CL^2\overline{a}e^{-M^2/4} + CL^2\overline{a}\sqrt{\frac{d}{n}},$$

where the last inequality holds because $\boldsymbol{u}^\mathsf{T}\boldsymbol{x}_i$ is Gaussian with variance $\|\boldsymbol{u}\|^2$, and using again the contraction inequality. Collecting various terms and optimizing over $M \geq 1$, we obtain

$$B_2 \leq CL^2\overline{a}\left\{M\sqrt{\frac{d}{n}} + e^{-M^2/4}\right\}$$

$$\leq CL^2\overline{a}\sqrt{\frac{d}{n}\log(n/d)}.$$

The proof of Eq. (J.7) is completed by bounding $B_1$ along the same lines.

**Gradient with respect to $a_\ell$.** Writing $\nabla_{a_\ell}\widehat{\mathscr{R}}_n$ and using symmetrization, we get

$$\mathbb{E}\sup_{(\boldsymbol{a},\boldsymbol{W})\in\mathcal{W}_{m,d}(\overline{a})}\left|g_{n,\ell}^a(\boldsymbol{a},\boldsymbol{W})\right| = \mathbb{E}\sup_{(\boldsymbol{a},\boldsymbol{W})\in\mathcal{W}_{m,d}(\overline{a})}g_{n,\ell}^a(\boldsymbol{a},\boldsymbol{W})$$

$$\leq 2\,\mathbb{E}\sup_{\boldsymbol{W},\boldsymbol{u}}\frac{1}{n}\sum_{i=1}^{n}\xi_iy_i\sigma(\boldsymbol{w}_\ell^\mathsf{T}\boldsymbol{x}_i) + 2\,\mathbb{E}\sup_{\boldsymbol{a},\boldsymbol{W},\boldsymbol{u}}\frac{1}{n}\sum_{i=1}^{n}\xi_i\sum_{j=1}^{m}\frac{a_j}{m}\sigma(\boldsymbol{w}_j^\mathsf{T}\boldsymbol{x}_i)\sigma(\boldsymbol{w}_\ell^\mathsf{T}\boldsymbol{x}_i)$$

$$=: D_1 + D_2.$$

Consider term $D_2$, and define the $L^2$-Lipschitz function $\psi(t_1, t_2) := \sigma(t_1)\sigma(t_2)$,

$$D_2 \leq 2\overline{a}\,\mathbb{E}\sup_{\boldsymbol{W},\boldsymbol{u}}\max_{j\leq m}\frac{1}{n}\sum_{i=1}^{n}\xi_i\sigma(\boldsymbol{w}_j^\mathsf{T}\boldsymbol{x}_i)\sigma(\boldsymbol{w}_\ell^\mathsf{T}\boldsymbol{x}_i)$$

$$\leq 2\overline{a}\,\mathbb{E}\sup_{\boldsymbol{w},\overline{\boldsymbol{w}}\in\mathsf{B}^d(1)}\frac{1}{n}\sum_{i=1}^{n}\xi_i\psi(\boldsymbol{w}^\mathsf{T}\boldsymbol{x}_i, \overline{\boldsymbol{w}}^\mathsf{T}\boldsymbol{x}_i)$$

$$\leq CL^2\overline{a}\sqrt{\frac{d}{n}}.$$

Term $D_1$ is controlled analogously, yielding the proof of Eq. (J.8). $\qquad\square$

We next prove some continuity properties of the population risk $\mathscr{R}$. It is useful to recall the form:

$$\mathscr{R}(\boldsymbol{a},\boldsymbol{W}) = \frac{1}{2}(\tau^2 + \|\varphi\|^2) - \frac{1}{m}\sum_{i=1}^{m}a_i\widehat{\varphi}(\boldsymbol{U}^\mathsf{T}\boldsymbol{w}_i) + \frac{1}{2m^2}\sum_{i,j=1}^{m}a_ia_jh(\boldsymbol{w}_i^\mathsf{T}\boldsymbol{w}_j). \tag{J.10}$$

**Lemma J.2.** *Under the data distribution of Section A, assume $\|\varphi\|_\infty \le L$ that $\varphi$ and $\sigma$ are bounded differentiable with Lipschitz continuous first derivative, $\|\sigma\|_\infty, \|\sigma'\|_\infty, \|\sigma'\|_\infty \le L$, $\|\varphi\|_\infty, \|\nabla\varphi\|_\infty, \|\nabla\varphi\|_{\text{Lip}} \le L$, $L \ge 1$. Then, there exists an absolute constant $C$ such that for any $(\boldsymbol{a}, \boldsymbol{W}), (\boldsymbol{a}, \tilde{\boldsymbol{W}}) \in \mathcal{W}_{m,d}(\bar{a})$:*

$$\left\|\nabla_{\boldsymbol{w}_\ell}\mathscr{R}(\boldsymbol{a}, \tilde{\boldsymbol{W}}) - \nabla_{\boldsymbol{w}_\ell}\mathscr{R}(\boldsymbol{a}, \boldsymbol{W})\right\| \le CL^2 \frac{|a_\ell|}{m}(1+\bar{a})\max_{j\le m}\left\|\tilde{\boldsymbol{w}}_j - \boldsymbol{w}_j\right\|, \qquad \text{(J.11)}$$

$$\left|\partial_{a_\ell}\mathscr{R}(\boldsymbol{a}, \tilde{\boldsymbol{W}}) - \partial_{a_\ell}\mathscr{R}(\boldsymbol{a}, \boldsymbol{W})\right| \le \frac{CL^2}{m}(1+\bar{a})\max_{j\le m}\left\|\tilde{\boldsymbol{w}}_j - \boldsymbol{w}_j\right\|, \qquad \text{(J.12)}$$

*and*

$$\left\|\nabla_{\boldsymbol{w}_\ell}\mathscr{R}(\tilde{\boldsymbol{a}}, \boldsymbol{W}) - \nabla_{\boldsymbol{w}_\ell}\mathscr{R}(\boldsymbol{a}, \boldsymbol{W})\right\| \le \frac{CL^2}{m}(1+\bar{a})|\tilde{a}_\ell - a_\ell| + CL^2\frac{|a_\ell|}{m^2}\|\tilde{\boldsymbol{a}} - \boldsymbol{a}\|_1, \qquad \text{(J.13)}$$

$$\left|\partial_{a_\ell}\mathscr{R}(\tilde{\boldsymbol{a}}, \boldsymbol{W}) - \partial_{a_\ell}\mathscr{R}(\boldsymbol{a}, \boldsymbol{W})\right| \le \frac{CL^2}{m^2}\|\tilde{\boldsymbol{a}} - \boldsymbol{a}\|_1. \qquad \text{(J.14)}$$

*Proof.* As a preliminary remark, the assumptions on $\varphi$, $\sigma$ imply similar smoothness properties of $\widehat{\varphi}$, $h$. In particular, recall that $h(q) = \mathbb{E}[\sigma(G_1)\sigma(G_q)]$ for $(G_1, G_q)$ jointly Gaussian, centered with unit variance and covariance $\mathbb{E}[G_1, G_q] = 1$, whence its $k$-th derivative is $h^{(k)}(q) = \mathbb{E}[\sigma^{(k)}(G_1)\sigma^{(k)}(G_q)]$ (whenever $\sigma \in C^{(k)}(\mathbb{R})$). Therefore, the assumptions on $\sigma$ imply that $\|h'\|_\infty$, $\|h'\|_{\text{Lip}} \le L^2$. Similarly, $\|\nabla\widehat{\varphi}\|_\infty, \|\nabla\widehat{\varphi}\|_{\text{Lip}} \le CL^2$.

**Proof of Eq. (J.11).** Differentiating Eq. (J.10)

$$\frac{m}{a_\ell}\nabla_{\boldsymbol{w}_\ell}\mathscr{R}(\boldsymbol{a}, \boldsymbol{W}) = -\boldsymbol{U}\nabla\widehat{\varphi}(\boldsymbol{U}^\mathsf{T}\boldsymbol{w}_\ell) + \sum_{j=1}^m \frac{a_j}{m}h'(\boldsymbol{w}_\ell^\mathsf{T}\boldsymbol{w}_j)\boldsymbol{w}_j. \qquad \text{(J.15)}$$

Therefore

$$\begin{aligned}
\frac{m}{|a_\ell|}\left\|\nabla_{\boldsymbol{w}_\ell}\mathscr{R}(\boldsymbol{a}, \tilde{\boldsymbol{W}}) - \nabla_{\boldsymbol{w}_\ell}\mathscr{R}(\boldsymbol{a}, \boldsymbol{W})\right\| \le & \left\|\nabla\widehat{\varphi}(\boldsymbol{U}^\mathsf{T}\tilde{\boldsymbol{w}}_\ell) - \nabla\widehat{\varphi}(\boldsymbol{U}^\mathsf{T}\boldsymbol{w}_\ell)\right\| \\
& + \sum_{j=1}^m \frac{|a_j|}{m}\left\|h'(\tilde{\boldsymbol{w}}_\ell^\mathsf{T}\tilde{\boldsymbol{w}}_j)\tilde{\boldsymbol{w}}_j - h'(\boldsymbol{w}_\ell^\mathsf{T}\boldsymbol{w}_j)\boldsymbol{w}_j\right\| \\
\le & CL^2\|\tilde{\boldsymbol{w}}_\ell - \boldsymbol{w}_\ell\| + \bar{a}\max_{j\le m}\left\|h'(\tilde{\boldsymbol{w}}_\ell^\mathsf{T}\tilde{\boldsymbol{w}}_j)\tilde{\boldsymbol{w}}_j - h'(\boldsymbol{w}_\ell^\mathsf{T}\boldsymbol{w}_j)\boldsymbol{w}_j\right\|.
\end{aligned}$$

Further, by the above smoothness properties of $h$,

$$\left\|h'(\tilde{\boldsymbol{w}}_\ell^\mathsf{T}\tilde{\boldsymbol{w}}_j)\tilde{\boldsymbol{w}}_j - h'(\boldsymbol{w}_\ell^\mathsf{T}\boldsymbol{w}_j)\boldsymbol{w}_j\right\| \le CL^2\|\tilde{\boldsymbol{w}}_j - \boldsymbol{w}_j\| + CL^2\|\tilde{\boldsymbol{w}}_\ell - \boldsymbol{w}_\ell\|.$$

Substituting above, this yields the claim (J.11).

**Proof of Eq. (J.12).** We proceed analogously to the previous point. Namely

$$m\partial_{a_\ell}\mathscr{R}(\boldsymbol{a}, \boldsymbol{W}) = -\widehat{\varphi}(\boldsymbol{U}^\mathsf{T}\boldsymbol{w}_\ell) + \sum_{j=1}^m \frac{a_j}{m}h(\boldsymbol{w}_\ell^\mathsf{T}\boldsymbol{w}_j), \qquad \text{(J.16)}$$

whence

$$\begin{aligned}
m\left|\partial_{a_\ell}\mathscr{R}(\boldsymbol{a}, \tilde{\boldsymbol{W}}) - \partial_{a_\ell}\mathscr{R}(\boldsymbol{a}, \boldsymbol{W})\right| \le & \left|\widehat{\varphi}(\boldsymbol{U}^\mathsf{T}\tilde{\boldsymbol{w}}_\ell) - \widehat{\varphi}(\boldsymbol{U}^\mathsf{T}\boldsymbol{w}_\ell)\right| + \sum_{j=1}^m \frac{|a_j|}{m}\left|h(\tilde{\boldsymbol{w}}_\ell^\mathsf{T}\tilde{\boldsymbol{w}}_j) - h(\boldsymbol{w}_\ell^\mathsf{T}\boldsymbol{w}_j)\right| \\
\le & CL^2\|\tilde{\boldsymbol{w}}_\ell - \boldsymbol{w}_\ell\| + C\bar{a}L^2\left(\|\tilde{\boldsymbol{w}}_\ell - \boldsymbol{w}_\ell\| + \|\tilde{\boldsymbol{w}}_j - \boldsymbol{w}_j\|\right),
\end{aligned}$$

which implies immediately Eq. (J.12)

**Proof of Eq. (J.13).** Recalling Eq. (J.15), we have

$$m\left\|\nabla_{\boldsymbol{w}_\ell}\mathscr{R}(\tilde{\boldsymbol{a}}, \boldsymbol{W}) - \nabla_{\boldsymbol{w}_\ell}\mathscr{R}(\boldsymbol{a}, \boldsymbol{W})\right\| \le \left\|\nabla\widehat{\varphi}(\boldsymbol{U}^\mathsf{T}\boldsymbol{w}_\ell)\right\||\tilde{a}_\ell - a_\ell| + \sum_{j=1}^m \frac{1}{m}\left\|h'_s(\boldsymbol{w}_\ell^\mathsf{T}\boldsymbol{w}_j)\boldsymbol{w}_j\right\||\tilde{a}_\ell\tilde{a}_j - a_\ell a_j|$$

$$\leq CL|\tilde{a}_\ell - a_\ell| + CL^2 \frac{|a_\ell|}{m} \|\tilde{\boldsymbol{a}} - \boldsymbol{a}\|_1 + CL^2 \overline{a} \, |\tilde{a}_\ell - a_\ell| \,,$$

which proves the desired claim.

**Proof of Eq.** (J.13). Recalling Eq. (J.16), we have

$$m \big| \partial_{a_\ell} \mathscr{R}(\tilde{\boldsymbol{a}}, \boldsymbol{W}) - \partial_{a_\ell} \mathscr{R}(\boldsymbol{a}, \boldsymbol{W}) \big| \leq \sum_{j=1}^m \frac{1}{m} |h(\boldsymbol{w}_\ell^\mathsf{T} \boldsymbol{w}_j)| \cdot |\tilde{a}_j - a_j|$$

$$\leq \frac{CL^2}{m} \|\tilde{\boldsymbol{a}} - \boldsymbol{a}\|_1 \,.$$

$\square$

Using the last lemma and triangle inequality we get the following.

**Corollary J.3.** *Under the assumptions of Lemma J.2, there exists an absolute constant $C$ such that, for all $(\boldsymbol{a}, \boldsymbol{W}), (\boldsymbol{a}, \tilde{\boldsymbol{W}}) \in \mathcal{W}_{m,d}^\infty(\overline{a})$:*

$$\max_{\ell \leq m} \big\| \nabla_{\boldsymbol{w}_\ell} \mathscr{R}(\tilde{\boldsymbol{a}}, \tilde{\boldsymbol{W}}) - \nabla_{\boldsymbol{w}_\ell} \mathscr{R}(\boldsymbol{a}, \boldsymbol{W}) \big\| \leq \frac{CL^2 \overline{a}}{m}(1 + \overline{a}) \max_{j \leq m} \|\tilde{\boldsymbol{w}}_j - \boldsymbol{w}_j\| + \frac{CL^2}{m}(1 + \overline{a})\|\tilde{\boldsymbol{a}} - \boldsymbol{a}\|_\infty \,,$$

$$\max_{\ell \leq m} \big| \partial_{a_\ell} \mathscr{R}(\tilde{\boldsymbol{a}}, \tilde{\boldsymbol{W}}) - \partial_{a_\ell} \mathscr{R}(\boldsymbol{a}, \boldsymbol{W}) \big| \leq \frac{CL^2}{m}(1 + \overline{a}) \max_{j \leq m} \|\tilde{\boldsymbol{w}}_j - \boldsymbol{w}_j\| + \frac{CL^2}{m}\|\tilde{\boldsymbol{a}} - \boldsymbol{a}\|_\infty \,.$$

We next consider $\boldsymbol{a}(t), \boldsymbol{W}(t)$ that follows GF with respect to the empirical risk, as per Eq. (A.10), which we rewrite as

$$\dot{\boldsymbol{a}}(t) = -m \nabla_{\boldsymbol{a}} \widehat{\mathscr{R}}_n(\boldsymbol{a}(t), \boldsymbol{W}(t)) \,,$$
$$\dot{\boldsymbol{w}}_i(t) = -m \boldsymbol{P}_{\boldsymbol{w}_i}^\perp \nabla_{\boldsymbol{w}_i} \widehat{\mathscr{R}}_n(\boldsymbol{a}(t), \boldsymbol{W}(t)) \quad \forall i = 1, \ldots, m \,, \tag{J.17}$$

and denote by $\boldsymbol{a}_0(t), \boldsymbol{W}_0(t)$ the GF with respect to population risk:

$$\dot{\boldsymbol{a}}_0(t) = -m \nabla_{\boldsymbol{a}} \mathscr{R}(\boldsymbol{a}_0(t), \boldsymbol{W}_0(t)) \,,$$
$$\dot{\boldsymbol{w}}_{0,i}(t) = -m \boldsymbol{P}_{\boldsymbol{w}_i}^\perp \nabla_{\boldsymbol{w}_i} \mathscr{R}(\boldsymbol{a}_0(t), \boldsymbol{W}_0(t)) \quad \forall i = 1, \ldots, m \,. \tag{J.18}$$

**Lemma J.4.** *Under the data distribution of Section A, there exists constant $c_* = c_*(\delta)$, $c_0 = c_0(\delta)$ depending uniquely on $\delta > 0$, and an absolute constant $C$ such that the following holds. Assume $\varphi, \sigma$ to be bounded, differentiable with Lipschitz continuous first derivative $\|\varphi\|_\infty, \|\varphi'\|_\infty, \|\varphi'\|_{\mathrm{Lip}} \leq L$. $\|\sigma\|_\infty, \|\sigma'\|_\infty, \|\sigma'\|_{\mathrm{Lip}} \leq L$, Further assume $n/d \geq \exp(c_0 L^2)$, $L \geq 1$. Let $(\boldsymbol{a}(t), \boldsymbol{W}(t))$, $(\boldsymbol{a}_0(t), \boldsymbol{W}_0(t))$, be defined as above, with $\boldsymbol{W}(0) = \boldsymbol{W}_0(0)$ and $\boldsymbol{a}(0) = \boldsymbol{a}_0(0)$ such that $\|\boldsymbol{a}(0)\|_\infty = \|\boldsymbol{a}_0(0)\|_\infty \leq a_0$. Define*

$$T_*(m; c) := \inf \left\{ t : (\|\boldsymbol{a}(t)\|_\infty \vee \|\boldsymbol{a}_0(t)\|_\infty) \geq \left( c_* L^{-2} \log \frac{ne}{d} \right)^{1/3} \right\} \wedge \left( c_* L^{-2} \log \frac{ne}{d} \right)^{1/3} \,. \tag{J.19}$$

*Then*

$$\sup_{t \leq T_*(m; c)} \Delta(t) \leq C(L^2 + \tau^2) \left( \frac{d}{n} \right)^{1/2 - \delta} \,, \quad \Delta(t) := \max_{\ell \leq m} \|\tilde{\boldsymbol{w}}_\ell(t) - \boldsymbol{w}_\ell(t)\| + \|\tilde{\boldsymbol{a}}(t) - \boldsymbol{a}(t)\|_\infty \,. \tag{J.20}$$

*Proof.* We will prove that the desired bound holds on the high-probability event of Lemma J.1, where by we set $\overline{a} = (c_1 L^{-2} \log ne/d)^{1/3}$. Throughout the proof, we use $c_0, c_1, C$ to denote constants that might change from line to line, with dependence on the parameters of the problem as per the statement of the lemma.

We start by noting that, letting $\boldsymbol{v}_i = -m \nabla_{\boldsymbol{w}_i} \widehat{\mathscr{R}}_n(\boldsymbol{a}, \boldsymbol{W})$ and $\boldsymbol{v}_{0,i} = -(n/d) \nabla_{\boldsymbol{w}_{0,i}} \mathscr{R}(\boldsymbol{a}_0, \boldsymbol{W}_0)$, and $\boldsymbol{P}_{\boldsymbol{w}}^\perp := \boldsymbol{I} - \boldsymbol{w}\boldsymbol{w}^\mathsf{T}$ the projector orthogonal to $\boldsymbol{w}$.

$$\big\| \boldsymbol{P}_{\boldsymbol{w}_i}^\perp \boldsymbol{v}_i - \boldsymbol{P}_{\boldsymbol{w}_{0,i}}^\perp \boldsymbol{v}_{0,i} \big\| \leq \big\| \boldsymbol{P}_{\boldsymbol{w}_i}^\perp (\boldsymbol{v}_i - \boldsymbol{v}_{0,i}) \big\| + \big\| (\boldsymbol{P}_{\boldsymbol{w}_i}^\perp - \boldsymbol{P}_{\boldsymbol{w}_{0,i}}^\perp) \boldsymbol{v}_{0,i} \big\|$$

$$\leq \|\boldsymbol{v}_i - \boldsymbol{v}_{0,i}\| + \|\boldsymbol{w}_i \boldsymbol{w}_i^\mathsf{T} - \boldsymbol{w}_{0,i} \boldsymbol{w}_{0,i}^\mathsf{T}\|_{\mathrm{op}} \|\boldsymbol{v}_{0,i}\|$$
$$\leq \|\boldsymbol{v}_i - \boldsymbol{v}_{0,i}\| + 2\|\boldsymbol{w}_i - \boldsymbol{w}_{0,i}\|_{\mathrm{op}} \|\boldsymbol{v}_{0,i}\|.$$

Hence, comparing the evolution of $\boldsymbol{w}_i(t)$ and $\boldsymbol{w}_{0,i}(t)$, we get

$$\frac{\mathrm{d}}{\mathrm{d}t} \|\boldsymbol{w}_i(t) - \boldsymbol{w}_{0,i}(t)\| \leq m \big\| \nabla_{\boldsymbol{w}_i} \widehat{\mathscr{R}}_n(\boldsymbol{a}(t), \boldsymbol{W}(t)) - \nabla_{\boldsymbol{w}_i} \mathscr{R}(\boldsymbol{a}_0(t), \boldsymbol{W}_0(t)) \big\|$$
$$+ m \|\nabla_{\boldsymbol{w}_i} \mathscr{R}(\boldsymbol{a}_0(t), \boldsymbol{W}_0(t)))\| \cdot \|\boldsymbol{w}_i(t) - \boldsymbol{w}_{i,0}(t)\|$$
$$=: D_1 + D_2 \cdot \|\boldsymbol{w}_i(t) - \boldsymbol{w}_{i,0}(t)\|.$$

Since we are working on the event of Lemma J.1, and using Corollary J.3, we get, for $t \leq T_*(m; c)$.

$$D_1 \leq m \big\| \nabla_{\boldsymbol{w}_i} \widehat{\mathscr{R}}_n(\boldsymbol{a}(t), \boldsymbol{W}(t)) - \nabla_{\boldsymbol{w}_i} \mathscr{R}(\boldsymbol{a}(t), \boldsymbol{W}(t)) \big\|$$
$$+ m \big\| \nabla_{\boldsymbol{w}_i} \mathscr{R}(\boldsymbol{a}(t), \boldsymbol{W}(t)) - \nabla_{\boldsymbol{w}_i} \mathscr{R}(\boldsymbol{a}_0(t), \boldsymbol{W}_0(t)) \big\|$$
$$\leq C(L^2 \bar{a} + \tau^2) \sqrt{\frac{d}{n} \log(ne/d)} + CL^2(1 + \bar{a}^2) \max_{j \leq m} \|\boldsymbol{w}_j(t) - \boldsymbol{w}_{0,j}(t)\|$$
$$+ CL^2(1 + \bar{a}) \|\boldsymbol{a}(t) - \boldsymbol{a}_0(t)\|_\infty.$$

Further

$$D_2 = |a_i| \Big\| \boldsymbol{U} \nabla \widehat{\varphi}(\boldsymbol{U}^\mathsf{T} \boldsymbol{w}_i) - \frac{1}{m} \sum_{j=1}^m a_j h'(\boldsymbol{w}_i^\mathsf{T} \boldsymbol{w}_j) \boldsymbol{w}_j \Big\|$$
$$\leq C\bar{a}\big(L^2 + \bar{a}L^2\big).$$

Collecting all the terms, and using $\bar{a} \geq 1$, we get

$$\frac{\mathrm{d}}{\mathrm{d}t} \|\boldsymbol{w}_i(t) - \boldsymbol{w}_{0,i}(t)\| \leq C\bar{a}(L^2 \bar{a} + \tau^2) \sqrt{\frac{d}{n} \log(ne/d)} + CL^2(1 + \bar{a}^2)\Delta(t). \qquad \text{(J.21)}$$

We next consider the evolution of second-layer weights:

$$\frac{\mathrm{d}}{\mathrm{d}t} |a_i(t) - a_{0,i}(t)\| \leq m \big\| \partial_{a_i} \widehat{\mathscr{R}}_n(\boldsymbol{a}(t), \boldsymbol{W}(t)) - \partial_{a_i} \mathscr{R}(\boldsymbol{a}_0(t), \boldsymbol{W}_0(t)) \big\|$$
$$\leq m \big\| \partial_{a_i} \widehat{\mathscr{R}}_n(\boldsymbol{a}(t), \boldsymbol{W}(t)) - \partial_{a_i} \mathscr{R}(\boldsymbol{a}(t), \boldsymbol{W}(t)) \big\|$$
$$+ m \big\| \partial_{a_i} \mathscr{R}_n(\boldsymbol{a}(t), \boldsymbol{W}(t)) - \partial_{a_i} \mathscr{R}(\boldsymbol{a}_0(t), \boldsymbol{W}_0(t)) \big\|$$
$$\leq C(L^2 \bar{a} + \tau^2) \sqrt{\frac{d}{n}} + CL^2(1 + \bar{a}) \max_{j \leq m} \|\boldsymbol{w}_j(t) - \boldsymbol{w}_{0,j}(t)\| + CL^2 \|\boldsymbol{a}(t) - \boldsymbol{a}_0(t)\|_\infty$$
$$\leq C(L^2 \bar{a} + \tau^2) \sqrt{\frac{d}{n}} + CL^2(1 + \bar{a})\Delta(t).$$

Using the last bound together with Eq. (J.21), we get

$$\frac{\mathrm{d}}{\mathrm{d}t} \Delta(t) \leq C\bar{a}(L^2 \bar{a} + \tau^2) \sqrt{\frac{d}{n} \log(ne/d)} + CL^2(1 + \bar{a}^2)\Delta(t)$$

whence the claim follows by Gromwall inequality for sufficiently small $c_1$. $\qquad \square$

We finally need a lemma from [10] approximating GF in the population risk by the mean field dynamics.

**Lemma J.5** (Corollary 1 and Proposition 3 [10]). *Let $\boldsymbol{a}_0(t)$, $\boldsymbol{W}_0(t)$ be GF with respect to the population risk (J.18) with initialization $|a_{0,i}(0)| \leq a_0$ and $(\boldsymbol{w}_{0,i}(0))_{i \leq m} \sim \mathsf{Unif}(S^{d-1})$. Recall that $a_i^{\mathrm{mfl}}(t)$, $\boldsymbol{v}_i^{\mathrm{mfl}}(t)$ is the solution of the ODEs (3.4) with initialization $a_i^{\mathrm{mfl}}(0) = a_{0,i}(0)$, $\boldsymbol{v}_i^{\mathrm{mfl}}(t) = 0$. Under the assumptions of Theorem 3.2, for any $\varepsilon > 0$ there exists constants $c_0, c_1$ depending uniquely on $L$, and an absolute constant $C$ such that letting $T_{\mathrm{lb}}(m) = ((c_0/\varepsilon)\log m)^{1/3}$, the following happens with probability at least $1 - 2\exp(-c_1 d)$,*

$$\sup_{t \leq T_{\mathrm{lb}}(m)} \frac{1}{m} \sum_{i=1}^m \Big( |a_i(t) - a_i^{\mathrm{mfl}}(t)| + \|\boldsymbol{v}_i(t) - \boldsymbol{v}_i^{\mathrm{mfl}}(t)\| \Big) \leq C m^\varepsilon \Big\{ \frac{1}{\sqrt{m}} + \frac{1}{\sqrt{d}} \Big\}, \qquad \text{(J.22)}$$

$$\sup_{t \leq T_{\mathrm{lb}}(m)} \Big( \mathscr{R}(\boldsymbol{a}(t), \boldsymbol{W}(t)) - e_{\mathrm{ts}}(t) \Big) \leq C m^\varepsilon \Big\{ \frac{1}{\sqrt{m}} + \frac{1}{\sqrt{d}} \Big\}. \qquad \text{(J.23)}$$

*Proof of Theorem 3.2.* Throughout the proof $L, \tau, \alpha$ are assumed to be fixed, and constants $C, c_0, \ldots$ depend on them and can change from line to line. We will further work on the high probability events of Theorem 3.1, Lemma J.4, and Lemma J.5. By Theorem 3.1, for all $t \leq T_{\mathrm{lb}}(m)$ we have $\|\boldsymbol{a}(t)\|_\infty \leq c_2(\log 2m)^{1/3}$ (where the constant $c_2$ can be made sufficiently small, by eventually reducing $c_1$). An analogous of of Theorem 3.1 for the population risk implies $\|\boldsymbol{a}_0(t)\|_\infty \leq c_2(\log 2m)^{1/3}$ as well for all $t \leq T_{\mathrm{lb}}(m)$. Hence we can apply Lemma J.4 and Lemma J.5, which yields the claim. $\qquad\square$

## K  Dynamical mean field theory for non-Gaussian model

The DMFT equations for GF in the original non-Gaussian model can be derived from the general theory of [13].

Given a (positive semi-definite) kernel $\boldsymbol{Q} : \mathbb{R}_{\geq 0} \times \mathbb{R}_{\geq 0} \to \mathbb{R}^{m \times m}$, $(t, z) \mapsto \boldsymbol{Q}(t, s)$, we write $\boldsymbol{z} \sim \mathsf{GP}(0, \boldsymbol{Q})$ if $\boldsymbol{z}$ is a centered Gaussian process with values in $\mathbb{R}^m$ and covariance $\mathbb{E}[\boldsymbol{z}(t)\boldsymbol{z}(s)^\mathsf{T}] = \boldsymbol{Q}(t, s)$.

The DMFT equations can be interpreted as a set of fixed point equations for the functions $C_{ij}, R_{ij}, a_i$.

We define the deterministic processes $\boldsymbol{a}(t)$, $\nu_i(t)$ and stochastic processes $\boldsymbol{w}^e(t) = (w_i^e(t) : i \leq m)$, $\boldsymbol{r}(t) = (r_i(t) : i \leq m)$, as the solution of

$$\frac{\mathrm{d}a_i(t)}{\mathrm{d}t} = \frac{\overline{\alpha}}{m} \mathbb{E}\{E(t)\,\sigma(r_i(t))\}, \tag{K.1}$$

$$\nu_i(t) = \frac{\overline{\alpha}}{m} a_i(t) \mathbb{E}\{E(t)\sigma'(r_i(t))r_i(t)\}, \tag{K.2}$$

$$\frac{\mathrm{d}w_i^e(t)}{\mathrm{d}t} = -\nu_i(t)w_i^e(t) - \frac{1}{m}\sum_{l=1}^m \int_0^t M_{i,l}(t,s)w_l^e(s)\,\mathrm{d}s \tag{K.3}$$

$$-\sum_{j=1}^k M_{i,j}(t,*)u_j + \eta_i(t), \qquad \boldsymbol{\eta} \sim \mathsf{GP}(0, \boldsymbol{C}^E),$$

$$r_i(t) = \frac{1}{m}\sum_{l=1}^m \int_0^t R_{il}(t,s)\,a_l(s)E(s)\,\sigma'(r_l(s))\,\mathrm{d}s + \xi_i(t), \qquad \boldsymbol{\xi} \sim \mathsf{GP}(0, \boldsymbol{C}), \tag{K.4}$$

$$E(t) := y - \frac{1}{m}\sum_{l=1}^m a_l(t)\sigma(r_l(t)). \tag{K.5}$$

Here, in the first equation, $(\boldsymbol{w}^e(0), \boldsymbol{u}) \sim \mathsf{N}(0, \boldsymbol{I}_m) \otimes \mathsf{N}(0, \boldsymbol{I}_k)$ are independent of $\boldsymbol{\eta}$. In the second equation, $y = \varphi(\boldsymbol{r}_0) + \varepsilon$ with $(\boldsymbol{r}_0, \varepsilon) \sim \mathsf{N}(0, \boldsymbol{I}_k) \otimes \mathsf{N}(0, \tau^2)$ independent of $\boldsymbol{\xi}$.

$$M_{ij}(t,s) = \overline{\alpha}\mathbb{E}\{S_{ij}(t)\}\,\delta(t-s) + \overline{\alpha}\sum_{l=1}^m \mathbb{E}\Big\{S_{il}(t)\frac{\partial r_l(t)}{\partial \xi_j(s)}\Big\}, \tag{K.6}$$

$$M_{ij}(t,*) = -\overline{\alpha}\frac{a_i(t)}{m}\mathbb{E}\{\sigma'(r_i(t))\nabla_j\varphi(\boldsymbol{r}_0)\} + \frac{\overline{\alpha}}{m}\sum_{l=1}^m \mathbb{E}\Big\{S_{il}(t)\frac{\partial r_l(t)}{\partial r_{0,j}}\Big\}, \tag{K.7}$$

$$C_{i,j}^E(t,s) = \overline{\alpha}\frac{a_i(t)a_j(s)}{m^2}\mathbb{E}\{E(t)E(s)\,\sigma'(r_i(t))\sigma'(r_j(s))\}, \tag{K.8}$$

$$S_{ij}(t) := -a_i(t)\,E(t)\sigma''(r_i(t))\delta_{ij} + \frac{a_i(t)a_j(t)}{m}\sigma'(r_i(t))\sigma'(r_j(t)), \tag{K.9}$$

and

$$C_{ij}(t,s) = \mathbb{E}\Big\{w_i^e(t)w_j^e(s)\Big\}, \tag{K.10}$$

$$R_{ij}(t,s) = \mathbb{E}\Big\{\frac{\partial w_i^e(t)}{\partial \eta_j(s)}\Big\}. \tag{K.11}$$

In solving the above, the random functions $\frac{\partial w_i^e(t)}{\partial \eta_j(s)}$ and $\frac{\partial r_i(t)}{\partial \xi_j(s)}$ (for $t > s$) are defined to be solutions of the following linear ODEs:

$$\frac{\mathrm{d}}{\mathrm{d}t}\frac{\partial w_i^e(t)}{\partial \eta_j(s)} = -\nu_i(t)\frac{\partial w_i^e(t)}{\partial \eta_j(s)} - \frac{1}{m}\sum_{l=1}^m \int_s^t M_{i,l}(t,t')\frac{\partial w_l^e(t')}{\partial \eta_j(s)}\,\mathrm{d}t', \tag{K.12}$$

$$\frac{\partial r_i(t)}{\partial \xi_j(s)} = -\frac{1}{m} \sum_{l,q=1}^{m} \int_s^t R_{il}(t,t') S_{lq}(t') \left[ \frac{\partial r_q(t')}{\partial \xi_j(s)} + \delta_{qj}\delta(t'-s) \right] \mathrm{d}t' \,, \tag{K.13}$$

$$\frac{\partial r_i(t)}{\partial r_{0,j}} = \frac{1}{m} \sum_{l=1}^{m} \int_0^t R_{il}(t,s) a_l(s) \sigma'(r_l(s)) \nabla_j \varphi(\boldsymbol{r}_0) \, \mathrm{d}s - \frac{1}{m} \sum_{l,q=1}^{m} \int_0^t R_{il}(t,s) S_{lq}(s) \frac{\partial r_q(s)}{\partial r_{0,j}} \mathrm{d}s \,, \tag{K.14}$$

with boundary condition $\frac{\partial w_i^e(t)}{\partial \eta_j(t)} = \delta_{ij}$ for the first equation.

## L  Derivation of the dynamical mean field theory equations

The study of the dynamics in such high-dimensional limit can be done via dynamical mean field theory (DMFT) [18]. The theoretical technology that we will employ is an evolution of the one first derived in [31, 32] to study gradient flow and stochastic gradient descent on models that are very much related to the Gaussian process we are discussing here [51, 42, 33]. We remark that the formalism considered here can be used to study both the single index model and the pure noise case. To obtain the pure noise model, one can set $h_t = \hat{\varphi} = 0$. Furthermore, the extension to multi-index models can be also done easily on the same lines.

The analysis of Eqs. (A.10) can be done by recasting them into a path integral representation. We follow the same procedure presented in [31]. Eqs.(A.10) can be packed into a dynamical partition function

$$1 = Z_{dyn} = \int D\boldsymbol{a} D\tilde{\boldsymbol{a}} \int D\boldsymbol{W} D\hat{\boldsymbol{W}} \exp\left[ A[\boldsymbol{a}, \tilde{\boldsymbol{a}}, \boldsymbol{W}, \hat{\boldsymbol{W}}] \right] \tag{L.1}$$

where the path measure $D\boldsymbol{a}(t)D\tilde{\boldsymbol{a}}(t)D\boldsymbol{W}D\hat{\boldsymbol{W}}$ is implicitly defined. The action $A$ reads

$$A = i \sum_{l=1}^{m} \int \tilde{a}_l(t) \left[ d\frac{\mathrm{d}a_l(t)}{\mathrm{d}t} + n\frac{\partial \widehat{\mathscr{R}}_n}{\partial a_l(t)} \right] \mathrm{d}t + i \sum_{l=1}^{m} \int \langle \hat{\boldsymbol{w}}_l(t), d\frac{\boldsymbol{w}_l(t)}{\mathrm{d}t} + d\nu_i(t)\boldsymbol{w}_i(t) + n\frac{\partial \widehat{\mathscr{R}}_n}{\partial \boldsymbol{w}_l(t)} \rangle \mathrm{d}t \,. \tag{L.2}$$

Eq. (L.2) can be rewritten by introducing Grassmann variables [62]. Call $\hat{a} = (t_a, \theta_a)$ a supertime coordinate, with $\theta_a$ a Grassmann variable. Define, with a slight abuse of notation

$$\begin{aligned} \boldsymbol{w}_l(\hat{a}) &= \boldsymbol{w}_l(t_a) + i\theta_a \hat{\boldsymbol{w}}_l \\ a_l(\hat{a}) &= a_l(t_a) + i\theta_a \tilde{a}_l(t_a) \quad l \leq m \,. \end{aligned} \tag{L.3}$$

Eq. (L.2) can be written as

$$A = \frac{d}{2} \sum_{i,j=1}^{m} \int_{\hat{a},\hat{b}} \mathcal{K}_{ij}(\hat{a},\hat{b}) \langle \boldsymbol{w}_i(\hat{a}), \boldsymbol{w}_j(\hat{b}) \rangle + \frac{d}{2} \sum_{i,j=1}^{m} \int_{\hat{a},\hat{b}} \tilde{\mathcal{K}}_{ij}(\hat{a},\hat{b}) a_i(\hat{a}) a_j(\hat{b}) - n \int_{\hat{a}} \widehat{\mathscr{R}}_n(\boldsymbol{\theta}(\hat{a})) \,. \tag{L.4}$$

The first two terms of the sum describe the kinetic terms of the dynamical equations of motion. The last term instead contains the interaction between the weights of the network. The empirical risk $\widehat{\mathscr{R}}_n$ depends on the training dataset. We are interested in understanding the behavior of the dynamics of gradient flow when we average over its realizations. Since the dynamical partition function is identically one we can average it directly over the dataset [2]. In this way we have

$$\begin{aligned} 1 = Z_{dyn} = \int D\boldsymbol{a}(\hat{a}) D\boldsymbol{W}(\hat{a}) \exp &\left[ \frac{d}{2} \sum_{i,j=1}^{m} \int_{\hat{a},\hat{b}} \mathcal{K}_{ij}(\hat{a},\hat{b}) \langle \boldsymbol{w}_i(\hat{a}), \boldsymbol{w}_j(\hat{b}) \rangle \right. \\ &\left. + \frac{d}{2} \sum_{l,l'=1}^{m} \int_{\hat{a},\hat{b}} \tilde{\mathcal{K}}_{ll'}(\hat{a},\hat{b}) a_l(\hat{a}) a_{l'}(\hat{b}) \right] \mathbb{E}\left[ \exp\left( -n \int_{\hat{a}} \widehat{\mathscr{R}}_n(\boldsymbol{\theta}(\hat{a})) \right) \right] \,. \end{aligned} \tag{L.5}$$

---

[2]We emphasize anyway that the average over the dataset is not mandatory: the resulting DMFT equations are self-averaging.

Performing standard manipulation, see [31], the dynamical partition function, for $d \to \infty$, can be written as

$$Z_{dyn} = \int D(\underline{a}, \tilde{Q}, R) \exp\left[S_{dyn}(\underline{a}, \tilde{Q}, R)\right] . \tag{L.6}$$

The dynamical action $S_{dyn}$ is given by

$$S_{dyn} = \frac{d}{2} \sum_{ll'=1}^{m} \int_{\hat{a}\hat{b}} \mathcal{K}_{ll'}(\hat{a}, \hat{b}) \left(\tilde{Q}_{ll'}(\hat{a}, \hat{b}) + r_l(\hat{a})r_{l'}(\hat{b})\right) + \frac{d}{2} \ln \det(\tilde{Q}) + \frac{\overline{\alpha}d}{2} \ln \det(\boldsymbol{I} + \Sigma_+)$$

$$+ \frac{d}{2} \int_{\hat{a}\hat{b}} \sum_{ll'} \tilde{\mathcal{K}}_{ll'}(\hat{a}, \hat{b}) a_l(\hat{a})a_{l'}(\hat{b}) \tag{L.7}$$

where $\overline{\alpha} = n/d$ and

$$\Sigma_+(\hat{a}, \hat{b}) = \tau^2 + h_t(1) + \frac{1}{m^2} \sum_{l,l'=1}^{m} a_l(\hat{a})a_{l'}(\hat{b})h\left(\tilde{Q}_{ll'}(\hat{a}, \hat{b}) + r_l(\hat{a})r_{l'}(\hat{b})\right)$$

$$- \frac{1}{m} \sum_{l=1}^{m} a_l(\hat{a})\hat{\varphi}(r_l(\hat{a})) - \frac{1}{m} \sum_{l=1}^{m} a_l(\hat{b})\hat{\varphi}(r_l(\hat{b})) . \tag{L.8}$$

The kinetic kernels $\mathcal{K}$ and $\tilde{\mathcal{K}}$ are implicitly defined in such a way that they reproduce the time derivative part of the dynamical equations (A.10).

In the large $d$ limit, fixing $m$ and $\overline{\alpha}$, the path integral in Eq. (L.5) concentrates on its saddle point. The corresponding equations are

$$0 = \sum_{\gamma=1}^{m} \int_{\hat{c}} \mathcal{K}_{l\gamma}(\hat{a}, \hat{c})Q_{\gamma l'}(\hat{c}, \hat{b}) + \frac{\overline{\alpha}}{m} a_l(\hat{a})\hat{\varphi}'(r_l(\hat{a}))r_{l'}(\hat{b}) \int_{\hat{d}} (\boldsymbol{I} + \Sigma)^{-1}(\hat{a}, \hat{d})$$

$$- \frac{\overline{\alpha}}{m^2} \sum_{\gamma=1}^{m} \int_{\hat{c}} (\boldsymbol{I} + \Sigma)^{-1}(\hat{a}, \hat{c})a_l(\hat{a})a_\gamma(\hat{c})h'(Q_{l\gamma}(\hat{a}, \hat{c}))Q_{\gamma l'}(\hat{c}, \hat{b}) + \delta_{ll'}(\hat{a}, \hat{b}) \tag{L.9}$$

and

$$0 = \sum_{\gamma=1}^{m} \int_{\hat{c}} \mathcal{K}_{l\gamma}(\hat{a}, \hat{c})r_\gamma(\hat{c}) + \frac{\overline{\alpha}}{m} a_l(\hat{a})\varphi'(r_l(\hat{a})) \int_{\hat{d}} (\boldsymbol{I} + \Sigma)^{-1}(\hat{a}, \hat{d})$$

$$- \frac{\overline{\alpha}}{m^2} \sum_{\gamma=1}^{m} \int_{\hat{c}} (\boldsymbol{I} + \Sigma)^{-1}(\hat{a}, \hat{c})a_l(\hat{a})a_\gamma(\hat{c})h'(Q_{l\gamma}(\hat{c}))r_\gamma(\hat{c}) \tag{L.10}$$

where

$$Q_{ll'}(\hat{a}, \hat{b}) = \tilde{Q}_{ll'}(\hat{a}, \hat{b}) + r_l(\hat{a})r_{l'}(\hat{b})$$

$$\Sigma(\hat{a}, \hat{b}) = \tau^2 + h_t(1) + \frac{1}{m^2} \sum_{ll'}^{m} a_l(\hat{a})a_{l'}(\hat{b})h\left(Q_{ll'}(\hat{a}, \hat{b})\right)$$

$$- \frac{1}{m} \sum_{l=1}^{m} a_l(\hat{a})\hat{\varphi}(r_l(\hat{a})) - \frac{1}{m} \sum_{l=1}^{m} a_l(\hat{b})\hat{\varphi}(r_l(\hat{b})) . \tag{L.11}$$

If Lagrange multipliers are added to constrain the norm of the the weights of the first layer, one should provide additional equations for them. Finally the equations for the dynamics of the second layer weights are given by

$$\sum_{\gamma=1}^{m} \int_{\hat{c}} \tilde{\mathcal{K}}_{l\gamma}(\hat{a}, \hat{c})a_\gamma(\hat{c}) = -\overline{\alpha} \int_{\hat{c}} (\boldsymbol{I} + \Sigma)^{-1}(\hat{c}, \hat{a}) \left[\frac{1}{m^2} \sum_{\gamma=1}^{m} a_\gamma(\hat{c})h\left[Q_{\gamma l}(\hat{c}, \hat{a})\right] - \frac{1}{m}\hat{\varphi}(r_l(\hat{a}))\right] \tag{L.12}$$

Eqs. (L.9)-(L.12) contain all the information about the dynamics. In order to fully specify the behavior of physical quantities such has the train and test error, it is useful to unfold the Grassmann structure of Eqs. (L.9)-(L.12).

## L.1 Unfolding the Grassmann structure

Causality of the dynamics implies that the following parametrization is the most general solution of the saddle point equations

$$
\begin{aligned}
r_\alpha(\hat{a}) &= r_\alpha(t_a) \\
a_\alpha(\hat{a}) &= a_\alpha(t_a) \\
Q_{\alpha\beta}(\hat{a}, \hat{b}) &= C_{\alpha\beta}(t_a, t_b) + \theta_a R_{\beta\alpha}(t_b, t_a) + \theta_b R_{\alpha\beta}(t_a, t_b) \\
(\boldsymbol{I} + \Sigma)^{-1}(\hat{a}, \hat{b}) &= C_A(t_a, t_b) + \theta_b R_A(t_a, t_b) + \theta_a R_A(t_b, t_a) .
\end{aligned}
\tag{L.13}
$$

Plugging this parametrization into the saddle point equations we get that the correlators in Eqs. (L.13) satisfy the following DMFT equations

$$
\begin{aligned}
\frac{\mathrm{d}a_\alpha(t)}{\mathrm{d}t} &= -\frac{\overline{\alpha}}{m} \int_0^t R_A(t, s) \left[ \frac{1}{m} \sum_{l=1}^m a_l(s) h\left[C_{l\alpha}(s, t)\right] - \hat{\varphi}(r_\alpha(t)) \right] \mathrm{d}s \\
&\quad - \frac{\overline{\alpha}}{m} \int_0^t C_A(t, s) \frac{1}{m} \sum_{l=1}^m a_l(s) h'[C_{l\alpha}(s, t)] R_{\alpha l}(t, s) \mathrm{d}s
\end{aligned}
\tag{L.14}
$$

$$
\begin{aligned}
\frac{\mathrm{d}r_\alpha(t)}{\mathrm{d}t} &= -\nu_\alpha(t) r_\alpha(t) + \frac{\overline{\alpha}}{m} a_\alpha(t) \hat{\varphi}'(r_\alpha(t)) \int_0^t R_A(t, s) \mathrm{d}s \\
&\quad - \frac{a_\alpha(t)}{m} \sum_{\gamma=1}^m \int_0^t M_{\alpha\gamma}^R(t, s) a_\gamma(s) r_\gamma(s) \mathrm{d}s
\end{aligned}
\tag{L.15}
$$

$$
\begin{aligned}
\frac{\partial C_{\alpha\beta}(t_a, t_b)}{\partial t_a} &= -\nu_\alpha(t_a) C_{\alpha\beta}(t_a, t_b) + \frac{\overline{\alpha}}{m} a_\alpha(t_a) \hat{\varphi}'(r_\alpha(t_a)) r_\beta(t_b) \int_0^{t_a} R_A(t_a, s) \mathrm{d}s \\
&\quad - \frac{a_\alpha(t_a)}{m} \sum_{\gamma=1}^m \int_0^{t_a} M_{\alpha\gamma}^R(t_a, s) a_\gamma(s) C_{\gamma\beta}(s, t_b) \mathrm{d}s \\
&\quad - \frac{a_\alpha(t_a)}{m} \sum_{\gamma=1}^m \int_0^{t_b} M_{\alpha\gamma}^C(t_a, s) a_\gamma(s) R_{\beta\gamma}(t_b, s) \mathrm{d}s
\end{aligned}
\tag{L.16}
$$

$$
\begin{aligned}
\frac{\partial R_{\alpha\beta}(t_a, t_b)}{\partial t_a} &= -\nu_\alpha(t_a) R_{\alpha\beta}(t_a, t_b) + \delta_{\alpha\beta}(t_a - t_b) \\
&\quad - \frac{a_\alpha(t_a)}{m} \sum_{\gamma=1}^m \int_{t_b}^{t_a} M_{\alpha\gamma}^R(t_a, s) a_\gamma(s) R_{\gamma\beta}(s, t_b) \mathrm{d}s .
\end{aligned}
\tag{L.17}
$$

Note that we used the notation according to which the prime sign denotes the derivatives of the functions with respect to their argument. The memory kernels $M^R$ and $M^C$ are defined by

$$
\begin{aligned}
M_{\alpha\gamma}^R(t, s) &= \frac{\overline{\alpha}}{m} \left[ R_A(t, s) h'(C_{\alpha\gamma}(t, s)) + C_A(t, s) h''(C_{\alpha\gamma}(t, s)) R_{\alpha\gamma}(t, s) \right] \\
M_{\alpha\gamma}^C(t, s) &= \frac{\overline{\alpha}}{m} C_A(t, s) h'(C_{\alpha\gamma}(t, s)) .
\end{aligned}
\tag{L.18}
$$

The kernels in Eq. (L.18) depend on $R_A$ and $C_A$ that are defined in Eqs. (L.13). The corresponding equations are

$$
\begin{aligned}
\int_{t'}^t \left[ \delta(t - s) + \Sigma_R(t, s) \right] R_A(s, t') \mathrm{d}s &= \delta(t - t') \\
\int_0^t \left[ \delta(t - s) + \Sigma_R(t, s) \right] C_A(s, t') \mathrm{d}s + \int_0^{t'} \Sigma_C(t, s) R_A(t', s) \mathrm{d}s &= 0
\end{aligned}
\tag{L.19}
$$

where

$$\Sigma_C(t,s) = \tau^2 + h_t(1) + \frac{1}{m^2} \sum_{ll'=1} a_l(t)a_{l'}(s)h[C_{ll'}(t,s)]$$

$$- \frac{1}{m} \sum_{l=1}^m a_l(t)\hat\varphi(r_l(t)) - \frac{1}{m} \sum_{l=1}^m a_l(s)\hat\varphi(r_l(s)) \tag{L.20}$$

$$\Sigma_R(t,s) = \frac{1}{m^2} \sum_{ll'=1}^m a_l(t)a_{l'}(s)h'[C_{ll'}(t,s)]R_{ll'}(t,s) .$$

The Lagrange multipliers $\nu_\alpha(t)$ have to be fixed self-consistently to enforce that $C_{\alpha,\alpha}(t,t) = 1$ given that $\boldsymbol{w}_\alpha \in \mathbb{S}^{d-1}$. The corresponding equations are

$$\nu_\alpha(t_a) = \frac{\overline\alpha}{km} \sum_{\tau=1}^k a_\alpha(t_a)\hat\varphi'(r_{\tau\alpha}(t_a))r_{\tau\alpha}(t_a) \int_0^{t_a} R_A(t_a,s)\mathrm{d}s$$

$$- \frac{a_\alpha(t_a)}{m} \sum_{\gamma=1}^m \int_0^{t_a} M_{\alpha\gamma}^R(t_a,s)a_\gamma(s)C_{\gamma\alpha}(s,t_a)\mathrm{d}s \tag{L.21}$$

$$- \frac{a_\alpha(t_a)}{m} \sum_{\gamma=1}^m \int_0^{t_a} M_{\alpha\gamma}^C(t_a,s)a_\gamma(s)R_{\gamma\alpha}(t_a,s)\mathrm{d}s$$

Finally we need to add a set of equation to propagate the diagonal elements of the correlation matrix:

$$\frac{\mathrm{d}C_{\alpha\beta}(t_a,t_a)}{\mathrm{d}t_a} = \lim_{t'\to t_a} \left[ \frac{\partial C_{\alpha\beta}(t_a,t')}{\partial t_a} + \frac{\partial C_{\beta\alpha}(t_a,t')}{\partial t_a} \right] . \tag{L.22}$$

These dynamical equations can be integrated from a set of initial conditions that fully specify the initial status of the neurons. We will consider a random initial condition for the weights of the first layer so that

$$\begin{aligned} r_\alpha(0) &= 0 & \forall \alpha &= 1,\dots,m \\ C_{\alpha\neq\beta}(0,0) &= 0 & \forall \alpha &\neq \beta = 1,\dots,m \\ C_{\alpha\alpha}(0,0) &= 1 & \forall \alpha &= 1,\dots,m \\ R_{\alpha\beta}(0,0) &= 0 & \forall \alpha,\beta &= 1,\dots,m . \end{aligned} \tag{L.23}$$

Finally, the initial conditions for the weights of the last layer $a_\alpha(0)$ are completely arbitrary. The solution of the DMFT equations gives access to the dynamics of the train and test error. The train error as a function of time is defined as

$$e_{\mathrm{tr}}(t) = \lim_{d\to\infty} \widehat{\overline{\mathscr{R}}}_n(t) . \tag{L.24}$$

A simple way to derive the expression of $e_{\mathrm{tr}}$ as a function of the solution of the DMFT equations in the $d\to\infty$ limit is to consider a deformation of Eq. (L.5) which consists in replacing

$$\exp\left(-n\int_{\hat a} \widehat{\overline{\mathscr{R}}}_n(\hat a)\right) \to \exp\left(-n\int_{\hat a} P(\hat a)\widehat{\overline{\mathscr{R}}}_n(\hat a)\right) . \tag{L.25}$$

For $P(\hat a) = 1$ we get back the original expression. The main idea of the derivation is to use $P(\hat a)$ as a source field. In particular we have that

$$e_{\mathrm{tr}}(t) = -\int \mathrm{d}\theta_a \left.\frac{\delta}{\delta P(\hat a)} \ln Z_{dyn}[P]\right|_{P=1} . \tag{L.26}$$

Note that the deformed dynamical partition function $Z_{dyn}[P]$ does not equal 1 for generic $P$ so that the formula above makes perfectly sense. The deformation of the partition function produces a deformation of $S_{dyn}$ in Eq. (L.7) which consist in replacing

$$\frac{\overline\alpha d}{2} \ln\det(\boldsymbol{I} + \Sigma_+) \to \frac{\overline\alpha d}{2} \ln\det(\boldsymbol{I} + \Sigma_*)$$

$$\Sigma_*(\hat a,\hat b) = P(\hat a)\Sigma_+(\hat a,\hat b) . \tag{L.27}$$

Performing explicitly the derivatives with respect to $P$ one gets

$$e_{\text{tr}}(t) = \frac{1}{2} \int_0^t \left[ R_A(t,s)\Sigma_C(t,s) + C_A(t,s)\Sigma_R(t,s) \right] \mathrm{d}s . \tag{L.28}$$

The computation of the test error can be done in analogous way

$$
\begin{aligned}
e_{\text{ts}}(t) &= \lim_{d\to\infty} \frac{1}{2}\mathbb{E}\left[ \left( y_{\text{new}}) - y_{\text{new}}^{(s)} \right)^2 \right] \\
&= \frac{1}{2}\left[ \tau^2 + \frac{1}{k}h_t(1) + \frac{1}{m^2}\sum_{ll'}^m h[C_{ll'}(t,t)] - 2\frac{1}{m}\sum_l^m \hat{\varphi}(r_l(t)) \right] .
\end{aligned}
\tag{L.29}
$$

The average in Eq. (L.29) is performed over the training set and an additional datapoint, not presented in the training set and having the same statistical structure.

In summary, the solution of the DMFT equations gives access to the train and test error dynamics in the large dimensional limit. These equations can be integrated numerically very efficiently. Our goal is to understand their behavior for infinite number of neurons, $m \to \infty$ at fixed sample complexity $\alpha$. We will be mostly interested in two types of questions: first, given a dataset that is pure noise, what are the sample complexities at which the network is able to interpolate the dataset. Second: given a dataset built out of a single index process what is the dynamics of the test and train error.

