# OpenReview forum: "Dynamical Decoupling of Generalization and Overfitting in Large Two-Layer Networks"
_NeurIPS.cc/2025/Conference — NeurIPS 2025 oral_

### Official Review · Reviewer_FYDY · 2025-06-20

**Clarity:** 3
**Significance:** 3
**Originality:** 3
**Rating:** 5
**Confidence:** 3

**Summary:**

This paper analyzes training dynamics in overparameterized two-layer networks using Dynamical Mean Field Theory (DMFT). It reveals a timescale separation and describes in details the corresponding dynamical regimes, under both lazy and mean-field initializations.

**Questions:**

1. The paper identifies a phase where test error increases with training ("feature unlearning"), necessitating early stopping. However, benign overfitting (where test error remains low despite interpolation) is common in practice. What specific aspects of the model or data assumptions are responsible for the observed non-benign overfitting?

2. While DMFT is proven to be rigorous for related settings, could the authors clarify which specific predictions in Section 2 are expected to be rigorous and which parts are based on further assumptions or numerics and might have potential limitations? What are the main barriers to full mathematical rigor for these results?

3.  Given DMFT can handle discrete steps, what are the key challenges in extending the core findings to discrete GD or SGD? Would you expect significant qualitative differences?

4. This paper takes n,d to infinity and then m to infinity. Recently there are a lot of papers taking the limit m,n,d to infinity together with m/d, n/md fixed. Do we expect them to be equivalent?

5. L159: what do you mean by fixing a(t)? Does this mean the second-layer weights are fixed while only the first-layer weights are trained? Is every element of the second layer the same?

6. Could you comment on eq.2.6? How far is it from Bayes optimal and what determines the gap?

**Ethical Concerns:**

["NO or VERY MINOR ethics concerns only"]

**Final Justification:**

This is a solid paper and I recommend acceptance. I do not give score 6 because of the existing limitation of DMFT (e.g. Q2 and Q4).

**Limitations:**

The authors should comment on whether the results might change when we use GD/SGD, which parts of the results are conjectured to be rigorous, and compare the results with existing literature, e.g. on implicit biases and benign overfitting

**Quality:**

3

**Strengths And Weaknesses:**

Overall this is a good paper that describes different training regimes in detaild. My main concern is that the presentation in Section 2 lacks clarity on which findings stem directly from the (conjectured to be rigorous) DMFT equations versus those derived from scaling hypotheses or numerical integration (which may have limitations). Moreover, it is worthwhild to compare with existing results on the implicit biases of GD, benign overfitting, etc.  Specifically, why does benign overfitting not occur here?

---

> ### Author Rebuttal · Authors · 2025-07-29
>
> $\textbf{Questions}$
>
> 1. In our setting we do observe benign overfitting. At the end of the "feature unlearning" phase, the test error is the one achieved by the neural tangent model. This model has benign overfitting [see, e.g.
> Ghorbani et al 2016; Montanari, Zhong, 2020].
> Namely, for $n\to\infty$ the excess error converges to $0$.
> However this happens only for $n$ superpolynomial in $d$. (The phenomenology established in [Montanari, Zhong, 2020]
> for  neural tangent models matches the general framework for benign overfitting in [Bartlett et al. 2019]
> where is known that the excess risk vanishes only very slowly with sample size)
> On the other  in the mean field/feature learning regime of training the excess error vanishes already when
> $n\gg d$.
>
> 2. There are several possible routes to making the analysis of Section 2.
> While a full discussion would be too complex, a possible route would be:
> (i) Use the DMFT equations for actual 2-layer networks
> that were proven in Celentano, Cheng, Montanari, arXiv preprint arXiv:2112.07572, 2021
> (instead of the simplified DMFT we use in our paper).
> (ii) Carry out the singular perturbation analysis of these equations.
> A first step towards (ii) would be to rigorize the singular perturbation analysis in the simplified
> DMFT that we use in our paper.
> Even in this simplified setting, step (ii) is the most challenging because
> it requires to control the solutions of complex non-linear PDEs in the long time limit.
> The rigorous results in Section 3 partially address this point.
>
> 3. There are no particular complications that make the analysis more complicated for SGD. Indeed, for $m=1$ and single index data, the DMFT analysis of SGD dynamics was performed in:
> Kamali, Urbani, arXiv preprint arXiv:2309.04788, 2023. We do expect a similar behavior.
>
> 4. The rigorous results in Section 3 imply that the results obtained from the DMFT analysis have a
> larger domain of applicability. In particular, no-overfitting take place for $t=o(m^{1/4})$ and long as
> $n/md = \alpha$ is kept constant (regardless of the order of limits).
> Similarly, the results in the first regime of DMFT are correct
> for $t=O((\log m)^c)$, and a small $c$  (for $n/md = \alpha$  constant
> regardless of the order of limits).
> In particular, these claims apply to cases in which $m/d$ is kept constant.
> While we do not characterize the full domain of applicability
> of our analysis, these results imply that it is broader than could be assumed from the original derivation.
>  We plan to make this point clearer by making explicit the dependence of the results in section 3 on $n$, $d$.
>
> 5. The reviewer is correct. In line 159, we fix the second layer weights at initialization and do not train them.
> Since we use a symmetric initialization, see Sec. C.4, all the second layer weights are the same.
>
> 6. The Bayes optimal value of the test error is $\tau^2/2$ (in the limit $n/d\to\infty$ considered here).
> This is achieved during the feature learning phase of training. In the lazy phase, the test error
> increases because the model converges to a neural tangent kernel model that only learns the linear component of the target.
>
> $\textbf{Limitations}$
>
> We appreciate the referee comments. As discussed above, we will try our best
> to address them and discuss limitations/extensions.

---

> > ### Comment · Reviewer_FYDY · 2025-08-05
> >
> > Thank you for your responses. I'm happy to keep my score.

---

### Official Review · Reviewer_97D6 · 2025-06-26

**Clarity:** 2
**Significance:** 3
**Originality:** 3
**Rating:** 5
**Confidence:** 4

**Summary:**

This paper uses dynamical mean-field theory to study the gradient flow dynamics of training a two-layer neural network on either random noise or a Gaussian multi-index model. The authors study both the lazy and the feature learning regimes, and characterize the timescales at which feature learning and overfitting occur.

**Questions:**

* It would be nice to state that the assumption $\mathbb{E}[\varphi(G)G] \neq 0$ corresponds to Information Exponent = 1 in Ben Arous et al. 2021, which would give the readers a context of the regime of problems this paper studies compared to the broader literature of high-dimensional SGD dynamics for learning Gaussian single-index models.

* The arguments in the main text seem to suggest usually $a_i(t)$ is constant in $i$ for all $t$, which is understandable in the setting when $a_i$ is close to initialization, but how can one intuitively see this over longer training times?

* What is z at the bottom of page 6? Does it represent rescaled time? It was a bit confusing to suddenly see z.


G. Ben Arous et al. "Online stochastic gradient descent on non-convex losses from high-dimensional inference"

**Ethical Concerns:**

["NO or VERY MINOR ethics concerns only"]

**Final Justification:**

I didn't have any major concerns with the paper in my original review, and the authors responses have also addressed my minor comments.

**Limitations:**

yes

**Paper Formatting Concerns:**

No concerns.

**Quality:**

3

**Strengths And Weaknesses:**

The most interesting aspect of the paper to me is the observation that long training time in the feature learning regime can also lead to overfitting, and some notion of early stopping is necessary to achieve optimal test performance. This is in contrast with most recent analyses of training two-layer neural nets to learn multi-index models, which either use $L_2$ regularization or constrain second layer weights, and never notice overfitting as $t/m \to \infty$. Relative to the these analyses, the DMFT machinery used here is powerful enough to characterize many different behaviors without significant modifications of the gradient flow.

My main concern with the paper is its presentation and clarity. The authors mostly describe different dynamical regimes informally throughout the paper, and some formal statements would help better understand each regime. Even having informal theorem statements could help to better keep track of different phenomena that occur during each regime. Also, from reading the main text alone it is not entirely clear which parts of the analysis are rigorous, which parts are non-rigorous calculations and which parts follow from solving DMFT numerically.

---

> ### Author Rebuttal · Authors · 2025-07-29
>
> $\textbf{Weaknesses}$
>
> We will address the problem of readability of the manuscript.
> As other reviewers requested we plan to:
>
>
> 1. Clarify the notations and terminology.
>
> 2. State formally a the most important conjectures/results arising from the DMFT analysis.
>
> 3. We will connect better the main text to the appendix.
>
> 4. Try to make the best selection for the material in main text.
>
>
>
> $\textbf{Questions}$
>
> 1. In the revised version of the manuscript we will clarify the role of
> the assumption $\mathbb{E}(\varphi(G)G)\neq 0$ and refer to earlier literature concerning this point.
>
> 2. The fact that $a_i$ is independent of  $i$ is a consequence of two facts.
> First, under the initialization, neurons are exchangeable (in the sense of probability theory),
> and hence they remain exchangeable at all subsequent times. As a consequence of exchangeability
> ${\mathbb E} a_i(t)$ does not depend on $i$.
> Second, second layer weights $a_i(t)$ concentrate around their expectation for any $t=O(1)$ as
> $n,d\to\infty$. Within DMFT, this is discussed in sec. C.4. We could expand on this point from a mathematical perspective.'
>
> The claim that (under the symmetric initialization that we use)
> the $a_i$ concentrate around a value independent of $i$ for $t=O(1)$
> as $n,d\to\infty$ can in fact be proven rigorously.
>
> 3. $z$ is the scaled time. We will make the notation clearer in the revised version.

---

> > ### Comment · Reviewer_97D6 · 2025-08-04
> >
> > Thank you for your responses, I'll keep my positive score.

---

### Official Review · Reviewer_hZhE · 2025-06-30

**Clarity:** 2
**Significance:** 3
**Originality:** 3
**Rating:** 5
**Confidence:** 3

**Summary:**

This paper presents an analytical study of the learning dynamics of a two-layer network, performing a supervised regression task, in the limit of large input size ($d$), large training set size ($n$), and (eventually) large number of hidden units ($m$). The scenario is (essentially) that of a teacher-student framework, with random gaussian inputs. The analysis is based on dynamical mean-field theory (DMFT), which reduces the training dynamics to a set of differential equations over a few selected macroscopic quantities. By manipulating the scale of the parameters initialization and analyzing different asymptotic limits, the authors identify several dynamical regimes in different scenarios, characterizing their properties in terms of training and test errors (and relatedly, feature learning and generalization gap). The findings of the DMFT analysis are supported by numerical experiments and some rigorous theorems.

**Questions:**

Main observations:
* The second layer weights, which play a crucial role in the analysis, are consdered as being initialized as either $O(1)$ or $O(\sqrt{m})$. But in a standard setup (with He initialization, for example) they would be initialized as $O(1/\sqrt{m})$ instead. I think this should be clarified.
* As I wrote above, I'm also unconvinced about the claim, first encountered in fig. 1 and then used in the discussion, that the $\ell_1$ norm of the second-layer weights is a proxy for model complexity. This should also be clarified/elaborated upon.
* As I already mentioned, the use of "mean field" or "lazy" initialization terminology, which is used ubiquitously, is never properly explained. So the reader finds e.g. fig. 2 and reads "Here we use mean field initialization", which was never defined.
* The details of the simulations are only given sparsely in the main text (e.g. in fig. 4 we get to know the size of $d$, but not in the other figures), and references to the appendix where the details are reported are missing. I don't think I have seen, or possibly I have missed, the place where the complete details of the simulations with the "actual 2-layer networks" trained with SGD are reported (fig. 2). What was the size, the activation function, the hyperparameters etc. Right now I don't think that the results are really reproducible. The settings used to produce each figure should be reported, with a clear reference.
* The choices of the $h(z)$ and $\hat{\varphi}(z)$ functions are never explained, and they sometimes change (in fig. 2 it's cubic, in fig. 3 it's quadratic, etc). I think there should be an explanation of how they were chosen, and what the choices entail in terms of the activation functions that give rise to them.
* In several places in the main text there is a reference to "SM", which should be the appendix. Also, in several places in the appendix, we find sentences like "see the appendices". But we are already in the appendices. Please review the pointers to the appendices and, where possible, add a reference to the relevant section. This would significantly improve the readability.
* At line 159 the fixed lazy regime is introduced, with $a(t)=\gamma \sqrt{m}$ non evolving with time. However, one could always absorb the entire weight in the activations $\sigma$ and set $a(t)=1$. Thus, evidently, some assumption about $\sigma$ and/or its input needs to be made, possibly encoded in the $h$ function. But I don't think I've seen such an assumption mentioned anywhere in the main text.
* I think regularization (beyond early stopping) should be discussed in the last part of the discussion, about overfitting and feature unlearning.

Minor observations:
- below line 60: calling alpha the "overparemeterization" parameter seems a bit strange, considering that it decreases with the number of parameters. Parameter load, maybe?
- line 94: one might mention that with the choice $k=1$ this is essentially a noisy "teacher" perceptron, as it's called in some related literature
- line 100: here the large $n, d, m$ limit is presented. But only much later it is clarified that $n,d \to \infty$ first at fixed ratio $\bar{\alpha}$, and only later is the limit $m \to \infty$ taken (at fixed $\alpha$, implying $\bar{\alpha}\to0$). This seems relevant as it underpins the whole analysis
- line 101: the rescaled $W_\mathrm{2nd}$ are analyzed using the norm-1. Why the norm-1 and not the more common norm-2?
- below line 120, eq. 2.2: I think the $\mathbf{\theta}_i$ in the second expression should be $\mathbf{w}_i$, as in eq. C.1, otherwise it doesn't make much sense. The same issue occurs in eq. B.4 in the appendix. Also, in eq. B.4, the quantity $a_i(t)$ is considered, which I think should be included here too, otherwise we don't get the full system. Also, the first expression is missing a closing $\rangle$.
- line 147: the notion of "good generalization" is a bit puzzling in the context of learning pure noise. If it's about the generalization gap, this can only be achieved by making the training error as bad as possible right? Which is hardly "good".
- line 165: I don't think that the quantity $\hat{\mathcal{R}}_n^g$ was ever mentioned in the main text, since it's part of the gaussian process description.
- the caption of fig.3 mentions an inset which is not there
- eq. 2.8 and line 199: suddenly there is no longer an explicit dependence of the quantities on $\alpha, \varphi, \tau$ etc. This is puzzling.

Some typos I found:
- line 100: "a $k=1$" should be "at $k=1$"?
- line 212: the limits for $t \to \infty$ are clearly wrong. Probably $m \to \infty$ was meant?
- line 237: "variables" -> "variable"
- line 257: missing a left $||$ for $\sigma^\prime$
- line 273: "scenario" -> "scenarios"
- line 525: is $\mathbf{u}_i$ here supposed to be $\mathbf{v}_i$?

**Ethical Concerns:**

["NO or VERY MINOR ethics concerns only"]

**Final Justification:**

After the discussion phase all my concerns have been addressed satisfactorily. I stand by my initial assessment of recommending acceptance, with minor changes required for clarity and reproducibility.

**Limitations:**

yes

**Quality:**

4

**Strengths And Weaknesses:**

Strenghts:
* Thorough, technically difficult and detailed analysis
* Non-trivial findings albrit in a simple model, advancing the knowledge of this class of systems and revealing some key features and dynamical regimes
* The overall analysis, its goals and the conclusions, are fairly well presented

Weaknesses:
* The details of the analysis are quite hard to follow at times. Mostly, I would say that the paper suffers from trying to cram a lot of content in the page limit, with a large appendix. In several places, it seems that the division between what went in the main text and what went in the appendix was a bit rushed. Also, some jargon is not explained, e.g. the use of "mean field initialization" vs "lazy inizialization" must be deduced from context. Overall, with the paper as it currently stands, the main text does not stand on its own. Furthermore, the main text does not point to specific places in the appendix where to look up the missing information (such as the details of the simulations, among other things), which is relevant when the appendix spans tens of paes.
* In terms of significance, I can't help but thinking that regularization would have a dramatic effect on the findings. Also, I'm not entirely convinced by the use of the norm of the second layer as a proxy for the model complexity.

---

> ### Author Rebuttal · Authors · 2025-07-29
>
> $\textbf{Weaknesses}$
>
> - $\textit{Writing.}$ We are going to improve the revised version of the manuscript in two ways:
>
> 1. Clarify the notations and terminology.
>
> 2. We will connect better the main text to the appendix.
>
> 3. Try to make the best selection for the material in main text.
>
> - $\textit{Regularization}$. We point out that we regularize explicitly the first layer weights by constraining them to have unit norm.
> Instead we do not regularize the second layer weights.
> From a technical viewpoint, adding a regularization for second-layer weights
> leads to a simple modification of the same DMFT equations.
> The final result will be that such a regularization stops the growth of second-layer weights at a value depending on the regularization parameter.
> For suitable choices of this parameter the unlearning/overfitting dynamical regime will become milder or totally eliminated. Even larger regularization strength will affect the first phase of training and hence induce underfitting.
> We regard this as a natural (and quite simple) follow-up project.
>
> - $\textit{Measure of complexity.}$
> The fact that the $\ell_1$ norm of second layer weights is a bound on complexity is a classical
> result by Bartlett, see Ref.[6] (cf lines 145-150 of the manuscript). In principle it could be that the
> complexity is significantly smaller because of some special structure in the first layer weights.
> However we see no evidence of this in our analysis of the dynamics:
> 1. First layer weights remain isotropically distributed (apart from the projection on the direction of the signal). Also, by construction their norm is fixed.
> 2. Increase of the generalization error is accurately tracked by the increase in norm of second layer weights.
> This is shown very clearly in Figure 5 right frame.
>
> $\textbf{Questions}$
>
> 1. The definition of the network in Eq.(1.1) has a factor $1/m$ in front of the sum. Therefore if one defines the second layer weights as $A_i=a_i/m$, then our
> "lazy initialization" corresponds to He initialization  $A_i =  \mathcal{O}(1/\sqrt m)$,
> while our "mean field initialization" corresponds to $A_i\sim \mathcal{O}(1/m)$.
> (We point out that the distinction between "lazy" and  "mean field" is standard in the theory literature.)
>
> 2. See discussion above for the use of the norm of second layer weights as a measure of complexity.
>
> 3. We will add definitions for this terminology: we agree that while common in the theory literature
> it is best to explain it for a broader audience.
>
> 4. The connection between activation function and the function  $h(z)$
> is straightforward the coefficients of polynomial expansion of the latter are the square of the Hermite coefficients of the former. A similar relation holds for $\widehat\varphi$. We will clarify this point.
> We also plan to add the details of the numerical simulations.
>
> 5. The choice of the functions $h(z)$ and $\widehat \varphi(z)$ in the figures was dictated by desire to select examples such that (i) The qualitative behavior of learning curves is the same as for generic such functions;
> (ii) The relevant phenomena are clearly visible.
> We underline that the singular perturbation theory analysis applies to general
> $h(z)$ and $\widehat \varphi(z)$
>
>
> 6. We apologize for these misprints. We will change "SM" to "the appendix" and replace the "see the appendix" present in the appendix with the corresponding sections.
>
> 7. Throughout the paper, it is understood that the activation function does not change with the width $m$.
> We will clarify this in the text.
>
>
> 8. We will add a discussion the effect of adding an explicit  regularization term in the dynamics.
>
>
> $\textbf{Minor comments.}$
>
> Thank you for the detailed comments. We will do our best to address them.

---

> > ### Comment · Reviewer_hZhE · 2025-08-03
> >
> > Thank you for your responses, which address nearly all my initial concerns. However, I think one of my points was not adequately answered, this one:
> > > The choices of the $h(z)$ and $\hat{\varphi}(z)$ functions are never explained, and they sometimes change (in fig. 2 it's cubic, in fig. 3 it's quadratic, etc). I think there should be an explanation of how they were chosen, and what the choices entail in terms of the activation functions that give rise to them.
> >
> > To which the authors replied:
> > > The connection between activation function and the function $h(z)$ is straightforward the coefficients of polynomial expansion of the latter are the square of the Hermite coefficients of the former. A similar relation holds for $\widehat\varphi$. We will clarify this point. We also plan to add the details of the numerical simulations.
> >
> > This is now what I was asking. I'm not wondering about how $h(z)$ would look like for a given $\sigma(z)$, that is indeed straightforward; I'm asking why you chose the specific forms presented in the paper, and (to help understanding without the reader having to "reverse-engineer" it) what $\sigma(z)$ do they correspond to.
> >
> > For the rest, I'm of course going to just trust that clarity, organization and reproducibility will be improved in the final version, which I don't have access to due to NeurIPS review process, and (having also gone through the other reviews and rebuttals) I'll confirm my initial assessment of the merits of the work and suitability for publication.

---

### Official Review · Reviewer_n5Nb · 2025-07-01

**Clarity:** 3
**Significance:** 2
**Originality:** 3
**Rating:** 4
**Confidence:** 4

**Summary:**

This paper analyzes the dynamic behavior of a two-layer neural network under gradient flow training through the dynamic mean field theory, and finds that there is a clear time scale separation in the training process: the early stage is the feature learning stage, and the later stage enters the overfitting and "feature forgetting" stage, which further reveals that the generalization ability is closely related to the training dynamics. This is the first paper that theoretically explains the impact of initialization, training time, and network size on overfitting and generalization.

**Questions:**

1: What challenges arise when extending the analysis to multi-index models (e.g., k=2 and orthogonal), and are there simple examples illustrating the difficulty?

2: Is the Gaussian input assumption crucial, or would the results generalize to more realistic input distributions?

3: Can the theory suggest a practical, observable criterion for determining when to apply early stopping?

**Ethical Concerns:**

["NO or VERY MINOR ethics concerns only"]

**Final Justification:**

The authors addressed most of my concerns in the rebuttal, particularly regarding the Gaussian input assumption and generalization of the DMFT framework to multi-index models. While I still find the modeling assumptions restrictive and the lack of real-data validation limiting, I acknowledge the theoretical depth and contribution of this work.

**Limitations:**

No. The paper does not explicitly discuss its limitations or potential societal impacts. It would be helpful if the authors could comment on the constraints of the single-index assumption and Gaussian input setting, as well as the applicability of their analysis to real-world deep learning scenarios.

**Quality:**

3

**Strengths And Weaknesses:**

Strengths:

1: The paper identifies a clear time scale separation between feature learning and overfitting in two-layer neural networks.

2: The paper leverages dynamical mean field theory to provide a precise and quantitative analysis of training dynamics.

3: The paper provides a theoretical explanation for early stopping, initialization effects, and scaling laws.

Weaknesses:

1: The analysis is limited to two-layer networks and gradient flow.

2: The use of a single-index model as a target function limits the generality of the results.

3: The assumption that input data is Gaussian simplifies analysis.

4: Empirical validation is mostly numerical and lacks experiments on real-world datasets.

---

> ### Author Rebuttal · Authors · 2025-07-29
>
> The reviewer points out a general weakness of our work.
> Admittedly, this is a theoretical work that focuses on:
>
> 1. A simple model of data (multi-index model with Gaussian features)
>
> 2. A simple neural network (2-layer fully connected).
>
> Focusing on a simple example is the price we pay for being able to carry out an analytical study of the gradient descent dynamics.
>
> $\textit{Earlier theoretical work was limited to significantly simpler cases}$ (eg linear models, networks with 1 neuron) or special regimes (eg neural tangent limit).
>
> Despite the simplicity of the setting, there is a lot that is poorly understood and a lot to learn from this setting. As we explain starting with lines 63 and 64, several of the fundamental question in this
> model generalize to more complex ML models and the conceptual picture we develop generalizes as well.
>
> Concerning the empirical validation, we do validate our results against numerical simulations on two layer neural networks, see for example Fig. 2 and 4.
>
> We limit ourselves to simulations with synthetic data because our theory can only capture the behavior on real data at a qualitative level. In machine learning, this is nearly always the case for a completely analytical theory. The purpose of our simulations is to verify that various approximations made in our analysis are indeed accurate.
>
> $\textbf{Questions.}$
>
> 1. Both the general DMFT theory and the rigorous analysis results of Section 3 apply to general multi-index models.
> Again, for moderate time-scales and large $m$, the dynamics is captured by the mean field
> theory of [Mei, Montanari, Nguyen, 2018], [Chizat, Bach, 2018]. If the latter converges to vanishing training error in (rescaled) times of order one, then our analysis extends verbatimly. However, in multi-index models
> convergence can take time slowly diverging with $d$. We do not expect that this will change the
> qualitative picture we derive, but the analysis becomes technically more complicated, to account for those cases.
>
> 2. We assume isotropic Gaussian feature vectors ${\boldsymbol x}_i$ Universality
> results can be used to show that the same DMFT captures the dynamics when the ${\boldsymbol x}_i$'s have independent
> entries (see, e.g. [Celentano, Cheng, Montanari, 2021]). Certain distributions with non-identity covariance can be
> also treated (e.g. linear transforms of iids), but the analysis becomes more intricate
> and we do not expect qualitatively new insights.
>
> 3. A general method that is supported by general theory is to monitor the error on a validation set and stop GF when this is minimal. Interestingly, our theory indicates that ---for large networks---
> the validation error will stay close to the minimum for a large interval of time.
> This suggests that large networks are naturally stable with respect to the choice of stopping time.

---

> > ### Comment · Reviewer_n5Nb · 2025-08-05
> >
> > The authors addressed most of my concerns in the rebuttal, particularly regarding the Gaussian input assumption and generalization of the DMFT framework to multi-index models. While I still find the modeling assumptions restrictive and the lack of real-data validation limiting, I acknowledge the theoretical depth and contribution of this work, so I will raise my score.

---

### Official Review · Reviewer_jCsC · 2025-07-02

**Clarity:** 3
**Significance:** 4
**Originality:** 4
**Rating:** 6
**Confidence:** 4

**Summary:**

The authors study a dynamical mean field theory (DMFT) characterization of two layer networks, where a few simplifying assumptions are made to allow analysis of large input dimension $d$, number of data points $n$, hidden layer width $m$, and training time $t$. Note this type of joint limits has been very difficult to study for a long time, and any analysis of this kind is highly desirable.

The authors found several interesting observations, in particular, the fact that overfitting happens on a slower time scale than feature learning, and that training for longer leads to feature unlearning and an increase in generalization error. The paper overall uses a combination of non-rigorous calculations and rigorous proofs, where the non-rigorous aspects include assuming the residuals forms a Gaussian process, several ansatz were used to analyze the DMFT equations, and some perturbative analysis. However, the authors did a good amount of simulations to justify quite a few of these non-rigorous approaches.

**Questions:**

Given the technical depth of this paper, I have many questions, but they will not affect my recommendation. I may also continue to ask further questions during the discussion period as I plan on reading this paper in more careful detail than it is required for the review.

On the order of limits

I believe the authors studied the DMFT equations originating in the regime where only $n,d \to \infty$, and $m,t$ stayed finite, and sequentially studied the regimes where $m,t$ are large. Do we know if the limits are different if we take $m\to\infty$ first, or jointly with $n,d,m$ together? I think at least in the case where $n,t$ are finite, the order of $d$ and $m$ do make a difference in the mean field regime.

On Figure 10

I am a bit confused reading Figure 10, as this should provide an empirical justification of the sharp transition, but I don't think I understand how this relaxation time is supporting this claim here. Can you provide a more intuition explanation of this Figure and what it is trying to achieve?

On Section G.2.1

The authors studied the first dynamical regime for the mean field initialization here, where $t = \Theta(1)$. In particular, there was a very strong simplification here where using a set of ansatz, the authors drastically simplified the DMFT systems (with memories) to a system of $k+1$ ODEs. I would like to understand this set of ansatz better, can you provide a more intuitive explanation of how you arrived at this set of ansatz, and why should we expect a simplification to an ODE system in constant order time?

**Ethical Concerns:**

["NO or VERY MINOR ethics concerns only"]

**Final Justification:**

I believe the submission was already in a good state before the rebuttal, minus some clarification points which the authors acknowledged as helpful. I remain supportive of accepting this paper.

**Quality:**

3

**Strengths And Weaknesses:**

Strengths

 - A fairly comprehensive analysis when the width $m$ and training time $t$ is large, which resulted in a very interesting observation of the slower overfitting time scale, as well as the feature unlearning behaviour.
 - A very good collection of simulations, some of them to justify the non-rigorous approaches, but also to complete the overall picture in understanding the problem setting.
 - In combination with non-rigorous analysis, there are also rigorous results lower bounding the overfitting timescale, which helps to demonstrate where rigorous mathematics are currently at and their limitations.

Weaknesses

While it's understandable that the NeurIPS template is not very forgiving for long technical theory papers, I did find the paper somewhat difficult to read. Although I was able to figure some of these things out by asking a series of questions to LLMs, I still think the authors should fix the issues below to improve the reading experience:
 - The main text did not mention the Gaussian approximation done in Appendix A. I don't think it's particularly problematic to make this assumption for the sake of a tractable calculation, I just prefer if the authors can reference this aspect in the main text.
 - For the Gaussian process covariance, I would also prefer if the authors can specific what the expectation is over, in e.g. equation A.4.
 - Also the authors mentioned that the approximation error is vanishing on constant time scales and small on large time scales. It would be nice to reference which Figures that contained these simulations.
 - In Figure 2, the authors defined an $h(z)$ function without specifying how the simulations were done for a real neural network. I understand it's possible to construct an activation function that achieves this correlation function, but it would be more clear to me if the authors specified exactly which network is simulated.
 - In Figure 2, the authors also plotted the value of $a$ as a scalar. This was confusing to me because I thought the weights were a vector. Of course, after I read the argument that since weights are exchangeable, all the $a_i$'s must be the same throughout training, but I think it would be more clear if the authors can point this out in the figure caption.
 - The use of $\gamma_{GF}$ was confusing to me. I think if the authors say something along the lines of "there exist a constant $\gamma_{GF}$ such that a sharp transition happens at..." that would clear up my confusion, because this variable was not defined but used directly.

Additionally, I think it might be helpful to move/copy some of the Figures from the appendix to the main text. In particular, I think Figures 30 and 31 are quite clear at demonstrating the feature unlearning effect, hence justifying the third component of the cartoon Figure 1. The fact that a theoretical approach can predict this surprising phenomenon is in my opinion very impressive, and should perhaps be demonstrated early on in the text.

---

> ### Author Rebuttal · Authors · 2025-07-29
>
> $\textbf{Weaknesses.}$
>
> - The Gaussian approximation was mentioned in line 123 of the manuscript. However this a short statement and  we plan to expand on it.
>
> - The expectation in Eq. A.4 is over the data distribution. We will clarify this and related points.
>
> - We will add a reference to the figure showing that approximation error is vanishing on constant time scales  (this can be seen, for instance, in Fig 4 right frame, by noting that the asymptote corresponds to the Bayes error.)
>
> - In the revised version of the manuscript we will specify the activation used in simulations.
>
> - Indeed, because of the symmetric initialization, second layer weights
> concentrate around a common deterministic function $a(t)$ for all $t$
> that remain bounded as $n,d\to\infty$. This is further discussed in Appendix
> C.4, but we will make it more explicit in the main text.
>
>
> -  $\gamma_{GF}$ is the threshold defined via Eq.(2.3) and lines 166-167:
> we will clarify this threshold phenomenon.
>
> - We agree  that it is remarkable that our theory is capable to predict the unlearning phenomenon.
> Moving figure 30 and 31 to the main text is probably hard because of space constraints. Note that
> Figure 5 also illustrate clearly the same phenomenon.
>
>
> $\textbf{Questions.}$
>
> - $\textit{Order of limits.}$ As the reviewer writes, in the DMFT analysis we take first $n$ and $d$ to infinity and then we study the large $m$ limit, allowing possibly the time to diverge with $m$.
> However, the rigorous results in Section 3 imply that the results obtained from the DMFT analysis have a
> larger domain of applicability. In particular, no-overfitting take place for $t=o(m^{1/4})$ and long as
> $n/md = \alpha$ is kept constant (regardless of the order of limits).
> Similarly, the results in the first regime of DMFT are correct
> for $t=O((\log m)^c)$, and a small $c$  (for $n/md = \alpha$  constant
> regardless of the order of limits). While we do not characterize the full domain of applicability
> of our analysis, these results imply that it is broader than could be assumed from the original derivation.
> In the revised version  we make the dependence on $m$, $d$ and $n$ more explicit in Sec. 3.
>
> - $\textit{Figure 10.}$ This figure refers to a setting in which second-layer weights are fixed to $\gamma\sqrt{m}$.
> When the control parameters (the overparametrization ratio $\alpha$ or the scale $\gamma$) are such that
> gradient flow coverges to zero train error, this convergence is exponentially fast in time.
> The exponential convergence rate can be used to define a relaxation timescale that we estimate as detailed in lines 858-868.
> When either $\gamma$ or $\alpha$ cross the interpolation threshold, the relaxation time diverges.
> In the inset of Figure 10 we plot the behavior of the relaxation time as a function of the control parameters.
> This allows to determine the interpolation threshold $\alpha_{GF}$.
> In the main panel we plot the relaxation time as a function of $\alpha_{GF}-\alpha$, to verify that the relaxation time diverges as
> $(\alpha_{GF}-\alpha)^{-\nu}$. The data suggest that $\nu$ is close to $2$.
> In the revised version of the appendix we will add a plot of the train error in linear-logarithmic scale. This will confirm the exponential relaxation to zero train error and  clarify Figure 10.
>
> - $\textit{Section G.2.1}.$ The ansatz used on this timescale is presented in
> Eqs.~(G.39)-(G.41). Among the scaling forms used in various dynamical regimes, we consider this particularly simple. We simply posit that all quantities have a finite limit as $m\to \infty$, without time rescaling. An indication that this should be
> the case is already contained in the mean field theory [Mei, Montanari, Nguyen, 2018; Chizat, Bach 2019] which obtains a finite non-trivial limit as
> $m\to\infty$ and $t=\Theta(1)$.
> Notice however that  the out-of-diagonal response $R_o$ turns out to be of order $1/m$ in this ansatz. This is intuitive because $R_o(t,s)$ captures the effect of changing the drift for the $j$-th neuron at time $s$ on the $i$-th neuron, $i\neq j$ at time $t$.
> It is reasonable to think that this effect is of order $1/m$ because each neuron is multiplied by a factor $1/m$ in the overall function.

---

> > ### Comment · Reviewer_jCsC · 2025-08-04
> > **Response**
> >
> > Thank you for the clarifications. I maintain my view that this paper is very strong and should be accepted. I also look forward to the updated version and reading the details more carefully.

---

### Note · Authors · 2025-08-14

Dear AC,
the reviewers were convinced by the novelty and impact of our work.
We will produce a revised presentation of our results following their advice.
Thanks for the evaluation process.

---

### Decision · Program_Chairs · 2025-09-17

**Decision:**

Accept (oral)

**Comment:**

This paper provides a precise characterization of the learning dynamics of two-layer networks via dynamical mean field theory (DMFT), under the joint regime of large input dimension $d$, sample size $n$, hidden layer width $m$, and training time $t$. Analyzing these joint limits is highly non-trivial, and the authors develop a refined approach to handle them. The findings are interesting and novel: the learning dynamics can be divided into three distinct stages—(i) mean-field feature learning at $O(1)$ time; (ii) extended feature learning for $1 \ll t \ll m$; and (iii) overfitting and feature unlearning when $t \gg m$. It can help to **mathematically explain several major phenomena in deep learning theory**, including lazy vs. non-lazy training, benign overfitting, double descent, and implicit bias or regularization (early stopping).

The main weakness of the work, as highlighted by reviewers, is clarity of presentation. Since DMFT is unfamiliar to much of the machine learning theory community, the paper would benefit from significant polishing and from making the technical framework more accessible as mentioned by the reviewers. Besides, I also encourage the authors to incorporate the reviewers’ feedback, for instance, the order of the joint limits, the relationship to the information exponent, the calculation between $h$ and activation function, the extension to gradient descent, and the connection between model capacity $||{\bf a}||_1$ (this is actually equivalent to the $\ell_1$-path norm given $||{\bf w}_i||=1$ in the first layer and path-norm has demonstrated to be good model capacity, both empirically and theoretically).

From my own reading, I also suggest consolidating the six motivating questions in the introduction into one or two central guiding questions. Framing the work around “how to precisely characterize learning dynamics for generalization, and identify the phase transition within implicit bias” would make the paper more focused and impactful. I also have some concerns but just name a few:

- Line 89: the authors claim "the **largest** possible gap between linear/kernel learning" (for Gaussian data). It seems too strong without further justification or analysis on "**largest**" if we use other symmetric distributions in recent work for single-index models.

- Line 1496: it seems that $Q_{ll'}$ should be expressed as a sum of the two terms $\tilde{Q}$ and $r_l r_{l'}$, not a difference; otherwise it contradicts Eq. (K.8), which uses $h(\tilde{Q}+r_l r_{l'})$.

---

Overall, this paper develops a promising DMFT-based framework for characterizing the nonlinear learning dynamics of two-layer neural networks. It lays mathematical foundations for understanding **nonlinear dynamics, major phenomena, and scaling laws beyond linear models** and has thus strong potential impact in the ML theory community. All of the reviewers vote for acceptance and I recommend this paper for oral presentation.